# When Data is the Algorithm: A Systematic Study and Curation of Preference Optimization Datasets

**Aladin Djuhera**[1], **Farhan Ahmed**[2], **Swanand Ravindra Kadhe**[2], **Syed Zawad**[2],
**Heiko Ludwig**[2], **Holger Boche**[1]
[1] Technical University Munich `{aladin.djuhera,boche}@tum.de`
[2] IBM Research `{farhan.ahmed,swanand.kadhe,szawad,hludwig}@ibm.com`

## Abstract

Aligning large language models (LLMs) is a central objective of post-training, often achieved through reward modeling and reinforcement learning methods. Among these, direct preference optimization (DPO) has emerged as a widely adopted technique that fine-tunes LLMs on preferred completions over less favorable ones. While most frontier LLMs do not disclose their curated preference pairs, the broader LLM community has released several open-source DPO datasets, including TuluDPO, ORPO, UltraFeedback, HelpSteer, and Code-Preference-Pairs. However, systematic comparisons remain scarce, largely due to the high computational cost and the lack of rich quality annotations, making it difficult to understand how preferences were selected, which task types they span, and how well they reflect human judgment on a per-sample level. In this work, we present the first comprehensive, data-centric analysis of popular open-source DPO corpora. We leverage the Magpie framework to annotate each sample for task category, input quality, and preference reward, a reward-model-based signal that validates the preference order without relying on human annotations. This enables a scalable, fine-grained inspection of preference quality across datasets, revealing structural and qualitative discrepancies in reward margins. Building on these insights, we systematically curate a new DPO mixture, **UltraMix**, that draws selectively from all five corpora while removing noisy or redundant samples. UltraMix is 30% smaller than the best-performing individual dataset yet exceeds its performance across key benchmarks. We publicly release all annotations, metadata, and our curated mixture to facilitate future research in data-centric preference optimization.

## 1 Introduction

Learning from preference feedback or commonly called Reinforcement Learning from Human Feedback (RLHF) is an important final step in aligning large language models (LLMs). This stage refines LLMs to enhance their performance on a wide range of downstream capabilities, including instruction following, math, and code (Wang et al., 2023; Shen et al., 2023).

Within preference tuning, two prominent approaches are *reward-based* and *reward-free* alignment. Reward-based methods train a reward model from preference data and optimize it using algorithms such as Proximal Policy Optimization (PPO) (Schulman et al., 2017). In contrast, reward-free methods, including Direct Preference Optimization (DPO) (Rafailov et al., 2023), remove the need for an explicit reward function. DPO, in particular, has become a favored technique for its efficiency and simplicity: models are directly trained to prefer one completion over another using curated preference pairs, and thus no reward model or policy rollouts are required (Ivison et al., 2024).

However, proprietary LLMs leverage vast undisclosed DPO corpora that remain inaccessible to practitioners, often because of licensing restrictions or intellectual property concerns. Nevertheless, the open-source community has made notable progress by releasing open DPO datasets, including *TuluDPO* (Lambert et al., 2025), *ORPO* (Labonne, 2024), *UltraFeedback* (Cui et al., 2024), *HelpSteer* (Wang et al., 2024a), and *Code-Preference-Pairs* (Vezora, 2024). These corpora have become essential

building blocks for many open models and contribute meaningfully to the broader reproducibility efforts in LLM preference tuning (Lambert et al., 2025; Bakouch et al., 2025; Allal et al., 2025).

Yet, systematic comparisons between DPO corpora remain largely absent, with existing analyses limited to small subsets and evaluated under different model architectures and training hyperparameters, leading to considerable methodological heterogeneity across studies. Thus, without a standardized basis for evaluation, it remains unclear which DPO datasets provide substantial benefits over others.

Another challenge is that while some datasets report their coarse compositions, they remain opaque at the sample level: rich annotations are mostly missing and only a few datasets provide explicit human-annotated preference scores to justify chosen completions, while others contain only binary pairs without ranking information, leaving it unclear how much better the preferred sample actually is. This lack of clarity hampers progress by making it difficult to design systematic curation recipes.

Thus, in this work, we take a quality-driven approach to bring diagnostic rigor, transparency, and reproducibility to effective preference data curation. Our main contributions are as follows:

- **Comparative Evaluation**: We present the *first systematic cross-analysis of five open DPO datasets*: TuluDPO, ORPO, UltraFeedback, HelpSteer, and Code-Preference-Pairs, spanning both general-purpose and domain-specific tasks. We conduct preference fine-tuning on eight different models of various scales and evaluate performance across 12 common benchmarks from the popular Open LLM Leaderboards (Fourrier et al., 2023; 2024) and two additional code generation benchmarks. By holding all training parameters constant, we allow for a fair comparison between DPO datasets.

- **Sample-Level Annotations and Analysis**: Using the Magpie framework (Xu et al., 2025), we annotate each preference pair with metadata for task category, difficulty, input quality, and *preference reward*, an independent, reward-model-based signal that helps evaluate whether chosen completions are indeed better justified than discarded ones. This allows us to assess preference signals at scale and to validate the reliability of the preference order, particularly when human annotations are unavailable or inconsistent. In our extensive analysis, we reveal substantial variation in dataset composition, quality, and alignment performance, both across tasks and model scales.

- **New DPO Mixture**: Using our extensive annotations, we construct **UltraMix**, a high-quality DPO mixture that draws from all five datasets and is systematically curated on a per-sample-level to retain only high-quality instructions, high-utility task categories, and coherent preference rewards, removing redundant or noisy samples in the process. UltraMix is *30% smaller than TuluDPO* and consistently achieves *better performance* on key benchmarks while improving task diversity.

Our extensive study shows several interesting findings: a) misaligned preference orders (i.e., the chosen answer is not always better) are common and degrade performance, b) prompt quality and preference rewards are strongly correlated, and c) filtering based on a single signal (prompt quality or preference reward) is insufficient and requires combining multiple signals for effective curation.

Our work provides practical guidance for curating effective data mixtures from open-source corpora. To support ongoing research in preference optimization, we publicly release our annotated versions of TuluDPO[1], ORPO[2], UltraFeedback[3], HelpSteer[4], Code-Preference-Pairs[5], and UltraMix[6].

## 2    BACKGROUND AND MOTIVATION

DPO is an offline RL approach that learns directly from preference feedback by optimizing policies on curated preference pairs. As a result, the quality and diversity of preference data are critical for improving model performance across a wide range of capabilities. Although several DPO mixtures have been released, no systematic performance comparisons exist that enable fair assessment, i.e., evaluations that fix training across datasets to remove cross-study heterogeneity. In particular, prior

---

[1]Annotated TuluDPO dataset: huggingface.co/datasets/aladinDJ/tulu-DPO-annotated
[2]Annotated ORPO dataset: huggingface.co/datasets/aladinDJ/orpo-DPO-annotated
[3]Annotated UltraFeedback dataset: huggingface.co/datasets/aladinDJ/ultrafeedback-DPO-annotated
[4]Annotated HelpSteer dataset: huggingface.co/datasets/aladinDJ/helpsteer-DPO-annotated
[5]Annotated Codepreferences dataset: huggingface.co/datasets/aladinDJ/codepreferences-DPO-annotated
[6]Annotated UltraMix dataset: huggingface.co/datasets/aladinDJ/ultramix-DPO-annotated

Table 1: DPO results for Llama-3.1-8B-TuluSFT and Qwen-2.5-7B-TuluSFT trained on TuluDPO, ORPO, Ultrafeedback, HelpSteer, and Code-Preference-Pairs, evaluated on Open LLM Leaderboards (averaged) and code tasks. The overall average is across all benchmarks. Best scores are in **bold**.

| | | Llama-3.1-8B-TuluSFT | | | | | | Qwen-2.5-7B-TuluSFT | | | | |
| --- | --- | --- | --- | --- | --- | --- | --- | --- | --- | --- | --- | --- |
| Benchmark | SFT | TuluDPO | ORPO | UltraFB | HelpSteer | CodePref | SFT | TuluDPO | ORPO | UltraFB | HelpSteer | CodePref |
| *Knowledge* | | | | | | | | | | | | |
| MMLU (5-shot) | 62.30 | **63.47** | 62.31 | 62.53 | 62.04 | 59.96 | 72.41 | 73.10 | 73.12 | **73.32** | 72.38 | 72.82 |
| MMLU-Pro (5-shot) | 28.08 | **28.98** | 27.90 | 28.41 | 27.43 | 27.53 | 43.32 | 43.48 | 43.73 | **43.94** | 43.07 | 43.38 |
| TruthfulQA (0-shot) | 46.84 | **56.78** | 52.26 | 50.26 | 47.43 | 56.15 | 51.64 | **54.46** | 53.53 | 53.95 | 52.27 | 52.10 |
| GPQA (0-shot) | 28.44 | 29.61 | 29.70 | 28.69 | 29.36 | **29.95** | 30.96 | **31.29** | 30.61 | 31.04 | 30.05 | 30.37 |
| *Reasoning* | | | | | | | | | | | | |
| ARC-C (25-shot) | 54.95 | **57.93** | 56.91 | 56.91 | 54.52 | 45.05 | 59.22 | **60.32** | 60.07 | 60.32 | 59.64 | 59.56 |
| BBH (3-shot) | 38.59 | 40.46 | **41.80** | 39.91 | 38.15 | 36.07 | 47.93 | 48.08 | 47.14 | **48.57** | 47.19 | 47.72 |
| MuSR (0-shot) | 43.25 | 41.93 | **43.78** | 41.93 | 42.06 | 41.53 | 48.15 | 47.16 | 47.88 | 47.75 | **48.41** | 46.30 |
| *Commonsense* | | | | | | | | | | | | |
| HellaSwag (10-shot) | 60.41 | **64.85** | 60.08 | 62.29 | 61.20 | 53.38 | 60.85 | **62.70** | 61.07 | 61.80 | 61.12 | 61.22 |
| WinoGrande (5-shot) | **76.40** | 75.30 | 74.27 | 75.53 | 76.16 | 69.93 | **73.17** | 72.96 | 72.06 | 72.85 | 72.11 | 70.48 |
| *Instruction Following* | | | | | | | | | | | | |
| IF-Eval (0-shot) | 72.45 | **80.35** | 71.65 | 71.59 | 73.93 | 69.90 | 68.06 | **78.04** | 73.02 | 75.74 | 70.16 | 71.05 |
| *Math* | | | | | | | | | | | | |
| GSM8K (5-shot) | 76.19 | 79.48 | **81.96** | 78.47 | 76.88 | 76.35 | 71.27 | 76.84 | **77.33** | 74.93 | 71.87 | 72.24 |
| MATH (4-shot) | 12.08 | **22.66** | 20.39 | 18.81 | 14.88 | 15.11 | 36.21 | **43.13** | 41.92 | 39.19 | 37.72 | 36.34 |
| *Code (pass@1)* | | | | | | | | | | | | |
| HumanEval | 57.93 | 67.24 | 64.63 | 61.59 | 59.15 | **70.68** | 72.66 | 80.49 | 78.66 | 76.05 | 76.27 | **84.54** |
| HumanEval+ | 43.29 | 46.36 | 41.63 | 42.49 | 42.93 | **50.61** | 55.85 | 61.83 | 59.10 | 58.01 | 56.02 | **65.49** |
| *Leaderboards* | | | | | | | | | | | | |
| Open LLM Leaderboard 1 | 62.85 | **66.30** | 64.63 | 64.33 | 63.04 | 60.14 | 64.76 | **66.73** | 66.20 | 66.19 | 64.90 | 64.74 |
| Open LLM Leaderboard 2 | 37.15 | **40.66** | 39.20 | 38.22 | 37.64 | 36.68 | 45.77 | **48.53** | 47.38 | 47.70 | 46.10 | 45.86 |
| *Overall* | 50.09 | **53.96** | 52.09 | 51.39 | 50.44 | 50.16 | 56.55 | **59.56** | 58.52 | 58.39 | 57.02 | 58.11 |

work lacks comprehensive, side-by-side evaluations of sample quality and task coverage, even though such a comparison is critical for designing principled and effective DPO mixtures.

We close these gaps by presenting the first systematic analysis of five widely used open DPO datasets: two general-purpose (TuluDPO, ORPO), two helpfulness-oriented (UltraFeedback, HelpSteer), and one code-focused (Code-Preference-Pairs). See App. B.1 for details on these datasets. Our efforts are inspired by our previous work (Djuhera et al., 2025), which systematically annotated and analyzed datasets for the SFT case.

While several preference datasets exist, we selected these five for their frequent appearance in research papers and open-source dataset hubs (Labonne, 2025). However, our analysis, annotation pipeline, and data curation recipes are generalizable and can, in principle, be applied to any DPO dataset. We provide a thorough comparison with related work in App. B.2. In the following sections, we benchmark the performance of each corresponding DPO dataset and conduct an extensive side-by-side comparison of their compositions on a per-sample basis.

## 3 BENCHMARKING DPO DATASETS

We begin with a side-by-side performance comparison of the five DPO datasets in Table 1. We perform DPO training using *Open-Instruct* (AllenAI, 2023) on two representative models: (i) *Llama-3.1-8B-TuluSFT*, an SFT-tuned Llama-3.1-8B-Base (Grattafiori et al., 2024), and (ii) *Qwen-2.5-7B-TuluSFT*, an SFT-tuned Qwen-2.5-7B-Base (Yang et al., 2025). Here SFT is performed using the popular TuluSFT dataset (Lambert et al., 2025). We benchmark downstream performance on 12 representative tasks from Open LLM Leaderboards V1 and V2 (Fourrier et al., 2023; 2024), as well as on two code generation tasks: HumanEval and HumanEval+ (Chen et al., 2021). We use this setup for all of our experiments. More details on training and evaluation are found in App. F.1. We further extend our evaluations to six additional open models in Sec. 5.3 with comprehensive results in App. F.3.

Table 1 shows that, for both Llama and Qwen, TuluDPO outperforms all other DPO datasets on average and on both leaderboard benchmarks, demonstrating strong curation across a wide range of tasks. In general, ORPO outperforms UltraFeedback on most code and particularly math tasks. However, for Qwen, UltraFeedback surpasses ORPO on instruction following and reasoning. For both models, HelpSteer falls behind UltraFeedback and ORPO. Further, Code-Preference-Pairs shows noticeable gains primarily on code benchmarks, but falls short on math, reasoning, and instruction following tasks compared to TuluDPO, ORPO, and UltraFeedback.

These findings raise several initial research questions: First, given the general-purpose data included in TuluDPO, can the strengths and weaknesses observed in DPO training be attributed to the distribution of specific task categories and the quality of their associated samples? Second, how can specialized preference datasets such as Code-Preference-Pairs be effectively combined with general-purpose datasets to create higher-performing mixtures on individual benchmarks? Third, do low performing preference datasets still contain some *high quality* samples that can be leveraged when curating preference data mixtures? Finally, how can we filter out *low quality* samples from the datasets by assessing preference pairs in an automated manner using LLM-based reward models? We address these questions by presenting a detailed analysis of all of above DPO datasets in the following sections, enabling more informed decisions about dataset composition and mixture strategies.

## 4 COMPARATIVE ANALYSIS OF DPO DATASETS

We conduct a detailed side-by-side analysis of all considered DPO datasets. Each sample is systematically annotated using the Magpie framework, a customizable self-synthesis annotation pipeline, yielding fine-grained labels for task category, difficulty, input quality, response quality, language, and safety. These structured annotations enable evaluation of preference pairs on a per-sample-level, allowing us to judge the preference order and correlate samples with quality and difficulty indicators. This detailed characterization forms a principled foundation for constructing effective DPO mixtures.

### 4.1 ANNOTATING DPO SAMPLES USING MAGPIE

Magpie leverages an LLM as a judge to assign labels for *task category* (12 classes), *query difficulty* (rated from very easy to very hard), *input quality* (rated from very poor to excellent), *safety*, and *language*, using specialized prompt templates (see App. C.1). As a novel component, we assess *preference rewards* by evaluating the *response quality* of chosen and discarded completions using *FsfairX*, a Llama-3-8B-Instruct-based reward model introduced by Dong et al. (2024), which was fine-tuned on a diverse mixture of high-quality preference datasets. This particular reward model has proven to achieve strong alignment with human preference judgments, making it a reliable signal for quantifying how well the model completions satisfy the instruction. To avoid over-reliance on a single reward model, we include a brief comparison with an alternative model in App. C.2, yielding similar principal outcomes and thereby validating our choice. For all other labels, we use Llama-3.3-70B-Instruct as the judge model given its demonstrated reliability, particularly for difficulty and quality annotations, as assessed by Djuhera et al. (2025). In the following analysis, we focus on task category, query difficulty, input quality, and preference reward as the primary signals for dataset comparison, and refer to App. D.5 for additional details on low-impact language and safety distributions.

### 4.2 TASK CATEGORIES, QUERY DIFFICULTY, AND INPUT QUALITY

The distribution of task categories in Fig. 1 shows that information seeking tasks dominate across all DPO datasets, mostly involving prompts that ask for detailed explanations, context-specific summaries, or factual clarifications that are typical for instruction following behavior. This trend aligns with the objectives of instilling helpfulness during preference fine-tuning and is especially prevalent in HelpSteer and UltraFeedback, where information seeking prompts account for 51% and 49% of the data, respectively. TuluDPO and ORPO also exhibit strong coverage in this category, with 38% and 37% of samples. Math is the second most prominent category with ORPO having the highest proportion at 29%, followed by TuluDPO at 17%. Interestingly, despite its smaller relative proportion, TuluDPO achieves stronger performance on math benchmarks (see Table 1), suggesting that absolute sample count and potentially higher quality samples may play a more critical role in math tuning. Coding, creative writing, and reasoning are similarly represented among UltraFeedback and TuluDPO, while more conversational task types such as editing, role playing, brainstorming, and planning are typically underrepresented across all DPO datasets, each comprising only 2–6% of the data. HelpSteer shows a relatively even distribution outside information seeking, though its limited size restricts its overall impact on downstream performance, falling behind others. We exclude Code-Preference-Pairs from this analysis, as it consists solely of coding samples by design. Overall, these findings indicate that DPO datasets place strong emphasis on instruction following through factual information seeking and structured reasoning, most notably in the domains of math and code.

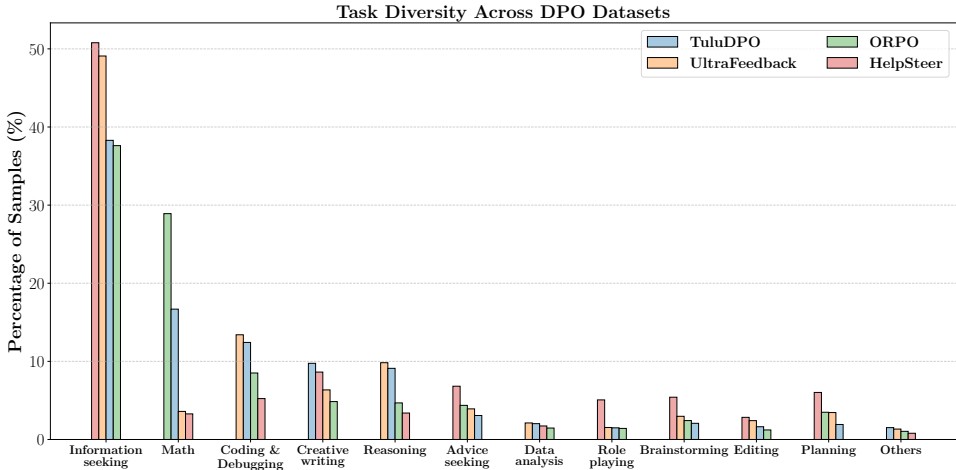

Figure 1: Task diversity across all datasets as annotated by Magpie (excluding Code-Preference-Pairs). Bars indicate the proportion of each dataset dedicated to different task categories. Information seeking dominates across all datasets, followed by math and coding. See App. D.1 for more details.

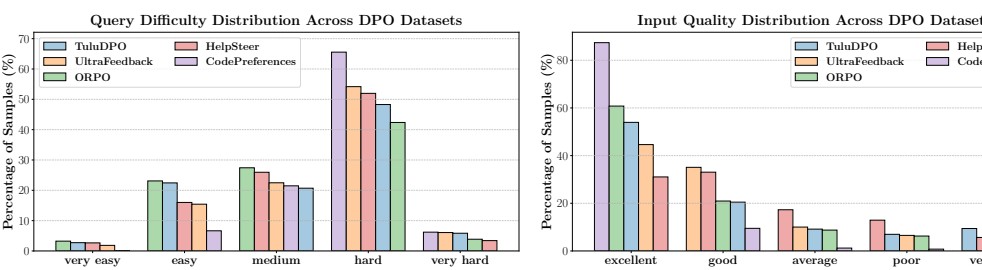

(a) Query difficulty: most instructions are labeled as hard or medium, suggesting that both general-purpose and domain-specific DPO datasets tend to include challenging prompts to encourage stronger alignment.

(b) Input quality: TuluDPO, ORPO, UltraFeedback, and CodePreferences are mostly labeled as good or excellent, indicating well-formulated prompts. HelpSteer contains a non-negligible portion of low-quality inputs.

Figure 2: DPO prompt analysis: (a) Distribution of query difficulty, (b) Distribution of input quality.

We show the corresponding distribution of query difficulty in Fig. 2a, revealing that most prompts are labeled as *"hard"* or *"medium"*, with a smaller fraction categorized as *"easy"*. This suggests that preference tuning across both general-purpose and domain-specific datasets tends to favor more challenging instruction-response pairs, which may help induce stronger alignment and improved downstream performance. Fig. 2b further shows the distribution of input quality for all datasets, showing that TuluDPO, ORPO, UltraFeedback, and Code-Preference-Pairs contain predominantly high-quality prompts, with more than 75% rated as either *"good"* or *"excellent"*. This reflects the filtering processes employed during curation. In contrast, HelpSteer, being more broadly distributed in topic and style, contains a non-negligible portion of 35% of prompts rated as *"average"*, *"poor"*, or *"very poor"*, indicating either lack of context or underspecified user intent. We provide additional insights into how query difficulty and input quality vary by task category in App. D.2 and App. D.3.

Our Magpie annotations in Fig. 1 and Fig. 2 reveal several shared patterns across the examined datasets: (a) preference alignment is most effectively instilled through factual information seeking, math, and code, (b) stronger alignment is elicited by more challenging instructions, and (c) quality matters as most high-performing DPO datasets contain precise prompts to maximize alignment signal.

### 4.3 PREFERENCE REWARD ANALYSIS

As we compute preference rewards with an independent reward model, it also serves as a verification mechanism for the overall preference ordering, which is particularly interesting in the case of human- or GPT-based annotations as found in UltraFeedback and HelpSteer. To this end, we use the preference

reward annotations to first evaluate whether the chosen completion in each preference pair indeed receives a higher reward than the rejected one. Surprisingly, we observe a non-negligible number of cases where the reward model assigns a higher score to the discarded response. Fig. 3 shows, for each DPO dataset, the proportion of samples where the chosen completion is correctly preferred according to the reward model. TuluDPO, ORPO, UltraFeedback, and HelpSteer exhibit similar patterns, with only 70–80% of samples aligning with the chosen preference. This suggests that preference decisions may sometimes be arbitrary or based on near-identical completions (see App. B.1). In addition, this shows that GPT-4-based reward annotations in UltraFeedback, as well as the simplistic use of mean scores in HelpSteer, are often misaligned with the judgments of a specialized preference reward model. Interestingly, Code-Preference-Pairs appears less affected by such inconsistencies. This may be due to more salient signals (e.g., missing features or syntax errors) in code tasks, making it easier for both humans and reward models to distinguish between preference pairs.

To further assess the reward distribution, we compute the difference between preference rewards of chosen and rejected completions for each sample, and present the resulting distribution in Fig. 4. The histogram reveals that for more comprehensive datasets such as TuluDPO, ORPO, and UltraFeedback, the reward differences are more broadly distributed, reaching the positive tails and indicating a clearer separation between good and bad completions. Code-Preference-Pairs similarly extends to the positive tail but also exhibits smaller differences, particularly for examples where the code is functionally identical but the chosen completion includes additional commentary, making the distinction more subtle. In contrast, HelpSteer shows a distribution centered more closely around zero, suggesting that most of its preference pairs consist of similarly rated completions. This supports our earlier observation that HelpSteer may provide weaker alignment signals due to near-identical completions.

To further examine the relationship between input quality and preference rewards, we analyze the average reward assigned to chosen completions across different input quality levels in Fig. 5. The results reveal a clear upward trend for most datasets: higher input quality is associated with higher preference rewards. This suggests that model completions are strongly correlated with the quality of the prompt, providing, to our knowledge, the first empirical evidence that poorly written instructions may lead to subpar preference alignment. Interestingly, Code-Preference-Pairs exhibits a notable outlier for inputs rated as *"very poor"*. Upon further inspection, many of these prompts contain either contradictory requirements or missing information, but completions still produce fully functional responses by falling back on reasonable assumptions or by using simple dummy examples to solve the task.

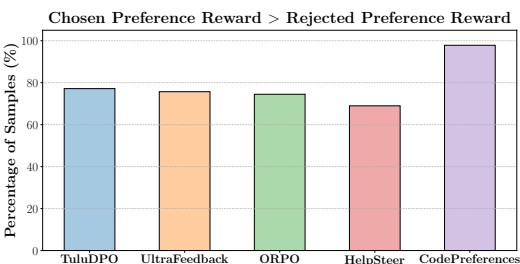

Figure 3: Proportion of samples where chosen completions are correctly preferred.

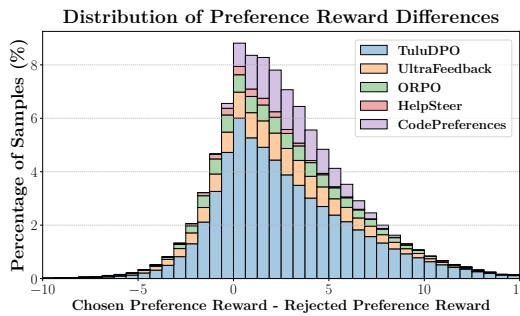

Figure 4: Histogram of reward differences between chosen and rejected completions.

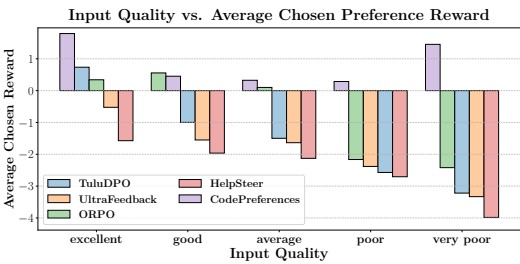

Figure 5: Average reward assigned to chosen completions across different input quality levels.

From this analysis, we conclude that our preference reward annotations uncover several non-trivial insights into dataset quality and curation practices. Most notably, our independent reward model is able to identify potentially misaligned preference pairs. Further, our findings highlight that both input quality as well as consistency between chosen and discarded completions emerge as useful filtering criteria, motivating the construction of potentially more performant DPO mixtures by selectively

retaining samples with high-quality prompts and coherent preference ordering. We refer to App. D.4 for additional analyses and to App. F.5 for more ablations on preference filtering in DPO datasets.

# 5 USING ANNOTATIONS TO DESIGN REWARD-BASED CURATION RECIPES

We leverage our annotations of query difficulty, input quality, and preference rewards to curate a new DPO mixture that draws from all five examined datasets. To this end, we design a set of filters and evaluate their effectiveness through ablation experiments. Our objective is to construct a new mixture that surpasses the current best-performing TuluDPO by selectively retaining only high-quality samples whose preference rewards are aligned with the judgments of our independent reward model.

## 5.1 QUALITY- AND REWARD-BASED CURATION RECIPE

**Initial Recipe.** We curate an initial mixture by selecting from each dataset only those samples where (a) input quality is high (*"excellent"* or *"good"*), (b) difficulty is greater than *"very easy"* (due to low correlation with downstream performance), and (c) the chosen preference reward is higher than the discarded one (ensuring alignment with the reward model). From each filtered dataset, we then retain samples for which the reward of the chosen response is above the 25th percentile for TuluDPO, ORPO, UltraFeedback, and HelpSteer, and apply a stricter threshold at the 80th percentile for Code-Preference-Pairs to avoid over-representing code-related data. To further reduce redundancy, we also perform deduplication, as TuluDPO contains a substantial overlap with UltraFeedback. This recipe yields a mixture of 170k samples (37% smaller than TuluDPO), which we refer to as *UltraMix-170k*.

**Performance Analysis.** Table 2 compares evaluation results for UltraMix-170k (UM-170k) against TuluDPO. For Llama, UltraMix-170k marginally outperforms TuluDPO on overall scores and OpenLLM leaderboard metrics. This improvement is driven primarily by a substantial increase on TruthfulQA from 56.78% to 61.45%, as well as gains on BBH reasoning. However, UltraMix-170k underperforms on key benchmarks, particularly on both code tasks, MATH, and IFEval, indicating that the curation process could be further refined to boost performance in these areas. For Qwen, UltraMix-170k shows a slightly stronger improvement over TuluDPO in both the overall average and individual leaderboard scores. Notable gains are again observed on TruthfulQA, reasoning benchmarks (ARC-C, BBH), and, unlike with Llama, both math tasks. Specifically, UltraMix-170k improves performance from 43.13% to 47.55% on MATH, suggesting that Qwen may be inherently stronger at math reasoning than Llama, supported by its substantially higher MATH score after identical SFT. However, as with Llama, it still underperforms TuluDPO on key code tasks and IFEval.

Given these results, our initial curation strategy may be too simplistic, not capturing enough samples required for strong downstream performance on code and math. In particular, filtering based solely on high quality and preference rewards may have removed valuable instruction following examples, especially from TuluDPO, thereby negatively impacting results on these benchmarks. We explore this hypothesis further through a distributional analysis of the resulting mixture per task category.

**Task Analysis.** Our initial mixture underperforms on benchmarks that heavily rely on instruction following capabilities. This is a critical skill that has been shown to influence performance across a wide range of tasks, as observed by Lambert et al. (2025), Djuhera et al. (2025), and Aakanksha et al. (2025). Since the majority of samples in our mixtures originate from TuluDPO (due to its significantly larger size compared to the other DPO datasets), we examine the distribution of instruction following task categories in Table 3. The analysis reveals that UltraMix-170k underrepresents information seeking and reasoning samples, with relative reductions of 20% and 13%, respectively. In contrast, the proportions of math and coding samples have increased. However, as previously noted, performance on these categories tends to depend on absolute sample count than relative proportion. These findings confirm that our quality- and reward-based curation recipe would benefit from task-aware filtering to preserve a stronger representation of instruction following tasks that increase overall performance.

## 5.2 QUALITY-, REWARD-, AND TASK-BASED CURATION RECIPE

**Adapted Recipe.** We extend our previous reward-based curation strategy using two boosting techniques. First, we increase the absolute sample count, particularly for math and code, to improve overall performance. To this end, we augment UltraMix-170k with additional math and code samples

Table 2: DPO results for Llama-3.1-8B-TuluSFT and Qwen-2.5-7B-TuluSFT trained on our curated DPO mixtures UltraMix-170k (UM-170k), UltraMix-187k (UM-187k), and UltraMix-190k (UM-190k), compared to TuluDPO on Open LLM Leaderboards (averaged) and code benchmarks. The overall average is across all benchmarks. Best scores are in **bold**.

| Benchmark | Llama-3.1-8B-TuluSFT | | | | | Qwen-2.5-7B-TuluSFT | | | | |
|---|---|---|---|---|---|---|---|---|---|---|
| | SFT | TuluDPO | UM-170k | UM-187k | UM-190k | SFT | TuluDPO | UM-170k | UM-187k | UM-190k |
| *Knowledge* | | | | | | | | | | |
| MMLU (5-shot) | 62.30 | 63.47 | 63.27 | 64.19 | **64.61** | 72.41 | 73.10 | 73.53 | 73.92 | **74.01** |
| MMLU-Pro (5-shot) | 28.08 | 28.98 | 28.50 | 30.24 | **30.96** | 43.32 | 43.48 | 43.16 | 44.50 | **44.65** |
| TruthfulQA (0-shot) | 46.84 | 56.78 | 61.45 | **63.85** | 63.32 | 51.64 | 54.46 | 57.60 | **58.24** | 58.00 |
| GPQA (0-shot) | 28.44 | 29.61 | 29.94 | 31.46 | **31.87** | 30.96 | 31.29 | 31.63 | 32.78 | **33.03** |
| *Reasoning* | | | | | | | | | | |
| ARC-C (25-shot) | 54.95 | 57.93 | 57.63 | 58.34 | **58.70** | 59.22 | 60.32 | 62.12 | 62.26 | **62.43** |
| BBH (3-shot) | 38.59 | 40.46 | 44.68 | 44.21 | **44.96** | 47.93 | 48.08 | 50.96 | 51.67 | **51.76** |
| MuSR (0-shot) | 43.25 | 41.93 | 42.43 | 43.42 | **44.02** | 48.15 | 47.16 | 47.87 | 48.58 | **48.82** |
| *Commonsense* | | | | | | | | | | |
| HellaSwag (10-shot) | 60.41 | **64.85** | 63.43 | 64.02 | 64.82 | 60.85 | 62.70 | 63.59 | 63.54 | **63.89** |
| WinoGrande (5-shot) | 76.40 | 75.30 | 74.98 | 76.38 | **77.06** | 73.17 | 72.96 | 73.69 | 74.59 | **74.64** |
| *Instruction Following* | | | | | | | | | | |
| IF-Eval (0-shot) | 72.45 | 80.35 | 79.38 | 80.19 | **81.13** | 68.06 | 78.04 | 77.28 | 78.67 | **79.88** |
| *Math* | | | | | | | | | | |
| GSM8K (5-shot) | 76.19 | 79.48 | 79.98 | 80.93 | **82.48** | 71.27 | 76.84 | 79.19 | 81.26 | **82.70** |
| MATH (4-shot) | 12.08 | 22.66 | 21.22 | 22.03 | **23.56** | 36.21 | 43.13 | 47.55 | 48.79 | **49.55** |
| *Code (pass@1)* | | | | | | | | | | |
| HumanEval | 57.93 | 67.24 | 65.61 | 68.06 | **69.05** | 72.66 | 80.49 | 78.05 | 81.10 | **82.27** |
| HumanEval+ | 43.29 | 46.36 | 45.76 | 47.67 | **48.08** | 55.85 | 61.83 | 60.49 | 62.78 | **63.05** |
| *Leaderboards* | | | | | | | | | | |
| Open LLM Leaderboard 1 | 62.85 | 66.30 | 66.79 | 67.95 | **68.50** | 64.76 | 66.73 | 68.29 | 68.97 | **69.28** |
| Open LLM Leaderboard 2 | 37.15 | 40.66 | 41.02 | 41.92 | **42.75** | 45.77 | 48.53 | 49.74 | 50.83 | **51.28** |
| *Overall* | 50.09 | 53.96 | 54.16 | 55.36 | **56.04** | 56.55 | 59.56 | 60.48 | 61.62 | **62.05** |

Table 3: Distribution of task categories associated with instruction following capabilities. Our initial UltraMix-170k dataset underrepresents information seeking and reasoning tasks relative to TuluDPO.

| Task Category | TuluDPO | UltraMix-170k | UltraMix-187k | UltraMix-190k |
|---|---|---|---|---|
| Math | 5.3% | 24.8% | 22.1% | 19.6% |
| Coding & Debugging | 12.7% | 17.3% | 14.5% | 12.9% |
| Information seeking | 48.6% | 28.5% | 33.4% | 38.7% |
| Reasoning | 19.0% | 5.8% | 7.9% | 9.2% |

from all datasets, again selecting only pairs where the chosen preference reward is higher than the discarded one. This yields 17,000 additional samples, resulting in *UltraMix-187k*.

Next, we specifically boost instruction following. Since the current augmentation already includes most very high-quality samples, we select additional information seeking and reasoning examples with preference rewards above the 70th percentile but with slightly lower *"average"* input quality. This adds another 3,000 samples, yielding *UltraMix-190k*, which remains 30% smaller than TuluDPO.

This adapted recipe captures a broader set of high-quality samples, especially for math and code, while also recovering strategically selected instruction following tasks by moderately relaxing quality constraints. As shown in Table 3, UltraMix-187k and UltraMix-190k contain a 5% and 10% increase in essential information seeking samples, respectively, as well as an increase in reasoning samples, relative to our previous mixture. We present our curation recipe in full detail in App. E.

**Performance Analysis.** Table 2 complements the previous analysis with corresponding evaluation results for UltraMix-187k and UltraMix-190k. Across nearly all benchmarks, the instruction-boosted UltraMix-190k significantly outperforms both 170k and 187k variants, and, most notably, TuluDPO.

Compared to UltraMix-170k, we observe substantial improvements for both new mixtures on IFEval, GSM8K, MATH, and coding, suggesting that the inclusion of more samples has increased overall performance across benchmarks. While UltraMix-187k still underperforms TuluDPO on IFEval and MATH for Llama, the additional 3,000 instruction following samples in UltraMix-190k have led to a compounding performance increase across all benchmarks. This corroborates the observations by Lambert et al. (2025), Djuhera et al. (2025), and Aakanksha et al. (2025) that instruction following samples can lead to noticeable cross-task performance improvements.

Table 4: DPO results for Llama-3.1-8B-TuluSFT and Qwen-2.5-7B-TuluSFT, along with six additional open SFT-tuned models, trained on all datasets, including our curated mixtures UltraMix-170k (UM-170k), UltraMix-187k (UM-187k), and UltraMix-190k (UM-190k). We report overall averages across the 14 benchmarks, with the best scores highlighted in **bold**.

| Model | SFT | Original DPO Datasets | | | | | UltraMix | | |
| | | TuluDPO | ORPO | UltraFB | HelpSteer | CodePref | UM-170k | UM-187k | UM-190k |
|---|---|---|---|---|---|---|---|---|---|
| *TuluSFT Models* | | | | | | | | | |
| Llama-3.1-8B-TuluSFT | 50.09 | 53.96 | 52.09 | 51.39 | 50.44 | 50.16 | 54.16 | 55.36 | **56.04** |
| Qwen-2.5-7B-TuluSFT | 56.55 | 59.56 | 58.52 | 58.39 | 57.02 | 58.11 | 60.48 | 61.62 | **62.05** |
| *Large Open Models* | | | | | | | | | |
| Apertus-8B-SFT | 44.90 | 47.66 | 46.66 | 46.45 | 45.38 | 45.06 | 47.95 | 49.17 | **49.60** |
| OLMo-2-7B-SFT | 45.04 | 48.27 | 47.08 | 47.26 | 45.78 | 46.51 | 49.14 | 49.80 | **50.02** |
| *Medium Open Models* | | | | | | | | | |
| SmolLM-3-3B-SFT | 46.61 | 50.87 | 45.69 | 49.55 | 46.51 | 45.92 | 50.55 | 51.74 | **52.04** |
| Instella-3B-SFT | 43.53 | 45.59 | 43.69 | 44.88 | 43.94 | 43.72 | 45.95 | 46.58 | **46.88** |
| *Small Open Models* | | | | | | | | | |
| SmolLM-2-1.7B-SFT | 34.04 | 35.31 | 34.05 | 35.13 | 33.66 | 32.63 | 34.78 | 36.17 | **36.57** |
| OLMo-2-1B-SFT | 33.29 | 37.63 | 35.68 | 36.56 | 34.99 | 35.30 | 37.78 | 38.55 | **38.74** |

For Llama, coding performance for UltraMix-190k now closely approaches that of Code-Preference-Pairs (see Table 1), with HumanEval and HumanEval+ reaching 69.05% and 48.08%, respectively, thus improving significantly from 67.24% and 46.36% on TuluDPO and from 64.61% and 45.76% on UltraMix-170k. IFEval further increases to 81.13% from 80.35% on TuluDPO and from 79.38% and 80.19% on UltraMix-170k and UltraMix-187k. In addition, GSM8K shows a notable improvement to 82.48% from 79.48% on TuluDPO, and MATH rises to 23.56%, improving over TuluDPO's 22.66%.

For Qwen, we observe similar trends for UltraMix-190k. GSM8K reaches 82.70% compared to 76.84% on TuluDPO, further improving from 79.19% on UltraMix-170k and from 81.26% on UltraMix-187k. MATH improves to 49.55% from 43.13% on TuluDPO and from 47.55% on UltraMix-170k. IFEval also increases to 79.88% from 78.04% on TuluDPO and from 77.28% and 78.67% on UltraMix-170k and UltraMix-187k. In addition, coding improves significantly to 82.77% and 63.05% on HumanEval and HumanEval+, outperforming TuluDPO and our previous mixtures.

Collectively, these results show that UltraMix-190k consistently achieves top-tier performance across a diverse set of benchmarks, offering substantial efficiency gains with 30% fewer samples than TuluDPO. We designate UltraMix-190k as our final **UltraMix** DPO dataset and refer to App. F.6 for a dedicated analysis on training efficiency. To assess the contribution of individual filtering steps, we provide granular ablations in App. F.4, showing that quality-, task, and reward-based filters alone are insufficient and that UltraMix's effectiveness stems from the principled combination of those signals.

## 5.3 GENERALIZATION TO DIFFERENT ARCHITECTURES AND MODEL SCALES

To demonstrate the effectiveness of UltraMix, we evaluate six additional *open-source models* of different architectures and scales: *large models* (Apertus-8B-SFT (Apertus-Team, 2025), OLMo-2-7B-SFT (OLMo et al., 2024)), *medium models* (SmolLM-3-3B-SFT (Bakouch et al., 2025), Instella-3B-SFT (Liu et al., 2025)), and *small models* (SmolLM-2-1.7B-SFT (Allal et al., 2025), OLMo-2-1B-SFT (OLMo et al., 2024)). In contrast to our earlier experiments with TuluSFT-tuned Llama and Qwen variants, these additional models have publicly released their SFT checkpoints, which enables an important axis for evaluation with instruct models that were SFT-tuned using different datasets.

Table 4 presents the overall average scores across all benchmarks for each model. Among the original DPO datasets, TuluDPO performs best, followed by UltraFeedback and ORPO. However, it is significantly outperformed by our UltraMix-190k mixture, which leads to substantial improvements on most benchmarks and surpasses both the 170k and 187k variants. We provide the full evaluations for each model in App. F.3. These additional results corroborate our previous analysis and confirm the effectiveness of our UltraMix dataset across diverse model architectures and scales.

# 6 CONCLUSION

In this work, we systematically analyzed five widely used DPO post-training datasets: TuluDPO, ORPO, UltraFeedback, HelpSteer, and Code-Preference-Pairs. By annotating each preference pair with labels for input quality, difficulty, and preference reward, we identified high-reward, high-quality samples and developed a principled data curation strategy that led to UltraMix, a new DPO mixture that is 30% smaller than TuluDPO while achieving superior performance across benchmarks. The results presented in this study support several key takeaways: (a) judicious incorporation of input quality, difficulty, task category, and preference rewards leads to systematic curation recipes, (b) coherent reward samples play a critical role in driving performance gains in DPO training, and (c) the optimal composition of data mixtures is task-sensitive, requiring careful trade-offs between quality and task diversity. We validate the competitiveness of UltraMix by conducting extensive evaluations across 14 diverse benchmarks and eight LLMs of different sizes. By illustrating how quality-, reward-, and task-aware curation can reduce data requirements while increasing performance, our work provides a strong foundation for future efforts in systematic, reward-aligned DPO dataset design. We discuss limitations and broader impact in App. G.

ACKNOWLEDGMENTS

This work was supported in part by the German Federal Ministry of Research, Technology and Space (BMFTR) within the research hub 6G-life (Grant 16KISK002), by the Bavarian Ministry of Science and the Arts and the Saxon Ministry for Science, Culture, and Tourism through the project Next Generation AI Computing (gAIn), by the Bavarian Ministry of Economic Affairs, Regional Development and Energy through the project 6G Future Lab Bavaria, and in part by IBM Research.

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

# A  POST-TRAINING LARGE LANGUAGE MODELS

While *pre-training* enables models to acquire broad linguistic competence and world knowledge, *post-training* adapts models to follow instructions, specialize in tasks, and behave helpfully.

## A.1  GENERAL POST-TRAINING METHODS

Post-training typically involves instruction tuning through supervised fine-tuning (SFT), followed by preference fine-tuning, for example using reinforcement learning (RL).

**Supervised Fine-Tuning (SFT).**  SFT adapts pre-trained LLMs for either broad or specialized domain knowledge. This is done using instruction-response examples, which may originate from human annotations or synthetic generation. Through next-token prediction, the model learns to generalize across diverse domains and specialized tasks, resulting in an instruction-tuned model that responds helpfully. Although SFT enhances a model's ability to follow instructions and handle a wide range of tasks, it does not necessarily align the model's outputs with pre-defined preferences.

**Preference Fine-Tuning.**  The goal of preference fine-tuning is is to align the SFT model's output distribution with behavior that appeals to human preferences or adheres to task-specific objectives. This is commonly achieved by leveraging a learned reward signal or with explicit preference annotations to steer the model toward generating responses that reflect pre-defined behavior such as increased helpfulness, harmlessness, honesty, and style. Widely used methods for preference-based alignment include Proximal Policy Optimization (PPO) (Schulman et al., 2017) and Direct Preference Optimization (DPO) (Rafailov et al., 2023).

**Reasoning Alignment.**  Recent work has explored RL methods that enhance a model's capacity for structured reasoning. For instance, Group Relative Policy Optimization (GRPO) (Shao et al., 2024) has been proposed to elicit deeper, multi-step thinking by refining reward signals through group-based preference aggregation. In parallel, several reasoning-centric datasets have been introduced (Bercovich et al., 2025; Team, 2025; Bespoke-Labs, 2025; Madhusudhan et al., 2025; Min et al., 2024), focusing on task formats that explicitly promote step-by-step problem solving and reflective thought processes.

## A.2  FOCUS ON DPO

The primary goal of this paper is to analyze the quality and composition of preference optimization datasets while keeping the overall training procedure fixed, enabling a fair comparison. In particular, we focus on DPO, which has become the most widely adopted approach for preference-based alignment due to its simplicity and efficiency (Ivison et al., 2024). Unlike PPO that requires both reward modeling and costly policy rollouts, DPO operates directly on preference pairs, making it well suited for dataset-centric comparisons given the large availability of open-source DPO corpora.

Similar efforts have been pursued by Djuhera et al. (2025) for post-training with SFT, demonstrating that task-aware dataset design can improve performance while reducing overall dataset size. However, the diversity of existing post-training methods, including SFT, PPO, and others, implies that dataset requirements are highly method-dependent, complicating direct comparisons across approaches. For instance, unlike SFT, DPO is based on the Bradley-Terry formulation, which relies on pairwise preference samples for optimization. As a result, curation strategies developed for SFT may not be directly applicable to DPO datasets. To address this gap, we focus specifically on post-training with DPO, analyzing the role of preference rewards and assessing the reliability of pre-annotated scores, thus going beyond binary preference labels to enable more insightful dataset construction.

Our evaluation demonstrates that reward-based dataset design is a critical factor in DPO training: by systematically annotating, analyzing, and curating open-source DPO corpora based on coheren preference rewards and quality, we construct UltraMix, a leaner mixture that outperforms larger individual datasets across diverse benchmarks.

## A.3 THEORETICAL PERSPECTIVE ON DATA QUALITY

DPO can be viewed as optimizing a KL–regularized reward objective where the only supervision comes from pairwise preferences Rafailov et al. (2023). Formally, given a reference policy $\pi_{\text{ref}}$ and a dataset of tuples $(x, y_c, y_r)$ with preferred completion $y_c$ and rejected completion $y_r$, the DPO loss for a policy $\pi_\theta$ can be written as

$$\mathcal{L}_{\text{DPO}}(\theta) \;=\; \mathbb{E}_{(x,y_c,y_r)\sim\mathcal{D}}\big[-\log\sigma\big(\beta\,m_\theta(x)\big)\big], \; m_\theta(x) \;=\; \log\frac{\pi_\theta(y_c\mid x)}{\pi_{\text{ref}}(y_c\mid x)} - \log\frac{\pi_\theta(y_r\mid x)}{\pi_{\text{ref}}(y_r\mid x)}, \tag{1}$$

where $\sigma$ is the logistic function and $\beta > 0$ controls the KL strength. The quantity $m_\theta(x)$ is the log-likelihood margin between chosen and rejected responses under the implicit reward model

$$r_\theta(x,y) \;=\; \beta\left(\log\pi_\theta(y\mid x) - \log\pi_{\text{ref}}(y\mid x)\right), \tag{2}$$

which DPO fits directly from preferences. Under the Bradley–Terry model, this implicit reward is provably equivalent (up to an additive baseline) to the reward learned in RLHF, and the optimal policy of DPO matches the KL–constrained RL optimum.

From this perspective, the data enter the optimization only through the distribution of margins $m_\theta(x)$ induced by the preference pairs. If preferences are consistent and the chosen response is substantially better than the rejected one, the margin is positive and large in magnitude, yielding a high signal-to-noise ratio (SNR) for the gradient. Conversely, incoherent or near-tied pairs induce small or even negative margins, which (i) reduce the effective SNR and (ii) generate conflicting gradients that pull $\pi_\theta$ in incompatible directions. Our empirical findings that 20–30% of pairs in popular datasets violate reward coherence, and that prompt quality and reward margins are strongly correlated, provide direct evidence for this mechanism.

A complementary view is given by the $\Psi$-preference optimization ($\Psi$-PO) framework of Azar et al. (2023), which treats learning from preferences as an offline contextual bandit problem. They define a general objective

$$J_\Psi(\pi) \;=\; \mathbb{E}_{x\sim\rho}\mathbb{E}_{y\sim\pi(\cdot\mid x),\,y'\sim\mu(\cdot\mid x)}\big[\Psi\big(p^*(y\succ y'\mid x)\big)\big] \;-\; \tau\,D_{\text{KL}}\big(\pi\,\|\,\pi_{\text{ref}}\big), \tag{3}$$

where $p^*(y\succ y'\mid x)$ is the human preference probability, $\mu$ is a behavior policy, $\Psi$ is any non-decreasing function, and $\tau > 0$ controls regularization. Under the Bradley–Terry model and the specific choice $\Psi(q) = \log\frac{q}{1-q}$, they show that $\Psi$-PO recovers both RLHF and DPO as special cases.

In this formulation, dataset quality is encoded directly in the preference probabilities. If prompts are underspecified, many pairs have $p^*(y_c\succ y_r\mid x)\approx 0.5$, so $\Psi\big(p^*(\cdot)\big)$, degrading the effective gradient. If preference labels are inconsistent (e.g., $p^*(y_c\succ y_r\mid x)<0.5$), updates push the policy toward worse responses. If entire task regions are underrepresented, the expectation over $x\sim\rho$ underweights those regimes, regardless of how clean individual pairs are.

Our Magpie-based annotations and curation recipe improve the $\Psi$-PO objective in three ways. First, higher prompt quality sharpens $p^*(y_c\succ y_r\mid x)$ by reducing ambiguity, increasing the expected margin $m_\theta(x)$ and lowering its variance—consistent with our observation that better inputs yield larger reward gaps. Second, reward coherence removes pairs where the rejected completion scores higher ($r(x,y_r)\geq r(x,y_c)$), eliminating preference inconsistencies that can cause overfitting and instability in margin-based objectives like DPO (Azar et al., 2023). This reduces systematic label noise and stabilizes the margin distribution. Third, task-aware curation reshapes the context distribution $\rho$ by rebalancing task categories. Since instruction-following drives cross-task gains while math and code depend more on absolute scale, this effectively reweights training toward contexts with abundant, high-quality, high-margin preferences that best support generalization.

In summary, existing theory already provides a principled link between data quality and DPO-style objectives: (i) DPO implicitly fits a reward model whose training signal is governed by the log-likelihood margin $m_\theta(x)$, and (ii) $\Psi$-PO shows that the shape of the preference probabilities $p^*$, as influenced by prompts, labels, and task mix, controls the effective optimization landscape. Our empirical work complements this by (a) measuring prompt quality and reward coherence at scale, (b) showing that misaligned and low-margin pairs are common and harmful, and (c) demonstrating that principled filtering on these signals yields consistent improvements across models and benchmarks.

# B    PRIOR AND RELATED WORK

## B.1    DPO DATASETS

In our analysis, we focus on the following five widely used open DPO datasets:

**TuluDPO.** Lambert et al. (2025) originally curated TuluDPO for post-training Llama-3.1 models (Grattafiori et al., 2024). The dataset consists of 272,898 preference pairs drawn from several distinct sources, consolidating on- and off-policy preference data from over seven constituent datasets, including UltraFeedback (Cui et al., 2024) and WildChat (Zhao et al., 2024). The corresponding responses were generated using a diverse set of open and frontier LLMs. While TuluDPO represents one of the most comprehensive, general-purpose DPO corpora available, its construction inherits samples from several upstream mixtures without per-sample metadata, making fine-grained analysis and targeted reuse difficult without additional annotations.

**ORPO.** Labonne (2024) curated ORPO as a mixture of open-source preference pairs designed for training with Optimized Rejection-based Preference Optimization (ORPO) (Hong et al., 2024) and DPO. It aggregates 44,218 high-scoring preference pairs from several high-quality sources, particularly from Argilla (Argilla, 2025), with chosen responses filtered to meet minimum reward thresholds. ORPO offers a broad distribution of general, factual, and safety-aligned instructions for training general-purpose preference models.

**UltraFeedback.** Cui et al. (2024) released curated preference pairs for over 60,000 prompts, spanning instruction following, truthfulness, honesty, and helpfulness. Instructions were sourced from high-quality public datasets such as FLAN (Longpre et al., 2023), Evol-Instruct (Luo et al., 2024), and TruthfulQA (Lin et al., 2022), and their completions were sampled from 17 diverse LLMs, including GPT-4 (Hurst et al., 2024). While UltraFeedback does provide scalar preference scores (assessed via GPT-4), they remain inconsistent, for example, assigning the same reward score for both chosen and discarded responses. This can occur when multiple responses demonstrate similar performance and the preference pair is thus selected arbitrarily.

**HelpSteer.** Wang et al. (2024a) introduced HelpSteer2 (referred to as just HelpSteer) as a high-quality instruction following dataset with human annotations for helpfulness, correctness, coherence, complexity, and verbosity. The small-scale dataset comprises 10,681 prompts with responses collected primarily from ShareGPT. Each response is scored by over 1,000 human annotators on a fine-grained Likert-scale, thus offering explicit, multi-dimensional preference signals rather than binary choices. These annotations can be used to construct preference pairs, for example, based on mean scores, as done in our work.

**Code-Preference-Pairs.** Vezora (2024) introduced this synthetic code dataset to train models for bug identification and correction. Each sample contains two code variants: an accepted version with correct functionality and inline explanations, and a discarded version where bugs are surgically introduced without explicit indicators. The dataset spans over 55,000 preference pairs across multiple languages and coding problems, with samples derived from several sources, including Evol-Instruct (Luo et al., 2024).

We specifically selected these datasets due to their prominence and frequent appearance in open-source community blogs and dataset hubs (Labonne, 2025). While we acknowledge that new, potentially better-performing datasets are continuously being released, our analysis, annotation pipeline, and data curation recipes are generalizable and can, in principle, be applied to any DPO dataset.

## B.2    COMPARISON WITH RELATED WORK

Quantifying data quality is fundamentally non-trivial and prior works differ from ours in how they define what constitutes a "high-quality" preference dataset. We highlight these distinctions and provide new experimental evidence showing that UltraMix's non-trivial combination of complementary signals, rather than quality alone, is essential for achieving the best performance.

**Unpacking DPO/PPO.** (Ivison et al., 2024) compares datasets only monolithically and draws two conclusions: (a) synthetic datasets outperform human-annotated ones, and (b) datasets created using multi-aspect annotations outperform those with aggregated labels. However, their analysis

Table 5: Comparison of RIP-style filtering versus UltraMix variants for Llama-3.1-8B-TuluSFT and Qwen-2.5-7B-TuluSFT. Benchmarks are grouped by category. Best scores per row are in **bold**.

| Benchmark | Llama-3.1-8B-TuluSFT | | | Qwen-2.5-7B-TuluSFT | | |
| --- | --- | --- | --- | --- | --- | --- |
| | RIP-only | RIP+UM-QF | UltraMix | RIP-only | RIP+UM-QF | UltraMix |
| *Knowledge* | | | | | | |
| MMLU (5-shot) | 62.93 | 63.27 | **64.61** | 73.10 | 73.53 | **74.01** |
| TruthfulQA (0-shot) | 61.69 | 61.45 | **63.32** | 54.47 | 57.60 | **58.00** |
| ARC-C (25-shot) | 57.83 | 57.63 | **58.70** | 60.33 | 62.12 | **62.43** |
| GSM8K (5-shot) | 77.27 | 79.98 | **82.48** | 76.85 | 79.19 | **82.70** |
| *Commonsense* | | | | | | |
| HellaSwag (10-shot) | 63.35 | 63.43 | **64.82** | 62.67 | 63.59 | **63.89** |
| WinoGrande (5-shot) | 75.17 | 74.98 | **77.06** | 72.98 | 73.69 | **74.64** |
| *Reasoning* | | | | | | |
| MMLU-Pro (5-shot) | 28.99 | 28.50 | **30.96** | 43.48 | 43.16 | **44.65** |
| BBH (3-shot) | 43.93 | 44.68 | **44.96** | 48.08 | 50.96 | **51.76** |
| GPQA (0-shot) | 30.12 | 29.94 | **31.87** | 31.28 | 31.63 | **33.03** |
| MuSR (0-shot) | 42.42 | 42.43 | **44.02** | 47.19 | 47.87 | **48.82** |
| *Instruction Following* | | | | | | |
| IF-Eval (0-shot) | 79.31 | 79.38 | **81.13** | 78.03 | 77.28 | **79.88** |
| *Math* | | | | | | |
| MATH (4-shot) | 21.53 | 21.22 | **23.56** | 43.13 | 47.55 | **49.55** |
| *Code (pass@1)* | | | | | | |
| HumanEval | 65.71 | 65.61 | **69.05** | 80.49 | 78.05 | **82.27** |
| HumanEval+ | 45.77 | 45.76 | **48.08** | 61.83 | 60.49 | **63.05** |
| *Leaderboards* | | | | | | |
| Open LLM Leaderboard 1 | 66.37 | 66.79 | **68.50** | 66.73 | 68.29 | **69.28** |
| Open LLM Leaderboard 2 | 41.05 | 41.02 | **42.75** | 48.53 | 49.74 | **51.28** |
| *Overall* | 54.00 | 54.16 | **56.04** | 59.56 | 60.48 | **62.05** |

does not examine per-sample quality, nor does it consider post-hoc curation. Our work goes beyond such monolithic comparisons by conducting a per-sample diagnosis, i.e., we look inside datasets and identify which concrete samples are subpar. We then leverage these sample-level annotations to propose a principled curation recipe. To this end, we provide the first large-scale quantification of prompt quality and reward coherence across widely used DPO datasets. Our results eventually show that even "high-quality" datasets (e.g., TuluDPO, UltraFeedback) contain 20–30% of samples with contradicting preference order.

**AIR.** (He et al., 2025) focuses on creating a new dataset from scratch (AIR), whereas our work targets post-hoc curation, which is a crucial distinction. In particular, AIR extends the UltraFeedback pipeline to construct new datasets based on moderate reward margins and high absolute scores. In contrast, our workflow, which combines detailed annotations with a principled curation recipe, is designed specifically for post-hoc optimization. Consequently, AIR cannot be applied retroactively to existing datasets without regenerating and reannotating all samples, whereas UltraMix can. AIR further measures prompt quality by (1) generating multiple responses per prompt using diverse LLMs, (2) assigning LLM-based numerical scores to each response, and (3) defining prompt quality via the variance of these scores. This procedure is feasible only when constructing datasets from scratch and incurs substantial computational cost due to the need for many LLMs. By contrast, our post-hoc curation framework quantifies prompt quality through systematic annotations (e.g., clarity, reward coherence) without multi-LLM generation, making UltraMix both scalable and efficient.

**RIP.** (Yu et al., 2025) investigates ways to filter preference data by maximizing reward differences. However, our approach instead evaluates reward coherence, that is, whether the chosen answer is actually better than the rejected one. Specifically, we show that relying on reward signals alone is insufficient. Our recipe thus balances prompt-level, reward-level, and task-level filtering. To further strengthen this point, we provide a direct comparison to RIP-style filtering in Table 5. We report results for RIP-only (filtering by 75th-percentile reward gap), RIP+UltraMix-Quality-Filter, and the full UltraMix recipe. The results show that RIP-only underperforms even when UltraMix's quality filter is applied on top, which suggests that many pairs have similar reward values. Thus, discarding them solely based on reward gaps harms performance. Consequently, preference order is the more informative signal and our combined recipe yields a superior data filter.

To further eliminate potential data bias, we provide additional results on the newer HelpSteer3 dataset (Wang et al., 2025). To this end, we annotate the single-turn HelpSteer3 variant[7] using our DPO-adapted Magpie pipeline and conduct a comparison with RIP-style filtering on both Llama-3.1-8B and Qwen-2.5-7B. Specifically, we compare three configurations: HelpSteer3 (Baseline), HelpSteer3 (RIP), where we keep only samples with high reward gaps > 60th percentile, and HelpSteer3 (Pref), where we keep only samples where chosen reward > rejected reward.

We report the averaged scores across all leaderboard benchmarks (LB1 and LB2) in Table 6. The results show that preference coherence is the superior filtering signal, yielding better results compared to RIP-style filtering, which actually hurts performance (-1.08% on Llama and -0.67% on Qwen). These findings thus confirm for a separate dataset that high reward gaps do not guarantee high-quality training signals if the underlying preference direction is noisy or if the data distribution is skewed.

Table 6: DPO performance comparison of Llama and Qwen models on HelpSteer3 variants. Training follows the hyperparameter setup of Wang et al. (2025). We report LB1, LB2, average score (Avg.), and the relative performance change ($\Delta$). Best scores within each model block are shown in **bold**.

| | **Llama-3.1-8B-TuluSFT** | | | | **Qwen-2.5-7B-TuluSFT** | | | |
|---|---|---|---|---|---|---|---|---|
| **Dataset** | LB1 | LB2 | Avg | $\Delta$ | LB1 | LB2 | Avg | $\Delta$ |
| *HelpSteer3 (Base)* | 63.46 | 39.21 | 51.53 | – | 65.74 | 47.44 | 58.46 | – |
| *HelpSteer3 (RIP)* | 63.22 | 37.70 | 50.45 | -1.08 | 65.42 | 46.59 | 57.79 | -0.67 |
| *HelpSteer3 (Pref)* | **63.98** | **39.61** | **52.25** | **+0.72** | **66.02** | **47.73** | **58.83** | **+0.37** |

---

[7]Annotated HelpSteer3 dataset: huggingface.co/datasets/aladinDJ/helpsteer3-single-turn-DPO-annotated

## C    MAGPIE ANNOTATIONS

This section provides a brief overview of the Magpie annotation framework.

### C.1    GENERAL OVERVIEW

Magpie (Xu et al., 2025) is a *self-synthesis* framework that derives alignment-oriented annotations from open-weight, instruction-tuned language models without requiring human labels or handcrafted seed prompts. Although the framework is also capable of generating new instruction-response pairs, in this work we rely exclusively on its *annotation* capabilities.

Magpie employs specialized judge models to automatically label individual samples across multiple dimensions (e.g., prompt quality, task type, and safety), making it possible to obtain large-scale annotations that would be prohibitively expensive to collect through manual labeling. This additional metadata enables more principled dataset filtering, stratification, and targeted analysis.

Magpie supports the following annotation labels:

- **Input Quality** (*very poor – excellent*): assesses prompt clarity, specificity, and structure, accompanied by a short textual justification (*"quality explanation"*).
- **Task Category**: maps each prompt to one of 12 categories, such as *Coding & Debugging, Reasoning, Information Seeking, Brainstorming, Creative Writing, Advice Seeking, Math, Planning, Editing, Role Playing, Data Analysis*, or *Other*.
- **Query Difficulty** (*very easy – very hard*): estimates the cognitive demand of the prompt. Each sample is further tagged with *intent* (user goal) and *knowledge* (required competence).
- **Response Quality**: provides a scalar judgment of the response, scored by a reward model.
- **Safety**: classified using a dedicated safety model.
- **Language**: detects the language of the prompt.

A key feature of Magpie is its modularity: any instruction-tuned LLM can, in principle, serve as the judge model. In its default configuration, Magpie relies on Llama-3-8B-Instruct (Grattafiori et al., 2024) for general annotation and Llama-Guard 2 (MetaAI, 2024) for safety assessment. In our pipeline, we use Llama-3.3-70B-Instruct (Grattafiori et al., 2024) as the judge model given its demonstrated reliability, particularly for quality and difficulty annotations, as assessed by Djuhera et al. (2025). In addition, we also incorporate the error-tolerant JSON parsing extensions to Magpie from Djuhera et al. (2025), which help handle formatting inconsistencies and free-form outputs during the annotation process.

### C.2    PREFERENCE REWARD ASSESSMENT

We generate preference rewards by adapting Magpie to evaluate the response quality of each completion. To this end, we rely on an independent, specialized reward model that has been fine-tuned on multiple preference datasets. Specifically, we use *FsfairX*, a Llama-3-8B-Instruct-based reward model introduced by Dong et al. (2024), which was trained using the Bradley-Terry formulation on a diverse set of high-quality preference datasets, including HH-RLHF (Anthropic, 2024), SHP (Ethayarajh et al., 2025), and UltraFeedback Cui et al. (2024). This reward model has demonstrated strong alignment with human preference judgments, making it a reliable signal for quantifying how well a model completion satisfies the corresponding instruction.

The model assigns a scalar reward score (higher is better) to both the chosen and rejected completions in each preference pair. We use these scores in our analysis to verify the correctness of the original preference ordering. Implementation details can be found in our publicly released Magpie codebase. We make all preference reward scores available as part of our annotated datasets.

While we acknowledge that other reward models can be used as alternatives, such as UltraRM-13B (Cui et al., 2024), Snorkel-Mistral-PairRM-DPO (Tran et al., 2023), ArmoRM (Wang et al., 2024b), Skywork-Reward (Liu et al., 2024), or more generally, pairwise preference models trained on high-quality preference datasets, Dong et al. (2024) demonstrated through various experiments that their Llama-based FsfairX model performs comparably or more reliably than these alternatives and aligns

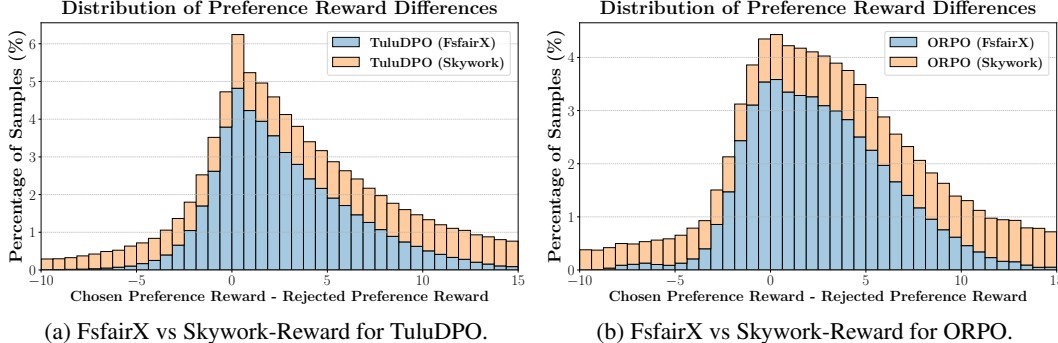

(a) FsfairX vs Skywork-Reward for TuluDPO.  (b) FsfairX vs Skywork-Reward for ORPO.

Figure 6: Histogram of reward differences between chosen and rejected completions for TuluDPO and ORPO datasets when using different reward models.

closely with human judgments, thereby justifying our choice. Nevertheless, we provide a side-by-side comparison to the Skywork reward model for TuluDPO and ORPO datasets in Fig. 6a and Fig. 6b, respectively, showing overall similar distributions for the difference in preference rewards.

In particular, the similar shape of the distributions indicates that preference reward assessment follows the same general trend and does not fundamentally alter the evaluation outcome. At the same time, we observe that the FsfairX model produces fewer negative outliers and more stable margins, making it particularly well-suited for providing reliable training signals without over-amplifying differences. In contrast, the Skywork reward model exhibits both stronger positive separations and more negative outliers, suggesting higher variance. For our purposes, FsfairX provides the consistency and robustness required for a balanced assessment of preference rewards, with its margins further validated against human feedback in several experiments (Dong et al., 2024).

Nevertheless, a comprehensive comparison across the growing landscape of reward models would be valuable but lies beyond the scope of this work. We therefore focus on an established, widely adopted model and leave broader evaluations to future research.

# D EXTENDED DATASET ANALYSIS

We present extended analyses of our annotated DPO post-training datasets, covering dataset composition, task distributions, difficulty and quality metrics, as well as preference reward evaluations.

## D.1 DATASET COMPOSITIONS PER TASK CATEGORY

Tables 7 and 8 compare the composition of each DPO dataset by task category, as labeled by Magpie. This provides a *sample-level* view of how different task types are distributed across all examined open-source datasets, as well as our curated DPO mixtures.

The breakdown in Table 7 complements the analysis presented in Fig. 1, showing that general-purpose datasets like TuluDPO and ORPO are more strategically distributed with high proportions in information seeking, math, and coding.

Table 8 compares the distributions of our curated UltraMix variants, showing that UltraMix-187k and UltraMix-190k increase the proportion of information seeking and reasoning samples. This leads to compounding improvements across benchmarks, driven by both a broader task mix associated with instruction following, and an increase in absolute sample count.

Table 7: Dataset composition per task category for all examined open-source DPO datasets.

| Task Category | TuluDPO | UltraFeedback | ORPO | HelpSteer | CodePref |
|---|---|---|---|---|---|
| Information seeking | 38.3% | 49.1% | 37.6% | 50.8% | 0.0% |
| Reasoning | 9.1% | 9.8% | 4.7% | 3.4% | 0.0% |
| Coding & Debugging | 12.4% | 13.4% | 8.5% | 5.2% | 100.0% |
| Editing | 1.6% | 2.4% | 1.2% | 2.8% | 0.0% |
| Math | 16.7% | 3.6% | 28.9% | 3.3% | 0.0% |
| Advice seeking | 3.1% | 3.9% | 4.4% | 6.8% | 0.0% |
| Planning | 1.9% | 3.4% | 3.5% | 6.0% | 0.0% |
| Creative writing | 9.8% | 6.3% | 4.9% | 8.6% | 0.0% |
| Brainstorming | 2.1% | 3.0% | 2.4% | 5.4% | 0.0% |
| Data analysis | 2.0% | 2.1% | 1.5% | 1.7% | 0.0% |
| Role playing | 1.5% | 1.5% | 1.4% | 5.1% | 0.0% |
| Others | 1.5% | 1.3% | 1.0% | 0.8% | 0.0% |

Table 8: Dataset composition per task category for our curated UltraMix variants.

| Task Category | UltraMix-170k | UltraMix-187k | UltraMix-190k (**UltraMix**) |
|---|---|---|---|
| Information seeking | 31.4% | 31.8% | 32.7% |
| Reasoning | 5.6% | 7.1% | 7.4% |
| Coding & Debugging | 21.5% | 21.0% | 20.7% |
| Editing | 1.6% | 1.5% | 1.5% |
| Math | 18.6% | 19.3% | 19.0% |
| Advice seeking | 2.9% | 2.6% | 2.5% |
| Planning | 3.1% | 2.4% | 2.3% |
| Creative writing | 9.7% | 8.9% | 8.8% |
| Brainstorming | 1.7% | 1.5% | 1.5% |
| Data analysis | 2.7% | 2.4% | 2.4% |
| Role playing | 0.9% | 1.1% | 1.1% |
| Others | 0.2% | 0.2% | 0.1% |

## D.2  QUERY DIFFICULTY PER TASK CATEGORY

Fig. 7 to Fig. 11 show the distribution of query difficulty across task categories for each DPO dataset. This analysis complements Fig. 2a, confirming that most prompts are labeled as *"hard"* or *"medium"*, with only a smaller fraction categorized as *"easy"*. This suggests that preference tuning, across both general-purpose and domain-specific datasets, emphasizes more challenging instruction-response pairs, which may contribute to stronger alignment and improved downstream performance.

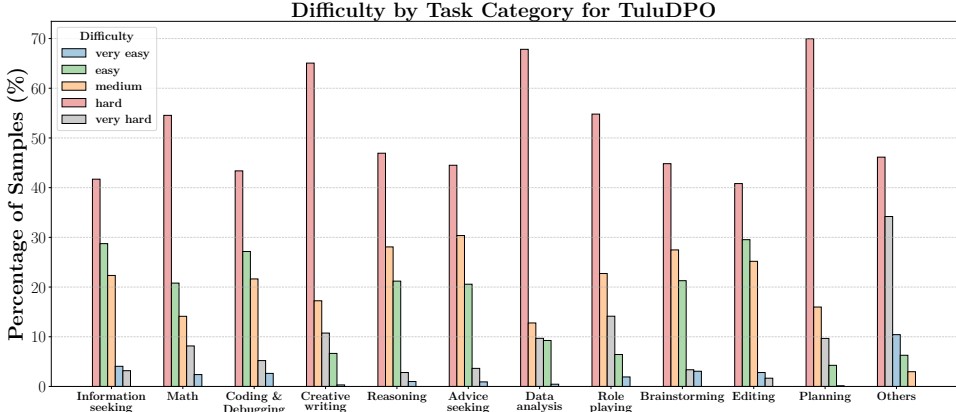

Figure 7: Distribution of query difficulty per task category for TuluDPO.

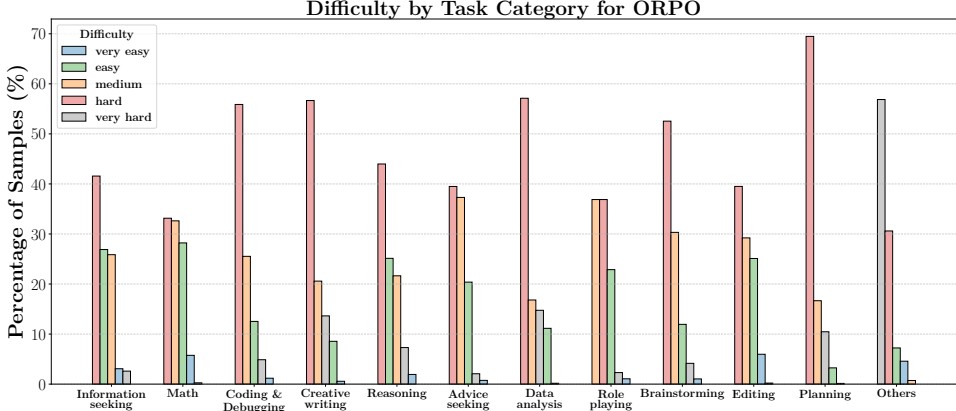

Figure 8: Distribution of query difficulty per task category for ORPO.

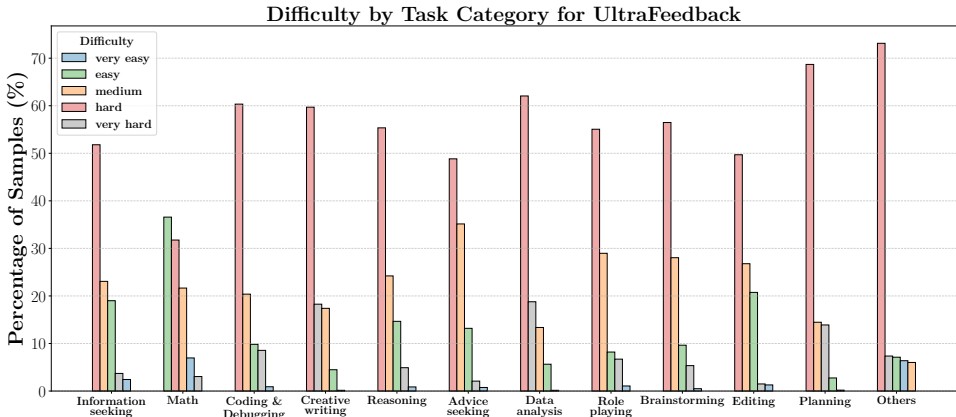

Figure 9: Distribution of query difficulty per task category for UltraFeedback.

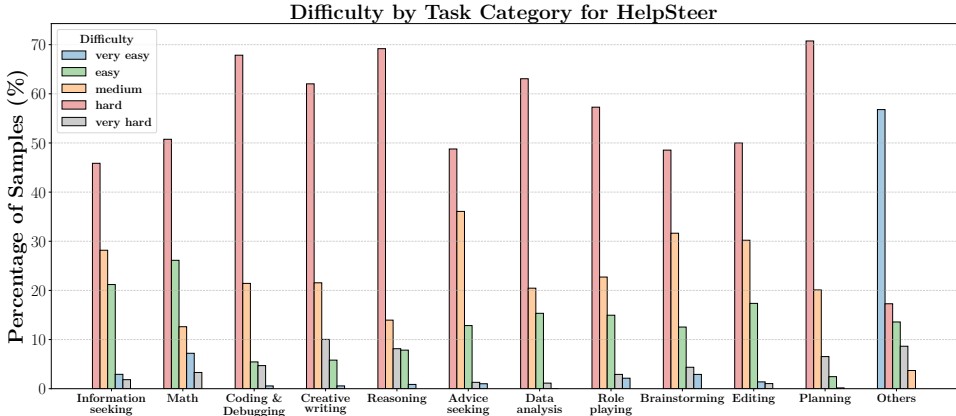

Figure 10: Distribution of query difficulty per task category for HelpSteer.

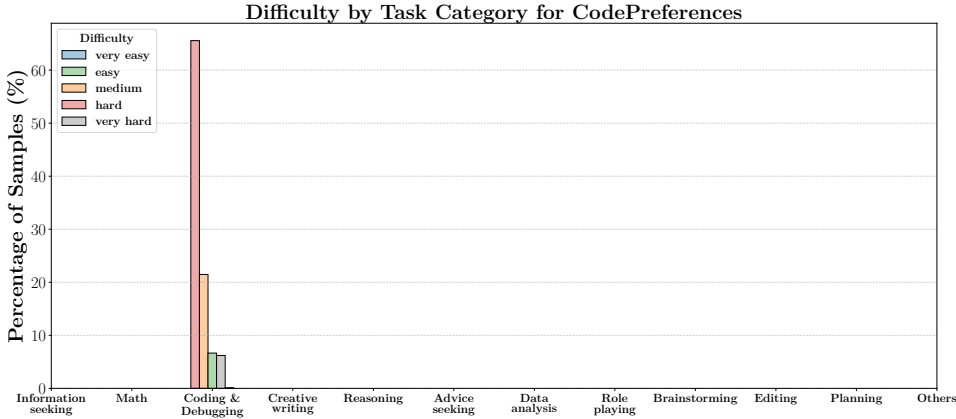

Figure 11: Distribution of query difficulty per task category for Code-Preference-Pairs.

### D.3 INPUT QUALITY PER TASK CATEGORY

Fig. 12 to Fig. 16 show the distribution of input quality across task categories for each DPO dataset. This analysis complements Fig. 2b, confirming that TuluDPO, ORPO, UltraFeedback, and Code-Preference-Pairs contain predominantly high-quality prompts, with more than 75% rated as either *"good"* or *"excellent"*. In contrast, HelpSteer exhibits a more even distribution, with a non-negligible portion of prompts rated as *"average"*, *"poor"*, or *"very poor"*. Many of these samples show a lack of context or underspecified user intent, suggesting that a quality filter could be applied to remove low-quality instances.

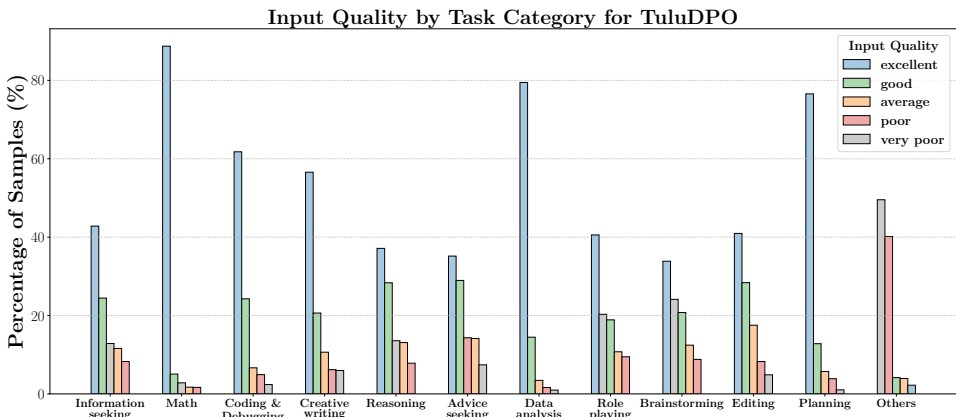

Figure 12: Distribution of input quality per task category for TuluDPO.

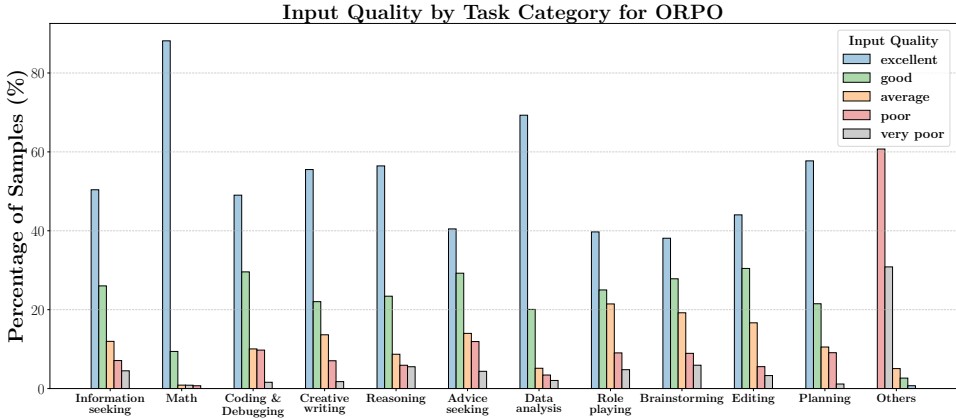

Figure 13: Distribution of input quality per task category for ORPO.

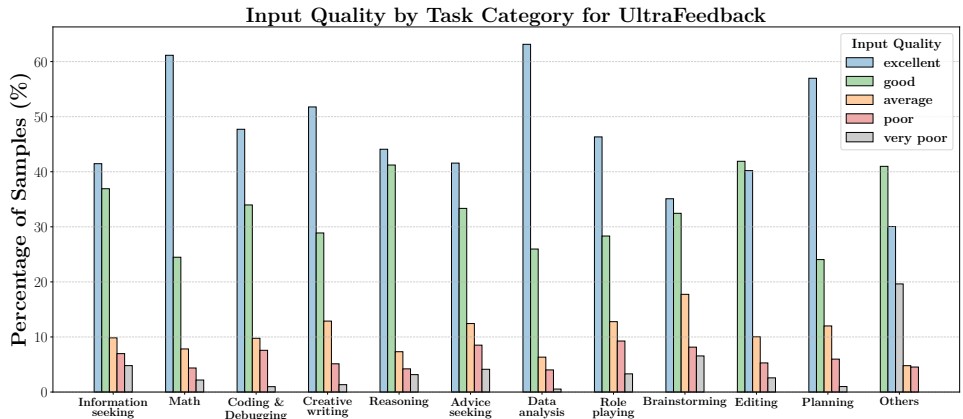

Figure 14: Distribution of input quality per task category for UltraFeedback.

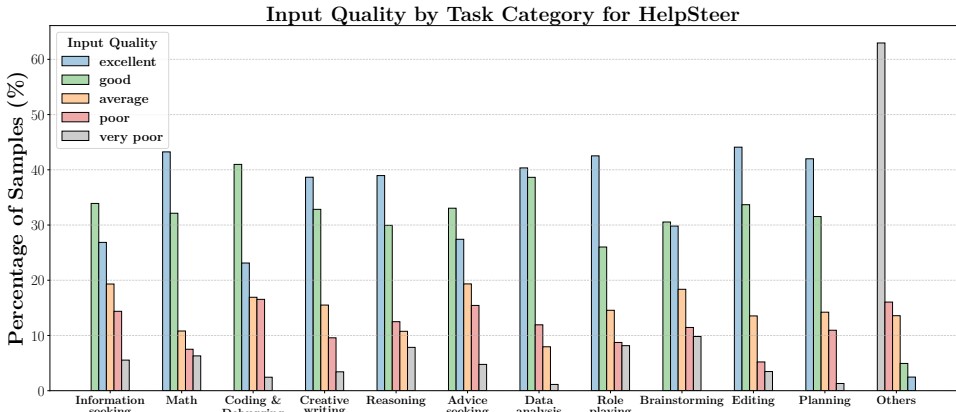

Figure 15: Distribution of input quality per task category for HelpSteer.

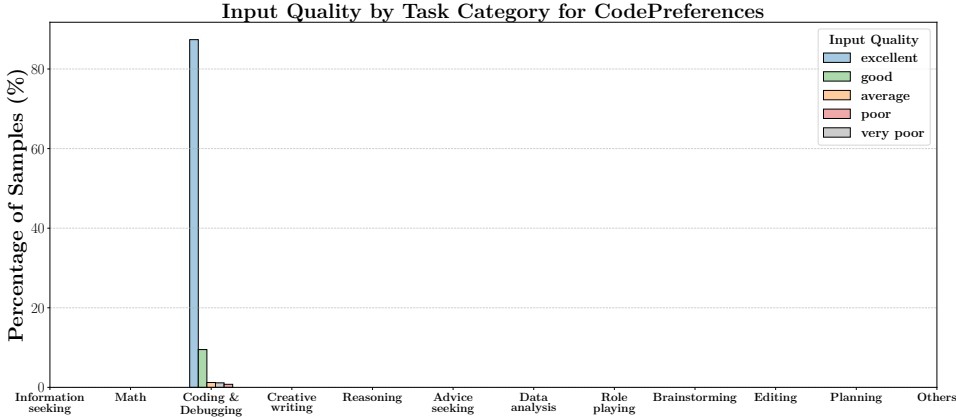

Figure 16: Distribution of input quality per task category for Code-Preference-Pairs.

### D.4 PREFERENCE REWARD ANALYSIS

We present additional results and insights on the distribution of preference rewards across task categories, as well as their relationship to query difficulty and input quality.

#### D.4.1 PREFERENCE REWARDS PER TASK CATEGORY

Fig. 17 shows the distribution of preference rewards per DPO dataset across task categories for all samples in which the chosen completion is correctly preferred over the rejected one (according to our reward model). This analysis complements Fig. 3 and confirms that most categories contain a non-negligible portion of misaligned rewards in preference pairs. This suggests that pre-annotated scores and binary preference orders do not always align with the judgment of an independent, reward-aligned model. These findings further support the use of reward-based filtering to identify high-reward samples that may enhance alignment during DPO while simultaneously reducing dataset size by filtering out noisy or redundant examples.

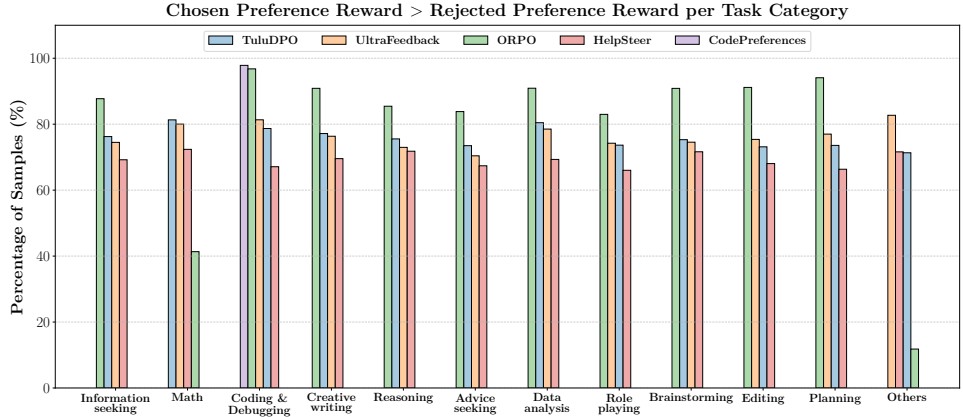

Figure 17: Distribution of preference rewards across task categories for all examined open-source DPO datasets. We present the reward-aligned proportions in which the chosen completions are correctly preferred over the rejected ones (according to our reward model).

#### D.4.2 PREFERENCE REWARD VS. QUERY DIFFICULTY

Fig. 18 compares average preference rewards for chosen and rejected completions (filtered and aligned with our reward model) across different levels of query difficulty. We observe that, for both chosen and discarded responses, reward scores tend to increase as query difficulty increases. This extends our previous findings, suggesting that higher-difficulty prompts are more likely to be associated with high-reward completions. The trend is most prominent in Fig. 18b, while Fig. 18a shows Code-Preference-Pairs as a notable outlier. As discussed in the main body and observed in Fig. 11 and Fig. 17, this skew arises from the dataset's unique composition, often featuring code samples where the only difference between completions is the presence or absence of inline comments, thus resulting in a disproportionate distribution of preference rewards across difficulty levels.

In addition, Fig. 19 shows the distribution of average chosen preference rewards across query difficulty levels for each DPO dataset individually, providing further insights into per-dataset-level characteristics. Specifically, Fig. 19a to Fig. 19e confirm our above statements, showing that more difficult samples tend to yield higher preference reward scores as observed by the visible shifts in chosen preference rewards for increasing difficulty across datasets.

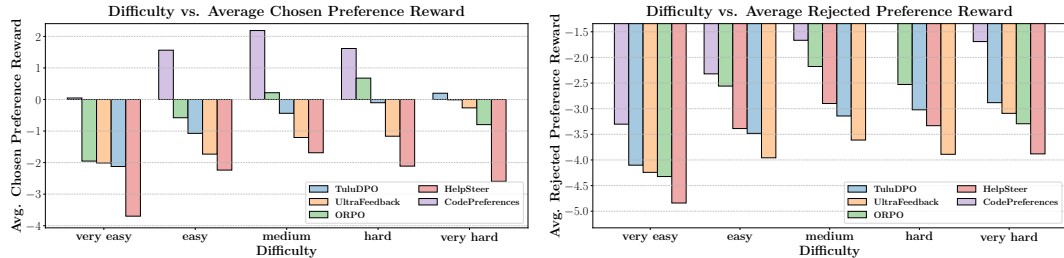

(a) Difficulty vs. average chosen preference reward.

(b) Difficulty vs. average rejected preference reward.

Figure 18: Comparison of average preference rewards against query difficulty.

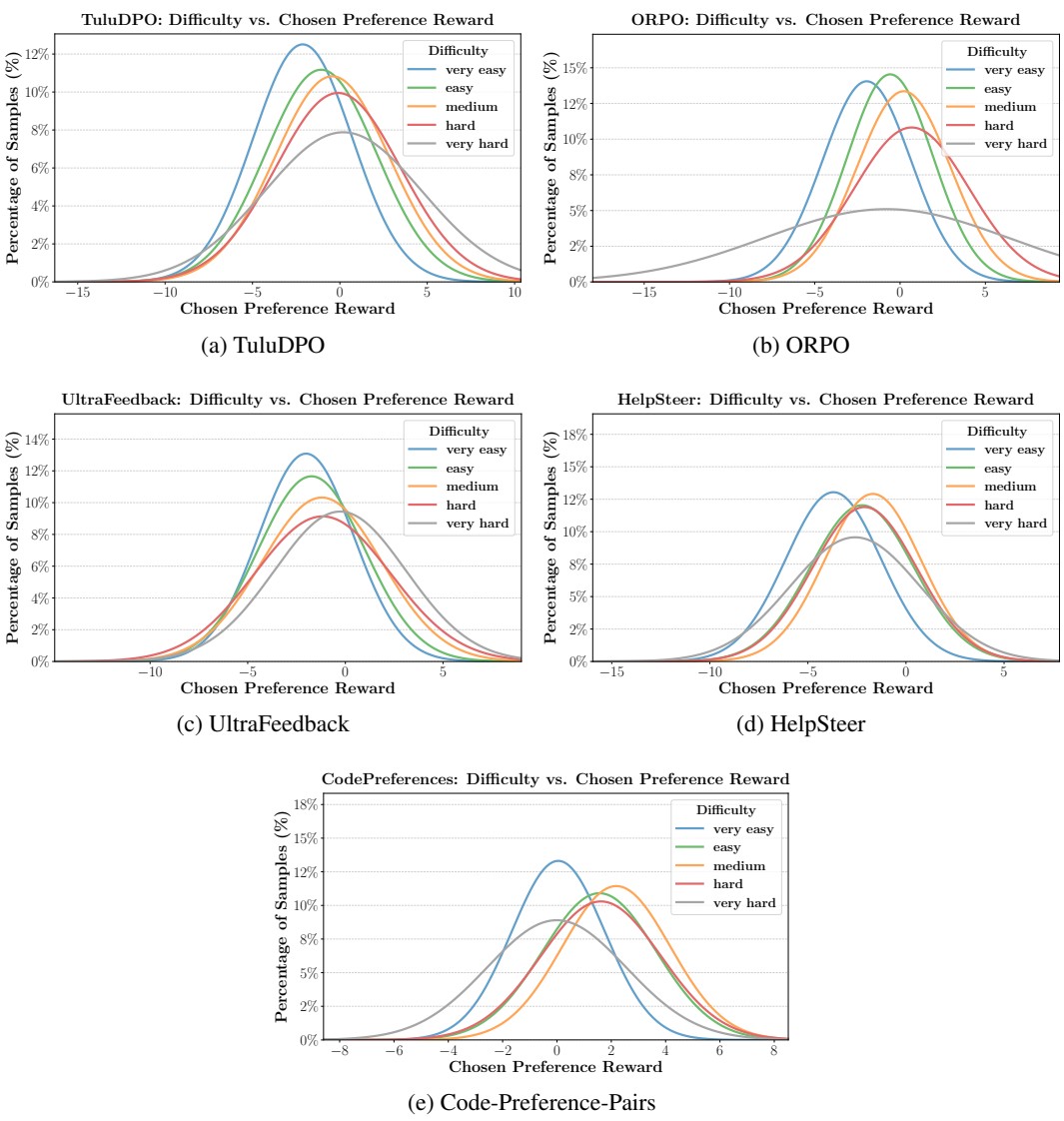

(a) TuluDPO

(b) ORPO

(c) UltraFeedback

(d) HelpSteer

(e) Code-Preference-Pairs

Figure 19: Comparison of chosen preference reward distributions against query difficulty for all examined open-source DPO datasets.

### D.4.3 PREFERENCE REWARD VS. INPUT QUALITY

Fig. 20 compares average preference rewards for chosen and rejected completions (filtered and aligned with our reward model) across different levels of input quality. We observe that, similarly to query difficulty, reward scores tend to increase as input quality increases. This further corroborates our previous statements, suggesting that high-quality instructions are likely to be associated with high-reward completions and are thus an important factor for effective DPO alignment. We provide additional per-dataset-level distributions in Fig. 22 for all our examined open-source DPO datasets.

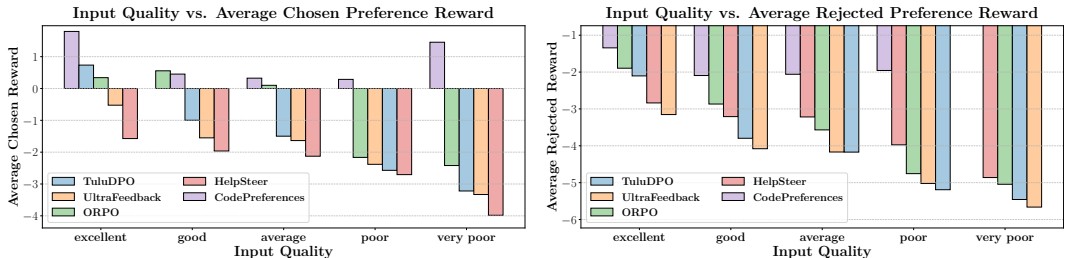

(a) Input quality vs. average chosen preference reward.  (b) Input quality vs. average rejected preference reward.

Figure 20: Comparison of average preference rewards against input quality.

## D.5 LANGUAGE AND SAFETY ANALYSIS

Fig. 21 shows the distribution of language and safety across all examined open-source DPO datasets. We find that most datasets are predominantly English (96–99%), with the exception of TuluDPO, which includes approximately 12% non-English samples, primarily in French and German. All datasets are also rated as overwhelmingly safe (95–98%). As already discussed by Djuhera et al. (2025), language and safety are not significantly correlated with post-training performance. For this reason, we do not include these attributes in our data curation recipes. Nevertheless, we release all language and safety labels as part of our annotated datasets.

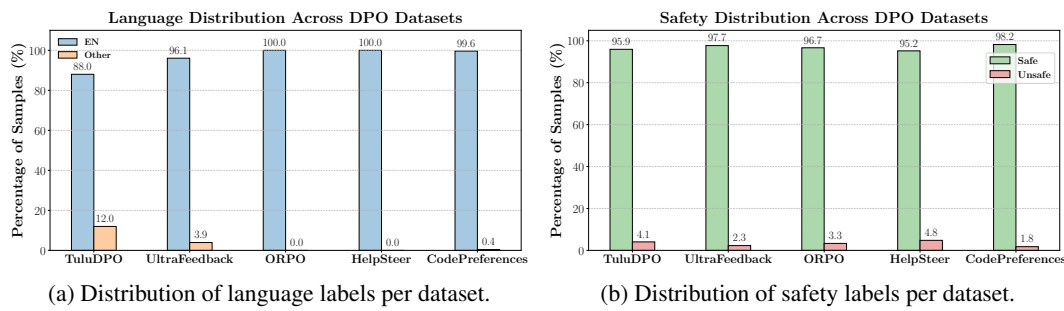

(a) Distribution of language labels per dataset.  (b) Distribution of safety labels per dataset.

Figure 21: Distribution of language and safety labels for all examined open-source DPO datasets.

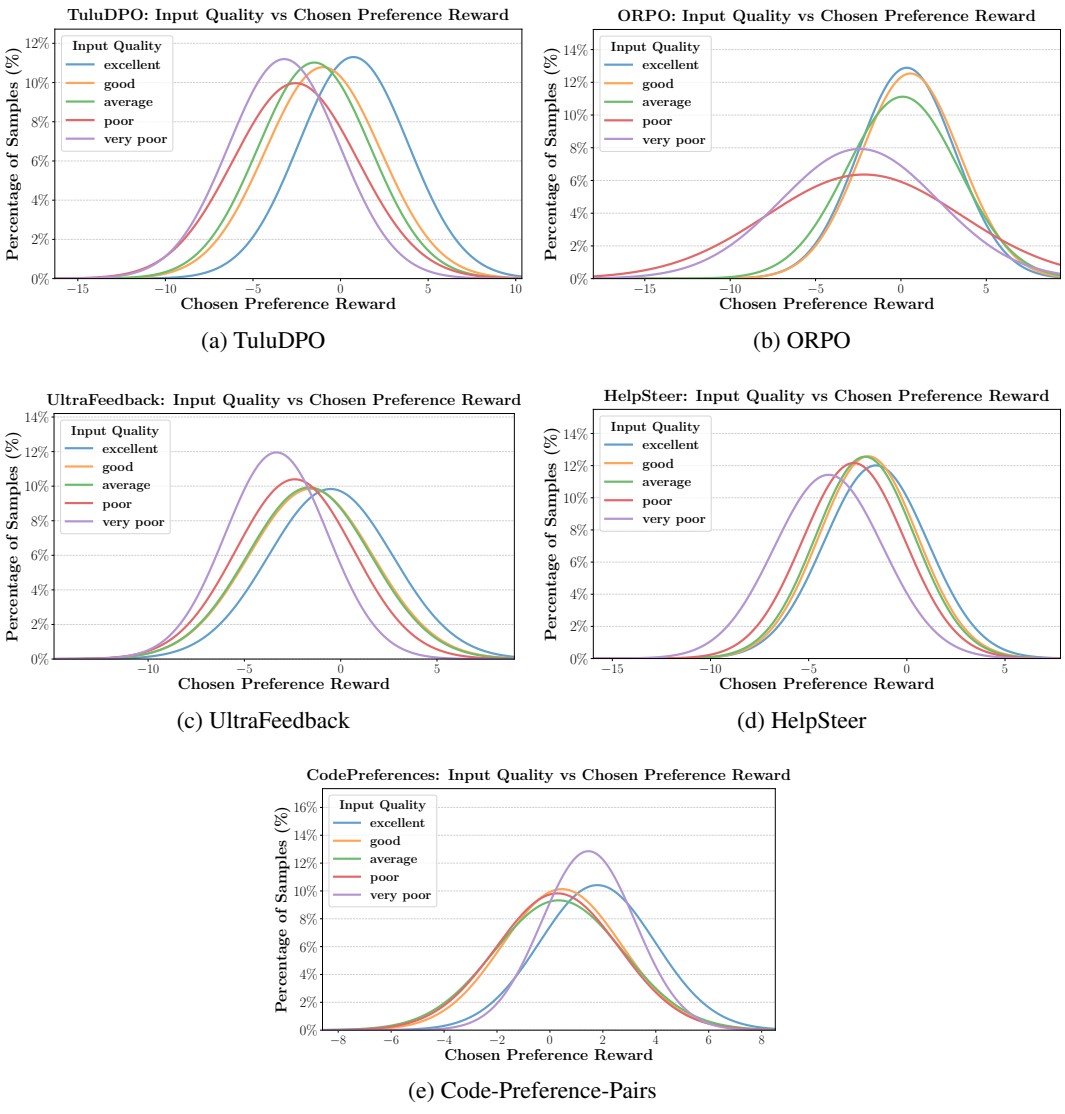

(a) TuluDPO

(b) ORPO

(c) UltraFeedback

(d) HelpSteer

(e) Code-Preference-Pairs

Figure 22: Comparison of chosen preference reward distributions against input quality for all examined open-source DPO datasets.

# E    DATA CURATION RECIPE DETAILS

This section details the quality-, reward-, and task-based data curation procedure used to construct our *UltraMix* DPO mixture. The general steps of the algorithm are summarized in Fig. 23.

In **Step 1**, we filter each dataset to retain only samples that satisfy three basic conditions:

- First: input quality is rated at least *"good"*.
- Second: difficulty is above *"very easy"*.
- Third: the chosen response achieves a higher preference reward than the discarded one.

This ensures that only meaningful preference pairs with coherent alignment signals are considered.

**Step 2** applies reward-based thresholds:

- For TuluDPO, ORPO, UltraFeedback, and HelpSteer, we retain samples with preference rewards above the 25th percentile of each dataset.
- For Code-Preference-Pairs, which is substantially larger and only contains code samples, we apply a stricter cutoff at the 80th percentile to avoid over-representing code tasks.

This step prioritizes high-reward, high-quality pairs while maintaining dataset diversity.

**Step 3** performs deduplication across sources. Since TuluDPO includes significant overlap with UltraFeedback, we remove duplicate prompts to prevent overweighting repeated instructions. Despite the two datasets containing different completions for identical prompts, we justify deduplication as differences are generally minor and mostly stylistic, likely due to the use of similar LLM families during response generation. Deduplication eventually yields our initial *UltraMix-170k* mixture.

**Step 4** augments underrepresented domains to restore task diversity. We first add back 25% more high-reward samples from all five datasets, yielding *UltraMix-187k* after deduplication. Next, to specifically strengthen instruction following performance, we selectively reintroduce additional information seeking and reasoning samples above the 70th percentile reward threshold, even when their input quality is slightly below *good*. After deduplication, this adds 3,000 instruction following samples, resulting in *UltraMix-190k*.

Together, these steps yield our final mixture, *UltraMix-190k* (referred to as *UltraMix*), a lean, high-quality mixture that is both reward-aligned and task-balanced. As shown in Table 2, *UltraMix-190k* outperforms the larger TuluDPO dataset across multiple benchmarks while containing 30% fewer samples, validating the effectiveness of our curation strategy. Table 9 presents the composition of all UltraMix variants in terms of their source DPO datasets, showing that the majority of *UltraMix* samples (81%) originate from the high-performing TuluDPO dataset. This makes our curation process a rigorous, reward- and quality-based filtering of TuluDPO.

Table 9: Composition of UltraMix variants, shown as percentages of their source DPO datasets.

| Source DPO Dataset | UltraMix-170k | UltraMix-187k | UltraMix-190k (**UltraMix**) |
|---|---|---|---|
| TuluDPO | 68.02% | 76.17% | 81.13% |
| UltraFeedback | 15.00% | 9.20% | 4.42% |
| ORPO | 9.01% | 5.47% | 5.46% |
| HelpSteer | 1.96% | 1.63% | 1.74% |
| Code-Preference-Pairs | 6.02% | 7.52% | 7.25% |

---

**Quality-, Reward-, and Task-Based Curation Recipe**

**Input:** Annotated datasets $\{\mathcal{D}_k\}_{k=1}^5$ with Magpie labels for input quality (`input_quality`), difficulty (`difficulty`), preference rewards (`reward_chosen, reward_rejected`), and task category (`task_category`).

In addition: per-dataset reward quantiles $\{q_k\}$ with a dedicated $q_{\text{code}}$ for the code-only corpus, instruction following categories $\mathcal{IF}$ (e.g., information seeking, reasoning), under-representation tolerance $\tau \in (0, 1)$, balancing thresholds $q_*$ (primary) and $r_{\text{avg}}$ (fallback).

**Output:** Curated set $\mathcal{D}_c$ that is high-quality, reward-aligned, and task-diverse.

**Recipe:**

1. **Initial Quality, Difficulty, and Reward Filter**: Build a candidate pool $\mathcal{P}$ of samples where
   $\mathcal{P} = \{S \in \cup_k \mathcal{D}_k \mid S[\text{input\_quality}] \in \{\text{good}, \text{excellent}\} \wedge S[\text{difficulty}] > \text{very easy} \wedge S[\text{reward\_chosen}] > S[\text{reward\_rejected}]\}$.

2. **Reward Thresholding**: For each dataset $k$, compute $P_{q_k}^{(k)}$ as the $q_k$-th percentile of $\{S[\text{reward\_chosen}] : S \in \mathcal{P} \cap \mathcal{D}_k\}$. Initialize

   $$\mathcal{D}_c \leftarrow \left\{S \in \mathcal{P} \cap \mathcal{D}_k \,\middle|\, S[\text{reward\_chosen}] \geq \begin{cases} P_{q_k}^{(k)}, & k \neq \text{code}, \\ P_{q_{\text{code}}}^{(\text{code})}, & k = \text{code} \end{cases}\right\}.$$

   *Note:* Raising/lowering $q_k$ (or $q_{\text{code}}$) globally tightens/loosens the mixture; no absolute sample counts are required.

3. **Task Coverage Check**: Let $\pi_{\mathcal{D}}(c)$ and $\pi_{\mathcal{D}_c}(c)$ denote the fraction of samples in task $c$ for the full union $\mathcal{D} = \cup_k \mathcal{D}_k$ and the current $\mathcal{D}_c$, respectively. Define the under-represented set as

   $$\mathcal{C}_\downarrow = \left\{c \,\middle|\, \pi_{\mathcal{D}_c}(c) < (1 - \tau)\,\pi_{\mathcal{D}}(c)\right\}.$$

4. **Task Boosting (for Instruction Following)**: For each $c \in \mathcal{C}_\downarrow \cap \mathcal{IF}$
   (a) form the residual pool $\mathcal{R}_c = \{S \in \mathcal{P} \setminus \mathcal{D}_c \mid S[\text{task\_category}] = c\}$,
   (b) compute the category-specific high-reward cutoff $P_{q_*}^{(c)}$ over $\{S[\text{reward\_chosen}] : S \in \mathcal{R}_c, S[\text{input\_quality}] \in \{\text{good}, \text{excellent}\}\}$,
   (c) add $\mathcal{D}_c \leftarrow \mathcal{D}_c \cup \{S \in \mathcal{R}_c \mid S[\text{input\_quality}] \in \{\text{good}, \text{excellent}\}, S[\text{reward\_chosen}] \geq P_{q_*}^{(c)}\}$.

   *Fallback (Quality Relaxation):* If this set is empty, recompute a looser cutoff $P_{r_{\text{avg}}}^{(c)}$ over *average*-quality samples in $\mathcal{R}_c$ and add

   $$\{S \in \mathcal{R}_c \mid S[\text{input\_quality}] = \text{average}, S[\text{reward\_chosen}] \geq P_{r_{\text{avg}}}^{(c)}\}.$$

   Repeat until $\pi_{\mathcal{D}_c}(c) \geq (1 - \tau)\pi_{\mathcal{D}}(c)$ or $\mathcal{R}_c$ is exhausted.

5. **Deduplication**: Compute a hash $h(S)$ over the prompts. For any duplicate hash $h$, keep only the instance with the highest $S[\text{reward\_chosen}]$.

Figure 23: Data curation algorithm for quality-, reward-, and task-based DPO mixtures. Steps 1–2 apply margin and per-dataset reward quantiles (with a separate code threshold), Step 3 detects task under-representation, Step 4 restores instruction following coverage using a primary threshold $q_*$ and a quality-relaxed fallback $r_{\text{avg}}$, and Step 5 deduplicates by prompt identity.

# F  EXPERIMENTAL SETUP AND ADDITIONAL RESULTS

This section presents supplementary DPO training results and provides details on the fine-tuning and evaluation configurations used in our experiments.

## F.1  FINE-TUNING CONFIGURATIONS

To ensure reproducibility and comparability, we fine-tune all models for both SFT and DPO using AllenAI's *Open-Instruct* framework[8]. Training was performed using BF16 mixed precision with Fully Sharded Data Parallelism (FSDP) on 8 × NVIDIA A100 80GB GPUs.

### F.1.1  SUPERVISED FINE-TUNING (SFT)

Since Llama-3.1-8B (Grattafiori et al., 2024) and Qwen-2.5-7B (Yang et al., 2025) do not provide open SFT checkpoints, we fine-tune both models on the same SFT dataset to ensure consistency. Specifically, we use TuluSFT, a well-established, popular dataset curated by Lambert et al. (2025) for post-training Llama-3.1 models. Given that Llama and Qwen are of comparable size, we adopt the same hyperparameters for SFT tuning. The full training configurations are summarized in Table 10.

We note that by default, Open-Instruct applies a sum-reduction over token-level losses, rather than the more commonly used mean-reduction. This design choice ensures length-equitable weighting, where short and long sequences contribute proportionally to the total loss, preventing shorter examples from disproportionately influencing the gradient due to having fewer tokens. This leads to more stable optimization by avoiding fluctuations in loss scale caused by variation in batch composition or sequence length distributions. We refer to a more detailed discussion in Lambert et al. (2025).

Table 10: SFT training hyperparameters for Llama-3.1-8B and Qwen-2.5-7B on the TuluSFT dataset.

| Parameter | Llama-3.1-8B | Qwen-2.5-7B |
|---|---|---|
| Total Batch Size | 128 | 128 |
| Per-Device Batch Size | 1 | 1 |
| Gradient Accumulation Steps | 16 | 16 |
| Max Sequence Length | 4096 | 4096 |
| Number of Epochs | 2 | 2 |
| Learning Rate | $5 \times 10^{-6}$ | $5 \times 10^{-6}$ |
| LR Scheduler | Linear | Linear |
| Warmup Ratio | 0.03 | 0.03 |
| Weight Decay | 0.0 | 0.1 |

### F.1.2  DIRECT PREFERENCE OPTIMIZATION (DPO)

Unlike SFT, where loss aggregation across tokens requires careful handling to avoid length bias, the DPO objective is inherently length-normalized (Rafailov et al., 2023; Lambert et al., 2025). This formulation ensures that completions of different lengths are treated equitably, and it avoids instabilities from batch composition or sequence length distributions observed in prior SFT pipelines.

To ensure consistency, we fix the training hyperparameters for each model across all datasets and apply DPO strictly on models that have already been fine-tuned via SFT using their instruction-tuning datasets. If available, we use the SFT checkpoints from HuggingFace. For Llama-3.1-8B-TuluSFT and Qwen-2.5-7B-TuluSFT, we adopt the same hyperparameters as in Lambert et al. (2025). For the additional six open models, Apertus-8B-SFT (Apertus-Team, 2025), OLMo-2-7B-SFT (OLMo et al., 2024), SmolLM-3-3B-SFT (Bakouch et al., 2025), Instella-3B-SFT (Liu et al., 2025), SmolLM-2-1.7B-SFT (Allal et al., 2025), and OLMo-2-1B-SFT (OLMo et al., 2024), we adopt the hyperparameters provided in their respective papers. Furthermore, we use the length-normalized DPO loss (`dpo_loss_type=norm`), following the recommendation in Lambert et al. (2025). Tables 11 and 12 report the DPO training configurations used for large and medium/small models, respectively.

---

[8]`https://github.com/allenai/open-instruct`

Table 11: DPO training hyperparameters for Llama-3.1-8B-TuluSFT, Qwen-2.5-8B-TuluSFT, Apertus-8B-SFT, and OLMo-2-7B-SFT.

| Parameter | Llama-3.1-8B-TuluSFT | Qwen-2.5-8B-TuluSFT | Apertus-8B-SFT | OLMo-2-7B-SFT |
|---|---|---|---|---|
| Total Batch Size | 128 | 128 | 128 | 128 |
| Per-Device Batch Size | 1 | 1 | 1 | 1 |
| Gradient Accumulation Steps | 16 | 16 | 16 | 16 |
| Max Sequence Length | 2048 | 2048 | 2048 | 2048 |
| Number of Epochs | 1 | 1 | 1 | 1 |
| Learning Rate | $5 \times 10^{-7}$ | $5 \times 10^{-7}$ | $5 \times 10^{-7}$ | $1 \times 10^{-6}$ |
| LR Scheduler | Linear | Linear | Linear | Linear |
| Warmup Ratio | 0.1 | 0.1 | 0.1 | 0.1 |
| Weight Decay | 0.0 | 0.0 | 0.0 | 0.0 |
| DPO Beta | 5 | 5 | 5 | 5 |

Table 12: DPO training hyperparameters for SmolLM-3-3B-SFT, Instella-3B-SFT, SmolLM-2-1.7B-SFT, and OLMo-2-1B-SFT.

| Parameter | SmolLM-3-3B-SFT | Instella-3B-SFT | SmolLM2-1.7B-SFT | OLMo-2-1B-SFT |
|---|---|---|---|---|
| Total Batch Size | 128 | 128 | 128 | 128 |
| Per-Device Batch Size | 1 | 1 | 2 | 2 |
| Gradient Accumulation Steps | 16 | 16 | 8 | 8 |
| Max Sequence Length | 2048 | 2048 | 1024 | 2048 |
| Number of Epochs | 1 | 1 | 2 | 1 |
| Learning Rate | $5 \times 10^{-7}$ | $5 \times 10^{-7}$ | $1 \times 10^{-6}$ | $2.5 \times 10^{-6}$ |
| LR Scheduler | Linear | Linear | Linear | Linear |
| Warmup Ratio | 0.1 | 0.1 | 0.1 | 0.1 |
| Weight Decay | 0.0 | 0.0 | 0.0 | 0.0 |
| DPO Beta | 5 | 5 | 5 | 0.5 |

## F.2 EVALUATION SETUP

We assess model performance using the *LM Evaluation Harness* framework (Gao et al., 2024), a widely adopted standard for evaluating language models across diverse benchmark suites. To ensure a comprehensive and task-diverse evaluation, we include benchmarks from *Open LLM Leaderboards* V1 and V2 (Fourrier et al., 2023; 2024) to gauge downstream task performance under competitive public standards. This spans *Knowledge* (i.e., MMLU (Hendrycks et al., 2021a), MMLU-Pro (Wang et al., 2024c), TruthfulQA (Lin et al., 2022), GPQA (Rein et al., 2023)), *Reasoning* (i.e., ARC-C (Clark et al., 2018), BBH (Suzgun et al., 2022), MuSR (Sprague et al., 2024)), *Commonsense Understanding* (i.e., HellaSwag (Zellers et al., 2019), WinoGrande (Sakaguchi et al., 2019)), *Instruction Following* (i.e., IF-Eval (Zhou et al., 2023)), and *Mathematical Reasoning* (i.e., GSM8K (Cobbe et al., 2021), MATH (Hendrycks et al., 2021b)). In addition, we evaluate *Coding* via HumanEval and HumanEval+ (Chen et al., 2021). These benchmarks ensure a fair comparison between models and data mixtures.

## F.3 ADDITIONAL DPO RESULTS

To demonstrate the effectiveness of UltraMix across different architectures and scales, we evaluate six additional open models that have publicly released their corresponding SFT checkpoints: Apertus-8B-SFT (Apertus-Team, 2025), OLMo-2-7B-SFT (OLMo et al., 2024), SmolLM-3-3B-SFT (Bakouch et al., 2025), Instella-3B-SFT (Liu et al., 2025)), SmolLM-2-1.7B-SFT (Allal et al., 2025), and OLMo-2-1B-SFT (OLMo et al., 2024). In contrast to our TuluSFT-tuned Llama and Qwen variants, this enables an important axis for evaluation with models that were SFT-tuned using different datasets.

Tables 13 to 18 report evaluation results across all considered benchmarks and DPO datasets, including our curated mixtures: UltraMix-170k (UM-170k), UltraMix-187k (UM-187k), and UltraMix-190k (UM-190k). Among the original DPO datasets, TuluDPO performs best, followed by UltraFeedback and ORPO. Our curated UltraMix-190k mixture consistently outperforms TuluDPO, achieving substantial improvements on nearly all benchmarks while being 30% smaller in size. These additional results corroborate the findings presented in the main paper and further demonstrate the effectiveness of our curated UltraMix dataset across diverse model architectures and scales.

### F.3.1 APERTUS-8B-SFT

Table 13: DPO results for Apertus-8B-SFT trained on all datasets, including our curated mixtures UM-170k, UM-187k, and UM-190k, evaluated on Open LLM Leaderboards (averaged) and code benchmarks. The overall average is across all benchmarks. Best scores are in **bold**.

| Benchmark | SFT | Original DPO Datasets | | | | | UltraMix | | |
|---|---|---|---|---|---|---|---|---|---|
| | | TuluDPO | ORPO | UltraFB | HelpSteer | CodePref | UM-170k | UM-187k | UM-190k |
| *Knowledge* | | | | | | | | | |
| MMLU (5-shot) | 60.87 | 61.11 | 60.92 | 60.37 | 59.85 | 57.94 | 61.79 | 62.25 | **62.79** |
| MMLU-Pro (5-shot) | 29.38 | 30.66 | 30.33 | 29.99 | 28.54 | 28.34 | 30.61 | 31.32 | **31.98** |
| TruthfulQA (0-shot) | 54.21 | 56.28 | 55.62 | 55.87 | 54.71 | 56.21 | 58.23 | **60.66** | 60.57 |
| GPQA (0-shot) | 25.67 | 27.43 | 27.52 | 27.43 | 27.35 | 26.85 | 27.35 | 28.35 | **28.77** |
| *Reasoning* | | | | | | | | | |
| ARC-C (25-shot) | 57.68 | 59.59 | 58.83 | 58.17 | 56.97 | 52.73 | 59.87 | **60.87** | 60.70 |
| BBH (3-shot) | 44.49 | 45.37 | 44.03 | 45.50 | 44.46 | 43.24 | 46.57 | 46.72 | **47.09** |
| MuSR (0-shot) | 35.85 | 35.07 | 34.98 | 35.94 | 34.47 | 32.48 | 35.94 | 36.07 | **36.28** |
| *Commonsense* | | | | | | | | | |
| HellaSwag (10-shot) | 57.32 | 59.52 | 57.82 | 57.14 | 56.90 | 51.31 | 59.90 | 60.69 | **61.62** |
| WinoGrande (5-shot) | 73.80 | 73.53 | 73.16 | 73.32 | 72.93 | 67.72 | 73.69 | 74.80 | **75.17** |
| *Instruction Following* | | | | | | | | | |
| IF-Eval (0-shot) | 65.32 | 72.36 | 70.79 | 70.61 | 71.69 | 66.39 | 72.13 | 74.16 | **74.45** |
| *Math* | | | | | | | | | |
| GSM8K (5-shot) | 53.07 | 59.97 | 57.16 | 60.80 | 55.27 | 54.28 | 60.10 | 61.29 | **62.15** |
| MATH (4-shot) | 4.53 | 9.65 | 8.06 | 8.70 | 6.08 | 6.80 | 9.06 | 10.97 | **11.59** |
| *Code (pass@1)* | | | | | | | | | |
| HumanEval | 40.24 | 46.34 | 43.07 | 42.07 | 40.02 | **51.24** | 45.46 | 47.68 | 48.07 |
| HumanEval+ | 26.22 | 30.41 | 28.05 | 27.32 | 26.10 | **35.24** | 30.63 | 32.61 | 33.11 |
| *Leaderboards* | | | | | | | | | |
| Open LLM Leaderboard 1 | 59.49 | 61.67 | 60.58 | 60.94 | 59.44 | 56.70 | 62.26 | 63.43 | **63.83** |
| Open LLM Leaderboard 2 | 34.21 | 36.76 | 35.95 | 36.36 | 35.43 | 34.02 | 36.94 | 37.93 | **38.36** |
| *Overall* | 44.90 | 47.66 | 46.45 | 46.66 | 45.38 | 45.06 | 47.95 | 49.17 | **49.60** |

### F.3.2 OLMO-2-7B-SFT

Table 14: DPO results for OLMo-2-7B-SFT trained on all datasets, including our curated mixtures UM-170k, UM-187k, and UM-190k, evaluated on Open LLM Leaderboards (averaged) and code benchmarks. The overall average is across all benchmarks. Best scores are in **bold**.

| Benchmark | SFT | Original DPO Datasets | | | | | UltraMix | | |
|---|---|---|---|---|---|---|---|---|---|
| | | TuluDPO | ORPO | UltraFB | HelpSteer | CodePref | UM-170k | UM-187k | UM-190k |
| *Knowledge* | | | | | | | | | |
| MMLU (5-shot) | 60.63 | 61.13 | 61.07 | 60.95 | 60.62 | 60.85 | 61.04 | 61.51 | **61.92** |
| MMLU-Pro (5-shot) | 25.94 | 27.10 | 26.39 | 26.75 | 26.02 | 25.78 | 27.10 | 27.69 | **27.84** |
| TruthfulQA (0-shot) | 48.61 | 55.95 | 54.57 | 53.77 | 50.16 | 50.41 | 57.67 | 57.95 | **58.23** |
| GPQA (0-shot) | 27.10 | 27.93 | 27.63 | 27.27 | 27.10 | 26.67 | 28.61 | 29.68 | **29.78** |
| *Reasoning* | | | | | | | | | |
| ARC-C (25-shot) | 55.29 | 56.23 | 56.06 | 56.48 | 55.55 | 52.47 | 57.17 | 57.30 | **57.57** |
| BBH (3-shot) | 38.88 | 39.26 | 39.65 | 39.44 | 38.85 | 38.15 | 40.45 | 40.83 | **41.09** |
| MuSR (0-shot) | 37.04 | 37.64 | 37.70 | 37.83 | 37.43 | 35.85 | 37.86 | 37.98 | **38.19** |
| *Commonsense* | | | | | | | | | |
| HellaSwag (10-shot) | 60.14 | 61.97 | 61.30 | 60.24 | 60.55 | 60.24 | 62.17 | 62.54 | **62.86** |
| WinoGrande (5-shot) | 76.56 | 77.19 | 76.92 | 76.66 | 76.40 | 75.35 | 77.54 | 77.95 | **78.03** |
| *Instruction Following* | | | | | | | | | |
| IF-Eval (0-shot) | 67.42 | 71.96 | 69.31 | 68.51 | 67.42 | 68.24 | 72.48 | 73.25 | **73.31** |
| *Math* | | | | | | | | | |
| GSM8K (5-shot) | 62.77 | 73.84 | 71.05 | 73.24 | 68.05 | 65.01 | 77.71 | 78.01 | **78.10** |
| MATH (4-shot) | 7.55 | 14.20 | 13.27 | 13.44 | 8.99 | 11.78 | 14.92 | 15.29 | **15.59** |
| *Code (pass@1)* | | | | | | | | | |
| HumanEval | 40.24 | 45.73 | 42.63 | 41.46 | 40.85 | **50.12** | 46.46 | 48.46 | 48.93 |
| HumanEval+ | 22.32 | 25.61 | 24.15 | 23.05 | 22.93 | **30.22** | 26.76 | 28.78 | 28.88 |
| *Leaderboards* | | | | | | | | | |
| Open LLM Leaderboard 1 | 60.67 | 64.39 | 63.50 | 63.56 | 61.89 | 60.72 | 65.55 | 65.88 | **66.12** |
| Open LLM Leaderboard 2 | 33.99 | 36.35 | 35.66 | 35.54 | 34.30 | 34.41 | 36.90 | 37.45 | **37.63** |
| *Overall* | 45.04 | 48.27 | 47.26 | 47.08 | 45.78 | 46.51 | 49.14 | 49.80 | **50.02** |

### F.3.3    SMOLLM-3-3B-SFT

Table 15: DPO results for SmolLM-3-3B-SFT trained on all datasets, including our curated mixtures UM-170k, UM-187k, and UM-190k, evaluated on Open LLM Leaderboards (averaged) and code benchmarks. The overall average is across all benchmarks. Best scores are in **bold**.

| Benchmark | SFT | Original DPO Datasets | | | | | UltraMix | | |
| | | TuluDPO | ORPO | UltraFB | HelpSteer | CodePref | UM-170k | UM-187k | UM-190k |
| --- | --- | --- | --- | --- | --- | --- | --- | --- | --- |
| *Knowledge* | | | | | | | | | |
| MMLU (5-shot) | 54.24 | 55.26 | 54.90 | 50.23 | 50.30 | 50.33 | 55.16 | 56.90 | **56.57** |
| MMLU-Pro (5-shot) | 25.92 | 26.41 | 26.12 | 23.90 | 23.48 | 22.41 | 26.32 | 27.14 | **27.37** |
| TruthfulQA (0-shot) | 52.14 | 60.28 | 58.46 | 49.21 | 49.33 | 49.79 | 59.05 | 60.25 | **60.57** |
| GPQA (0-shot) | 28.57 | 30.29 | 29.78 | 26.01 | 26.43 | 26.01 | 30.79 | 31.42 | **32.03** |
| *Reasoning* | | | | | | | | | |
| ARC-C (25-shot) | 54.27 | 56.73 | 55.03 | 52.68 | 52.72 | 51.71 | 56.77 | 57.51 | **57.45** |
| BBH (3-shot) | 38.43 | 40.32 | 39.31 | 36.81 | 37.35 | 35.33 | 39.98 | 41.33 | **41.46** |
| MuSR (0-shot) | 40.92 | 45.68 | 44.81 | 39.11 | 39.39 | 38.58 | 44.38 | 45.43 | **46.19** |
| *Commonsense* | | | | | | | | | |
| HellaSwag (10-shot) | 53.24 | 58.58 | 54.98 | 54.90 | 54.22 | 52.91 | 58.85 | 59.35 | **59.81** |
| WinoGrande (5-shot) | 71.74 | 74.01 | 72.22 | 69.54 | 70.56 | 66.12 | 72.35 | 73.14 | **73.38** |
| *Instruction Following* | | | | | | | | | |
| IF-Eval (0-shot) | 68.47 | 70.19 | 69.74 | 62.61 | 64.41 | 61.40 | 69.25 | 72.49 | **72.81** |
| *Math* | | | | | | | | | |
| GSM8K (5-shot) | 59.22 | 65.17 | 63.73 | 62.45 | 65.88 | 61.14 | 64.99 | 66.59 | **67.27** |
| MATH (4-shot) | 24.04 | 31.21 | 29.68 | 27.19 | 27.95 | 24.46 | 30.98 | 32.07 | **32.50** |
| *Code (pass@1)* | | | | | | | | | |
| HumanEval | 52.20 | 65.25 | 63.61 | 57.97 | 60.01 | **67.07** | 65.72 | 66.84 | 66.96 |
| HumanEval+ | 29.20 | 32.79 | 31.27 | 27.08 | 29.05 | **35.62** | 33.07 | 33.97 | 34.20 |
| *Leaderboards* | | | | | | | | | |
| Open LLM Leaderboard 1 | 57.48 | 61.67 | 59.89 | 56.50 | 57.17 | 55.33 | 61.19 | 62.29 | **62.51** |
| Open LLM Leaderboard 2 | 37.72 | 40.68 | 39.91 | 35.94 | 36.50 | 34.70 | 40.28 | 41.65 | **42.06** |
| *Overall* | 46.61 | 50.87 | 49.55 | 45.69 | 46.51 | 45.92 | 50.55 | 51.74 | **52.04** |

### F.3.4    INSTELLA-3B-SFT

Table 16: DPO results for Instella-3B-SFT trained on all datasets, including our curated mixtures UM-170k, UM-187k, and UM-190k, evaluated on Open LLM Leaderboards (averaged) and code benchmarks. The overall average is across all benchmarks. Best scores are in **bold**.

| Benchmark | SFT | Original DPO Datasets | | | | | UltraMix | | |
| | | TuluDPO | ORPO | UltraFB | HelpSteer | CodePref | UM-170k | UM-187k | UM-190k |
| --- | --- | --- | --- | --- | --- | --- | --- | --- | --- |
| *Knowledge* | | | | | | | | | |
| MMLU (5-shot) | 57.14 | 58.19 | 57.93 | 57.43 | 57.54 | 57.39 | 57.70 | 58.28 | **58.96** |
| MMLU-Pro (5-shot) | 25.44 | 26.06 | 25.95 | 25.14 | 25.39 | 25.19 | 26.44 | 27.17 | **27.86** |
| TruthfulQA (0-shot) | 52.48 | 54.21 | 53.35 | 52.67 | 52.70 | 52.19 | 54.45 | 54.91 | **54.93** |
| GPQA (0-shot) | 25.84 | 26.76 | 26.43 | 26.01 | 26.43 | 26.10 | 27.35 | 27.43 | **27.46** |
| *Reasoning* | | | | | | | | | |
| ARC-C (25-shot) | 52.13 | 53.16 | 52.22 | 52.30 | 52.56 | 50.77 | 53.16 | 53.67 | **53.92** |
| BBH (3-shot) | 42.68 | 43.39 | 43.19 | 43.05 | 42.87 | 42.39 | 44.62 | 44.89 | **44.95** |
| MuSR (0-shot) | 36.24 | 36.92 | 36.19 | 35.98 | 35.98 | 35.71 | 37.39 | 37.59 | **37.99** |
| *Commonsense* | | | | | | | | | |
| HellaSwag (10-shot) | 56.20 | 57.51 | 56.72 | 56.30 | 56.24 | 55.99 | 58.23 | 58.56 | **58.74** |
| WinoGrande (5-shot) | 71.82 | 72.05 | 71.86 | 70.64 | 71.19 | 66.46 | 72.22 | 72.69 | **72.89** |
| *Instruction Following* | | | | | | | | | |
| IF-Eval (0-shot) | 66.84 | 74.02 | 71.34 | 65.57 | 67.24 | 64.80 | 72.54 | 74.52 | **75.35** |
| *Math* | | | | | | | | | |
| GSM8K (5-shot) | 67.93 | 70.66 | 69.85 | 70.58 | 68.31 | 67.55 | 71.95 | 72.57 | **72.74** |
| MATH (4-shot) | 12.31 | 15.71 | 14.39 | 13.52 | 12.76 | 12.08 | 15.56 | 16.47 | **16.88** |
| *Code (pass@1)* | | | | | | | | | |
| HumanEval | 31.76 | 35.32 | 35.37 | 31.71 | 32.98 | **39.48** | 36.61 | 37.83 | 37.98 |
| HumanEval+ | 10.65 | 14.29 | 13.47 | 10.70 | 12.94 | **16.04** | 15.06 | 15.49 | 15.61 |
| *Leaderboards* | | | | | | | | | |
| Open LLM Leaderboard 1 | 59.62 | 60.96 | 60.32 | 59.99 | 59.76 | 58.39 | 61.28 | 61.78 | **62.03** |
| Open LLM Leaderboard 2 | 34.89 | 37.14 | 36.25 | 34.88 | 35.11 | 34.38 | 37.32 | 38.01 | **38.42** |
| *Overall* | 43.53 | 45.59 | 44.88 | 43.69 | 43.94 | 43.72 | 45.95 | 46.58 | **46.88** |

### F.3.5 SMOLLM-2-1.7B-SFT

Table 17: DPO results for SmolLM-2-1.7B-SFT trained on all datasets, including our curated mixtures UM-170k, UM-187k, and UM-190k, evaluated on Open LLM Leaderboards (averaged) and code benchmarks. The overall average is across all benchmarks. Best scores are in **bold**.

| Benchmark | SFT | Original DPO Datasets | | | | | UltraMix | | |
| | | TuluDPO | ORPO | UltraFB | HelpSteer | CodePref | UM-170k | UM-187k | UM-190k |
|---|---|---|---|---|---|---|---|---|---|
| *Knowledge* | | | | | | | | | |
| MMLU (5-shot) | 49.46 | 49.49 | 49.63 | 48.65 | 48.50 | 47.47 | 48.97 | 49.32 | **49.89** |
| MMLU-Pro (5-shot) | 18.54 | 19.76 | 19.44 | 18.98 | 18.81 | 18.27 | 19.31 | 20.50 | **20.87** |
| TruthfulQA (0-shot) | 40.94 | 45.73 | 45.50 | 39.18 | 41.19 | 44.43 | 48.10 | **50.38** | 50.24 |
| GPQA (0-shot) | 27.60 | 28.27 | 27.94 | 28.10 | 28.61 | 25.59 | 27.92 | 29.85 | **30.43** |
| *Reasoning* | | | | | | | | | |
| ARC-C (25-shot) | 48.89 | **50.94** | 49.32 | 48.98 | 46.84 | 46.84 | 48.62 | 49.82 | 50.48 |
| BBH (3-shot) | 35.62 | 36.08 | 35.48 | 35.29 | 35.71 | 34.65 | 35.24 | 36.24 | **36.96** |
| MuSR (0-shot) | 33.60 | 34.09 | 34.31 | 31.35 | 32.67 | 30.48 | 33.22 | 36.66 | **37.14** |
| *Commonsense* | | | | | | | | | |
| HellaSwag (10-shot) | 52.91 | 53.42 | 53.69 | 53.98 | 52.73 | 50.23 | 52.84 | 54.93 | **55.09** |
| WinoGrande (5-shot) | 65.32 | 67.01 | 66.85 | 66.93 | 66.77 | 62.35 | 65.48 | 67.27 | **67.86** |
| *Instruction Following* | | | | | | | | | |
| IF-Eval (0-shot) | 50.54 | 51.97 | 51.76 | 50.02 | 45.46 | 42.79 | 50.80 | 51.81 | **52.54** |
| *Math* | | | | | | | | | |
| GSM8K (5-shot) | 46.84 | 48.84 | 48.98 | 47.99 | 48.07 | 44.98 | 47.96 | 49.45 | **50.14** |
| MATH (4-shot) | 4.46 | 5.63 | 5.87 | 5.44 | 4.08 | 4.42 | 5.35 | 5.82 | **6.05** |
| *Code (pass@1)* | | | | | | | | | |
| HumanEval | 1.22 | 1.83 | 1.83 | 1.22 | 1.22 | **2.44** | 1.83 | **2.44** | 2.44 |
| HumanEval+ | 0.61 | 1.22 | 1.22 | 0.61 | 0.61 | **1.83** | 1.22 | 1.83 | 1.83 |
| *Leaderboards* | | | | | | | | | |
| Open LLM Leaderboard 1 | 50.73 | 52.57 | 52.33 | 50.95 | 50.68 | 49.38 | 52.00 | 53.53 | **53.95** |
| Open LLM Leaderboard 2 | 28.39 | 29.30 | 29.13 | 28.20 | 27.56 | 26.03 | 28.64 | 30.15 | **30.67** |
| *Overall* | 34.04 | 35.31 | 35.13 | 34.05 | 33.66 | 32.63 | 34.78 | 36.17 | **36.57** |

### F.3.6 OLMO-2-1B-SFT

Table 18: DPO results for OLMo-2-1B-SFT trained on all datasets, including our curated mixtures UM-170k, UM-187k, and UM-190k, evaluated on Open LLM Leaderboards (averaged) and code benchmarks. The overall average is across all benchmarks. Best scores are in **bold**.

| Benchmark | SFT | Original DPO Datasets | | | | | UltraMix | | |
| | | TuluDPO | ORPO | UltraFB | HelpSteer | CodePref | UM-170k | UM-187k | UM-190k |
|---|---|---|---|---|---|---|---|---|---|
| *Knowledge* | | | | | | | | | |
| MMLU (5-shot) | 40.62 | 41.78 | 41.66 | 41.21 | 40.54 | 40.16 | 41.45 | 42.04 | **42.17** |
| MMLU-Pro (5-shot) | 13.02 | 14.07 | 14.01 | 13.45 | 12.83 | 14.28 | 14.00 | 14.64 | **14.77** |
| TruthfulQA (0-shot) | 42.07 | 44.40 | 44.34 | 43.67 | 43.75 | 41.63 | 44.51 | 44.63 | **44.86** |
| GPQA (0-shot) | 26.09 | 26.32 | 26.49 | 25.92 | 25.00 | 23.99 | 26.90 | 27.04 | **27.09** |
| *Reasoning* | | | | | | | | | |
| ARC-C (25-shot) | 38.14 | 43.60 | 42.02 | 41.81 | 38.65 | 38.40 | 43.34 | 44.17 | **44.30** |
| BBH (3-shot) | 33.29 | 34.14 | 33.43 | 33.02 | 33.31 | 33.59 | 34.79 | 35.20 | **35.31** |
| MuSR (0-shot) | 34.52 | 35.41 | 34.20 | 33.88 | 33.73 | 34.32 | 36.07 | 36.52 | **36.85** |
| *Commonsense* | | | | | | | | | |
| HellaSwag (10-shot) | 48.53 | 52.35 | 50.18 | 49.84 | 49.34 | 50.45 | 52.29 | 53.15 | **53.41** |
| WinoGrande (5-shot) | 65.67 | 66.17 | 66.14 | 64.09 | 66.54 | 64.38 | 66.75 | 67.29 | **67.56** |
| *Instruction Following* | | | | | | | | | |
| IF-Eval (0-shot) | 53.49 | 65.56 | 62.49 | 60.76 | 58.07 | 56.28 | 65.12 | 67.04 | **67.13** |
| *Math* | | | | | | | | | |
| GSM8K (5-shot) | 37.15 | 50.36 | 49.35 | 49.48 | 43.73 | 40.44 | 50.51 | 52.13 | **52.36** |
| MATH (4-shot) | 3.47 | 4.64 | 4.15 | 4.08 | 4.15 | 3.42 | 4.18 | 5.02 | **5.27** |
| *Code (pass@1)* | | | | | | | | | |
| HumanEval | 19.88 | 29.51 | 26.52 | 23.73 | 26.83 | **32.61** | 30.02 | 31.24 | 31.51 |
| HumanEval+ | 10.12 | 18.54 | 16.80 | 14.63 | 13.37 | **20.19** | 18.98 | 19.61 | 19.83 |
| *Leaderboards* | | | | | | | | | |
| Open LLM Leaderboard 1 | 45.36 | 49.78 | 48.95 | 48.35 | 47.09 | 45.91 | 49.81 | 50.57 | **50.78** |
| Open LLM Leaderboard 2 | 27.31 | 30.02 | 29.13 | 28.52 | 27.85 | 27.65 | 30.18 | 30.91 | **31.07** |
| *Overall* | 33.29 | 37.63 | 36.56 | 35.68 | 34.99 | 35.30 | 37.78 | 38.55 | **38.74** |

## F.4 Assessing the Role of Individual Filtering Steps

To understand the contribution of each filtering step in our data-curation recipe, we provide granular ablations and show that UltraMix's effectiveness arises from the combination of diverse signals, rather than from coherence reward or quality-based filtering alone.

We first demonstrate that coherent preference rewards are essential for constructing effective DPO data mixtures and report evaluation results for UltraMix-No-Preference-Reward-Filter (UM-No-PF) in Table 19, a variant of UltraMix where we omitted the preference reward-based filtering step and only applied quality- and task-aware filtering according to our principled curation recipe in Fig. 23. The results show that despite careful quality-based filtering and task-aware addition of instruction following data, UM-No-PF underperforms both TuluDPO and our reward-filtered UltraMix variants UM-170k and UM-190k in overall and OpenLLM leaderboard scores. This suggest that DPO training is particularly sensitive to the preservation of coherent preference orderings, which implies that relying solely on quality and task-based filtering may introduce distortions that obscure preference consistency and degrade downstream performance.

To further substantiate these claims, we provide additional evaluations in Table 20 for (a) UM-only-PF, which applies preference-reward filtering only, and (b) UM-only-QF, which applies input-quality filtering only, and compare them against (c) UM-No-PF and (d) the full UltraMix recipe that combines all steps. The results show that no single step outperforms the TuluDPO baseline on its own. UM-only-QF and UM-No-PF generally outperform UM-only-PF, showing preference filtering is important but not the main driver. Thus, best performance is only achieved when every step is combined. These ablations clarify the role of each step in our curation recipe, demonstrating that all stages are complementary and the effectiveness of UltraMix stems from the combination of signals.

This important insight distinguishes our work from other post-training data curation recipes, such as for SFT (Djuhera et al., 2025), showing that DPO datasets require a joint curation that involves careful quality- and task-based curation, and relies on the preservation of a clear and consistent preference ordering. The preference reward filtering step in our principled curation recipe removes incoherent preference pairs (where a chosen completion receives a lower reward than its rejected counterpart), thereby improving consistency in preference assignment and allowing for subsequent quality- and task-based filtering.

Table 19: DPO results for Llama-3.1-8B-TuluSFT and Qwen-2.5-7B-TuluSFT trained on our curated DPO mixtures UltraMix-190k (UM-190k), UltraMix-170k (UM-170k), and UltraMix-No-Preference-Reward-Filter (UM-No-PF), compared to TuluDPO on Open LLM Leaderboards (averaged) and code benchmarks. The overall average is across all benchmarks. Best scores are in **bold**.

| Benchmark | Llama-3.1-8B-TuluSFT | | | | | Qwen-2.5-7B-TuluSFT | | | | |
| --- | --- | --- | --- | --- | --- | --- | --- | --- | --- | --- |
| | SFT | TuluDPO | UM-No-PF | UM-170k | UM-190k | SFT | TuluDPO | UM-No-PF | UM-170k | UM-190k |
| *Knowledge* | | | | | | | | | | |
| MMLU (5-shot) | 62.30 | 63.47 | 62.69 | 63.27 | **64.61** | 72.41 | 73.10 | 73.05 | 73.53 | **74.01** |
| MMLU-Pro (5-shot) | 28.08 | 28.98 | 28.55 | 28.50 | **30.96** | 43.32 | 43.48 | 43.52 | 43.16 | **44.65** |
| TruthfulQA (0-shot) | 46.84 | 56.78 | 56.80 | 61.45 | **63.32** | 51.64 | 54.46 | 55.20 | 57.60 | **58.00** |
| GPQA (0-shot) | 28.44 | 29.61 | 29.52 | 29.94 | **31.87** | 30.96 | 31.29 | 31.12 | 31.63 | **33.03** |
| *Reasoning* | | | | | | | | | | |
| ARC-C (25-shot) | 54.95 | 57.93 | 57.76 | 57.63 | **58.70** | 59.22 | 60.32 | 60.92 | 62.12 | **62.43** |
| BBH (3-shot) | 38.59 | 40.46 | 40.28 | 44.68 | **44.96** | 47.93 | 48.08 | 48.29 | 50.96 | **51.76** |
| MuSR (0-shot) | 43.25 | 41.93 | 41.87 | 42.43 | **44.02** | 48.15 | 47.16 | 47.30 | 47.87 | **48.82** |
| *Commonsense* | | | | | | | | | | |
| HellaSwag (10-shot) | 60.41 | **64.85** | 62.84 | 63.43 | 64.82 | 60.85 | 62.70 | 62.16 | 63.59 | **63.89** |
| WinoGrande (5-shot) | 76.40 | 75.30 | 75.72 | 74.98 | **77.06** | 73.17 | 72.96 | 72.19 | 73.69 | **74.64** |
| *Instruction Following* | | | | | | | | | | |
| IF-Eval (0-shot) | 72.45 | 80.35 | 79.93 | 79.38 | **81.13** | 68.06 | 78.04 | 75.39 | 77.28 | **79.88** |
| *Math* | | | | | | | | | | |
| GSM8K (5-shot) | 76.19 | 79.48 | 79.76 | 79.98 | **82.48** | 71.27 | 76.84 | 77.69 | 79.19 | **82.70** |
| MATH (4-shot) | 12.08 | 22.66 | 19.79 | 21.22 | **23.56** | 36.21 | 43.13 | 42.88 | 47.55 | **49.55** |
| *Code (pass@1)* | | | | | | | | | | |
| HumanEval | 57.93 | 67.24 | 60.98 | 65.61 | **69.05** | 72.66 | 80.49 | 74.39 | 78.05 | **82.27** |
| HumanEval+ | 43.29 | 46.36 | 43.59 | 45.76 | **48.08** | 55.85 | 61.83 | 59.03 | 60.49 | **63.05** |
| *Leaderboards* | | | | | | | | | | |
| Open LLM Leaderboard 1 | 62.85 | 66.30 | 65.93 | 66.79 | **68.50** | 64.76 | 66.73 | 66.87 | 68.29 | **69.28** |
| Open LLM Leaderboard 2 | 37.15 | 40.66 | 39.99 | 41.02 | **42.75** | 45.77 | 48.53 | 48.08 | 49.74 | **51.28** |
| *Overall* | 50.09 | 53.96 | 52.86 | 54.16 | **56.04** | 56.55 | 59.56 | 58.79 | 60.48 | **62.05** |

Table 20: DPO results for Llama-3.1-8B-TuluSFT and Qwen-2.5-7B-TuluSFT trained on ablated DPO mixtures UM-only-PF, UM-only-QF, and UM-No-PF, compared to TuluDPO on Open LLM Leaderboards (averaged) and code benchmarks. The overall average is across all benchmarks. Best scores are in **bold**.

| | Llama-3.1-8B-TuluSFT | | | | | Qwen-2.5-7B-TuluSFT | | | | |
| Benchmark | TuluDPO | UM-only-PF | UM-only-QF | UM-No-PF | UM-190k | TuluDPO | UM-only-PF | UM-only-QF | UM-No-PF | UM-190k |
|---|---|---|---|---|---|---|---|---|---|---|
| *Knowledge* | | | | | | | | | | |
| MMLU (5-shot) | 63.47 | 63.42 | 63.35 | 62.69 | **64.61** | 73.10 | 72.90 | 73.17 | 73.05 | **74.01** |
| MMLU-Pro (5-shot) | 28.98 | 29.58 | 28.79 | 28.55 | **30.96** | 43.48 | 42.56 | 43.43 | 43.52 | **44.65** |
| TruthfulQA (0-shot) | 56.78 | 57.43 | 58.44 | 56.80 | **63.32** | 54.46 | 55.68 | 54.13 | 55.20 | **58.00** |
| GPQA (0-shot) | 29.61 | 30.21 | 29.78 | 29.52 | **31.87** | 31.29 | 30.62 | 29.95 | 31.12 | **33.03** |
| *Reasoning* | | | | | | | | | | |
| ARC-C (25-shot) | 57.93 | 57.68 | 58.02 | 57.76 | **58.70** | 60.32 | 61.97 | 61.18 | 60.92 | **62.43** |
| BBH (3-shot) | 40.46 | 41.22 | 40.90 | 40.28 | **44.96** | 48.08 | 49.78 | 48.43 | 48.29 | **51.76** |
| MuSR (0-shot) | 41.93 | 40.08 | 42.06 | 41.87 | **44.02** | 47.16 | 44.12 | 46.43 | 47.30 | **48.82** |
| *Commonsense* | | | | | | | | | | |
| HellaSwag (10-shot) | 64.85 | 62.26 | 63.94 | 62.84 | 64.82 | 62.70 | 62.07 | 63.03 | 62.16 | **63.89** |
| WinoGrande (5-shot) | 75.30 | 74.03 | 76.56 | 75.72 | **77.06** | 72.96 | 70.61 | 71.11 | 72.19 | **74.64** |
| *Instruction Following* | | | | | | | | | | |
| IF-Eval (0-shot) | 80.35 | 77.61 | 79.74 | 79.93 | **81.13** | 78.04 | 76.55 | 77.70 | 75.39 | **79.88** |
| *Math* | | | | | | | | | | |
| GSM8K (5-shot) | 79.48 | 74.97 | 76.65 | 79.76 | **82.48** | 76.84 | 70.13 | 71.49 | 77.69 | **82.70** |
| MATH (4-shot) | 22.66 | 13.82 | 20.17 | 19.79 | **23.56** | 43.13 | 40.30 | 42.60 | 42.88 | **49.55** |
| *Code (pass@1)* | | | | | | | | | | |
| HumanEval | 67.24 | 59.05 | 64.07 | 60.98 | **69.05** | 80.49 | 73.66 | 73.78 | 74.39 | **82.27** |
| HumanEval+ | 46.36 | 43.07 | 43.61 | 43.59 | **48.08** | 61.83 | 57.89 | 58.05 | 59.03 | **63.05** |
| *Leaderboards* | | | | | | | | | | |
| Open LLM Leaderboard 1 | 66.30 | 64.96 | 66.16 | 65.93 | **68.50** | 66.73 | 65.56 | 65.68 | 66.87 | **69.28** |
| Open LLM Leaderboard 2 | 40.66 | 38.75 | 40.24 | 39.99 | **42.75** | 48.53 | 47.32 | 48.09 | 48.08 | **51.28** |
| *Overall* | 53.96 | 51.74 | 53.29 | 52.86 | **56.04** | 59.56 | 57.77 | 58.18 | 58.79 | **62.05** |

## F.5 ASSESSING THE ROLE OF INCOHERENT PREFERENCE ORDERS

As discussed in the main body, our independently trained reward model disagrees with the preference orderings in many of the existing DPO datasets we evaluate. There are several likely reasons for this mismatch. For instance, the use of GPT-4-generated scalar scores in UltraFeedback (Cui et al., 2024) does not guarantee consistent or reliable preference signals, and the authors explain why:

- **Ambiguous Scores**: GPT-4 frequently assigns identical scores to chosen/rejected responses as completions are near-ties (sourced from 17 LLMs).

- **Limited Human Agreement**: UltraFeedback reports only 59–68% human agreement, which is close to the 75% consistency we observe.

- **Reward Model Accuracy**: their own reward model achieves only 66–71% accuracy, confirming nontrivial noise in its preference signals.

We provide additional empirical evidence for this point: Tables 21 and 22 show that, for both Llama and Qwen models, filtering out misaligned preference pairs ("[Pref]" variants) consistently improves downstream performance on almost all benchmarks, especially on aggregate leaderboard metrics.

Table 21: DPO results for Llama-3.1-8B-TuluSFT trained on all preference datasets with (indicated by [Pref]) and without reward-based filtering. The overall average is across all benchmarks. Best scores are in **bold**.

| Benchmark | SFT | TuluDPO | TuluDPO [Pref] | UltraFB | UltraFB [Pref] | ORPO | ORPO [Pref] | HelpSteer | HelpSteer [Pref] | CodePref | CodePref [Pref] |
|---|---|---|---|---|---|---|---|---|---|---|---|
| *Knowledge* | | | | | | | | | | | |
| MMLU (5-shot) | 62.30 | 63.47 | 64.14 | 62.53 | 62.96 | 62.31 | 62.96 | 62.04 | 62.77 | 59.96 | 60.02 |
| MMLU-Pro (5-shot) | 28.08 | 28.98 | 28.96 | 28.41 | 28.06 | 27.90 | 27.98 | 27.43 | 27.31 | 27.53 | 27.69 |
| TruthfulQA (0-shot) | 46.84 | 56.78 | 61.31 | 50.26 | 54.77 | 52.26 | 56.88 | 47.43 | 48.22 | 56.15 | 57.35 |
| GPQA (0-shot) | 28.44 | 29.61 | 29.69 | 28.69 | 29.61 | 29.70 | 29.78 | 29.36 | 30.70 | 29.95 | 30.03 |
| *Reasoning* | | | | | | | | | | | |
| ARC-C (25-shot) | 54.95 | 57.93 | 57.40 | 56.91 | 57.08 | 56.91 | 56.23 | 54.52 | 54.61 | 45.05 | 45.54 |
| BBH (3-shot) | 38.59 | 40.46 | 43.93 | 39.91 | 41.61 | 41.80 | 42.13 | 38.15 | 39.48 | 36.07 | 36.49 |
| MuSR (0-shot) | 43.25 | 41.93 | 42.68 | 41.93 | 41.67 | 43.78 | 43.27 | 42.06 | 42.14 | 41.53 | 42.74 |
| *Commonsense* | | | | | | | | | | | |
| HellaSwag (10-shot) | 60.41 | 64.85 | 64.24 | 62.29 | 62.83 | 60.08 | 60.78 | 61.20 | 61.39 | 53.38 | 53.46 |
| WinoGrande (5-shot) | 76.40 | 75.30 | 76.19 | 75.53 | 76.40 | 74.27 | 75.69 | 76.16 | 76.32 | 69.93 | 70.75 |
| *Instruction Following* | | | | | | | | | | | |
| IF-Eval (0-shot) | 72.45 | 80.35 | 80.74 | 71.59 | 72.18 | 71.65 | 73.12 | 73.93 | 74.94 | 69.90 | 72.23 |
| *Math* | | | | | | | | | | | |
| GSM8K (5-shot) | 76.19 | 79.48 | 80.74 | 78.47 | 80.06 | 81.96 | 81.83 | 76.88 | 78.32 | 76.35 | 76.44 |
| MATH (4-shot) | 12.08 | 22.66 | 22.79 | 18.81 | 20.02 | 20.39 | 21.98 | 14.88 | 16.16 | 15.11 | 15.33 |
| *Code (pass@1)* | | | | | | | | | | | |
| HumanEval | 57.93 | 67.24 | 68.76 | 61.59 | 68.29 | 64.63 | 65.63 | 59.15 | 61.91 | 70.68 | 70.24 |
| HumanEval+ | 43.29 | 46.36 | 47.56 | 42.49 | 44.39 | 41.63 | 43.11 | 42.93 | 44.76 | 50.61 | 50.41 |
| *Leaderboards* | | | | | | | | | | | |
| Open LLM Leaderboard 1 | 62.85 | 66.30 | **67.34** | 64.33 | **65.68** | 64.63 | **65.73** | 63.04 | **63.60** | 60.14 | **60.59** |
| Open LLM Leaderboard 2 | 37.15 | 40.66 | **41.46** | 38.22 | **38.86** | 39.20 | **39.71** | 37.64 | **38.46** | 36.68 | **37.42** |
| *Overall* | 50.09 | 53.96 | **54.94** | 51.39 | **52.85** | 52.09 | **52.96** | 50.44 | **51.36** | 50.16 | **50.62** |

Table 22: DPO results for Qwen-2.5-7B-TuluSFT trained on all preference datasets with (indicated by [Pref]) and without reward-based filtering. The overall average is across all benchmarks. Best scores are in **bold**.

| Benchmark | SFT | TuluDPO | TuluDPO [Pref] | UltraFB | UltraFB [Pref] | ORPO | ORPO [Pref] | HelpSteer | HelpSteer [Pref] | CodePref | CodePref [Pref] |
|---|---|---|---|---|---|---|---|---|---|---|---|
| *Knowledge* | | | | | | | | | | | |
| MMLU (5-shot) | 72.41 | 73.10 | 73.28 | 73.32 | 73.47 | 73.12 | 73.36 | 72.38 | 73.38 | 72.82 | 72.85 |
| MMLU-Pro (5-shot) | 43.32 | 43.48 | 43.40 | 43.94 | 43.88 | 43.73 | 43.80 | 43.07 | 44.11 | 43.38 | 43.54 |
| TruthfulQA (0-shot) | 51.64 | 54.46 | 55.30 | 53.95 | 54.49 | 53.53 | 53.62 | 52.27 | 52.51 | 52.10 | 52.39 |
| GPQA (0-shot) | 30.96 | 31.29 | 31.51 | 31.04 | 31.12 | 30.61 | 30.89 | 30.05 | 31.12 | 30.37 | 31.35 |
| *Reasoning* | | | | | | | | | | | |
| ARC-C (25-shot) | 59.22 | 60.32 | 61.35 | 60.32 | 60.41 | 60.07 | 60.41 | 59.64 | 59.73 | 59.56 | 59.96 |
| BBH (3-shot) | 47.93 | 48.08 | 50.03 | 48.57 | 49.71 | 47.14 | 48.00 | 47.19 | 48.38 | 47.72 | 48.42 |
| MuSR (0-shot) | 48.15 | 47.16 | 47.71 | 47.75 | 47.92 | 47.88 | 48.09 | 48.41 | 48.85 | 46.30 | 47.09 |
| *Commonsense* | | | | | | | | | | | |
| HellaSwag (10-shot) | 60.85 | 62.70 | 62.64 | 61.80 | 61.84 | 61.07 | 61.40 | 61.12 | 61.22 | 61.22 | 62.27 |
| WinoGrande (5-shot) | 73.17 | 72.96 | 73.19 | 72.85 | 73.07 | 72.06 | 72.51 | 72.11 | 74.27 | 70.48 | 70.40 |
| *Instruction Following* | | | | | | | | | | | |
| IF-Eval (0-shot) | 68.06 | 78.04 | 78.30 | 75.74 | 76.06 | 73.02 | 74.35 | 70.16 | 72.80 | 71.05 | 73.19 |
| *Math* | | | | | | | | | | | |
| GSM8K (5-shot) | 71.27 | 76.84 | 77.70 | 74.93 | 75.54 | 77.33 | 78.18 | 71.87 | 72.86 | 72.24 | 73.61 |
| MATH (4-shot) | 36.21 | 43.13 | 49.38 | 39.19 | 41.06 | 41.92 | 42.84 | 37.72 | 39.79 | 36.34 | 39.04 |
| *Code (pass@1)* | | | | | | | | | | | |
| HumanEval | 72.66 | 80.49 | 80.88 | 76.05 | 78.66 | 78.66 | 78.89 | 76.27 | 79.27 | 84.54 | 84.13 |
| HumanEval+ | 55.85 | 61.83 | 61.98 | 58.01 | 59.27 | 59.10 | 60.98 | 56.02 | 61.59 | 65.49 | 65.27 |
| *Leaderboards* | | | | | | | | | | | |
| Open LLM Leaderboard 1 | 64.76 | 66.73 | **67.24** | 66.19 | **66.47** | 66.20 | **66.58** | 64.90 | **65.66** | 64.74 | **65.25** |
| Open LLM Leaderboard 2 | 45.77 | 48.53 | **50.06** | 47.70 | **48.29** | 47.38 | **48.00** | 46.10 | **47.51** | 45.86 | **47.10** |
| *Overall* | 56.55 | 59.56 | **60.48** | 58.39 | **59.04** | 58.52 | **59.09** | 57.02 | **58.56** | 58.11 | **58.82** |

## F.6 Efficiency Gains

To assess efficiency and training cost, we report the number of processed tokens (computed with each model's distinct tokenizer), estimates for training FLOPs, and total GPU hours (on an 8 x A100 GPU cluster) in Table 23 for DPO training of Llama-3.1-8B-TuluSFT, Instella-3B-SFT, SmolLM-2-1.7B-SFT, and OLMo-2-1B-SFT on TuluDPO and UltraMix (UM-190k), thereby covering four distinct model architectures and scales. We find that the reduction in dataset size translates approximately linearly into efficiency gains, such that a 30% reduction in data for UltraMix results in a proportionate reduction in the number of processed tokens, FLOPs, and total GPU hours. These additional experiments validate the efficiency of our curated UltraMix dataset.

Table 23: Efficiency comparison of DPO training for Llama-3.1-8B-TuluSFT, Instella-3B-SFT, SmolLM-2-1.7B-SFT, and OLMo-2-1B-SFT models on TuluDPO vs UltraMix (UM-190k). We report processed tokens (per tokenizer), estimated ExaFLOPs, and GPU hours (rounded to the nearest half hour, excluding the initial warmup phase).

| Efficiency Metric | Llama-3.1-8B-TuluSFT | | Instella-3B-SFT | | SmolLM-2-1.7B-SFT | | OLMo-2-1B-SFT | |
|---|---|---|---|---|---|---|---|---|
| | TuluDPO | UltraMix | TuluDPO | UltraMix | TuluDPO | UltraMix | TuluDPO | UltraMix |
| Tokens ($\downarrow$) | 215.6M | 153.8M | 220.9M | 157.2M | 224.7M | 159.7M | 217.1M | 154.7M |
| ExaFLOPs ($\downarrow$) | 10.4 | 7.4 | 4.1 | 2.9 | 2.44 | 1.73 | 1.93 | 1.37 |
| GPU Hours ($\downarrow$) | 11 | 8 | 7 | 5 | 5 | 3.5 | 3 | 2 |

## G    LIMITATIONS AND BROADER IMPACT

**Limitations.**    While our study provides a comprehensive and principled analysis of open-source DPO datasets, several limitations remain. First, our annotations rely on the Magpie framework with LLM-as-a-judge prompting to evaluate input quality, query difficulty, and task type, complemented by a reward-model-based preference score to validate chosen completions. Although this combination allows scalable and fine-grained inspection of preference pairs, the subjectivity inherent in LLM-based judgments and the biases of the underlying reward model may affect label accuracy. Future improvements in judge models or reward assessment may thus shift the results. Second, our focus is exclusively on DPO due to its popularity and the availability of large open-source corpora. Other alignment methods such as PPO, ORPO, and GRPO employ different data requirements, and a comparative study across methods would provide additional insights. Third, although we design general curation rules with percentile-based thresholds, we only explore a limited set of quantiles and balancing strategies due to computational constraints. A broader ablation study would help determine optimal thresholds across tasks and model scales. Finally, as *UltraMix* is derived from existing open datasets, it inevitably inherits their respective coverage gaps and biases.

**Broader Impact.**    By releasing detailed annotations, curated mixtures, and reproducible curation recipes, our work lowers the barrier to systematic research on preference optimization and promotes transparency in corresponding dataset design. Our annotations can be directly reused by researchers and practitioners to probe dataset quality, conduct controlled ablations, or build tailored preference mixtures for specialized domains such as math, code, or reasoning. *UltraMix*, our curated dataset, achieves stronger benchmark performance with 30% fewer samples than TuluDPO, offering both alignment improvements and compute efficiency gains in DPO training. While we apply our recipe to five prominent DPO corpora, the general approach, consisting of quality filtering, reward verification, and task balancing, can be applied to other datasets in principle. Nevertheless, we acknowledge that preference datasets, like any general-purpose alignment corpus, may carry dual-use risks: preference optimization can amplify biases, encode harmful behaviors, or be misapplied in sensitive domains. We therefore encourage responsible use and support future work on adversarial safety evaluations, fairness-aware preference curation, and multilingual extensions of DPO datasets.

**Contributions.**    This work presents the first systematic, sample-level comparison of widely used open-source DPO datasets. We evaluate eight models of different architectures and scales across 14 benchmarks covering instruction following, coding, math, and reasoning. Grounded in extensive Magpie-based annotations and preference reward validation, we provide a detailed dissection of TuluDPO, ORPO, UltraFeedback, HelpSteer, and Code-Preference-Pairs, revealing differences in composition, reward reliability, and task specialization. In particular, we show that pre-annotated reward scores do not reflect the assessment of specialized reward models, indicating bias and misalignment in existing DPO datasets. Leveraging these insights, we curate *UltraMix*, a reward-aligned high-quality DPO dataset that consistently outperforms the best-performing TuluDPO corpus while reducing compute requirements. Our methodology, which combines automated annotations, reward-based filtering, and task-aware balancing, offers a reusable framework for future dataset curation. By making both the annotations and *UltraMix* publicly available, we aim to establish a reproducible foundation for data-centric preference optimization research and accelerate progress in transparent and efficient LLM post-training.

