# OpenReview forum: "When Data is the Algorithm: A Systematic Study and Curation of Preference Optimization Datasets"
_ICLR.cc/2026/Conference — ICLR 2026 Poster_

### Official Review · Reviewer_nUbS · 2025-10-17

**Soundness:** 3
**Presentation:** 3
**Contribution:** 3
**Rating:** 4
**Confidence:** 4

**Summary:**

This paper presents a systematic comparison of five widely used DPO post-training datasets: TuluDPO, ORPO, UltraFeedback, HelpSteer, and Code-Preference-Pairs. The authors analyze each dataset through task category, input quality, difficulty, and preference reward annotations, and propose a new DPO mixture, UltraMix, which is 30% smaller than TuluDPO but outperforms it on various benchmarks. The paper highlights the importance of systematic curation, task diversity, and the impact of data quality on the performance of DPO-based models. It provides a detailed evaluation across multiple models, demonstrating that the optimal composition of data mixtures is task-sensitive and requires careful consideration of quality and diversity, particularly for instruction-following tasks.

**Strengths:**

The paper’s detailed dataset analysis and creation of UltraMix reflect a high level of thoroughness. The inclusion of annotations for quality, difficulty, and preference rewards provides a nuanced and comprehensive look at the DPO process.

**Weaknesses:**

1. Comparison of different datasets: While the comparison of different datasets in Table 1 is useful, it does not account for size differences between datasets. For example, TuluDPO has 273k pairs, while ORPO only has 44k.
2. Preference Signal Accuracy: The authors mention that current preference signals have only about 70% accuracy, especially in datasets like UltraFeedback. Since this dataset is annotated by GPT-4o, it would be expected to have a higher level of accuracy. The paper should clarify why the preference signal in this dataset is weaker than anticipated.
3. Model Completion Correlation: The paper claims that model completion is correlated with prompt quality (line 289), but similar results have already been discussed in previous work, such as in [1].
4. Some related works have already discussed what contributes to the quality of a preference dataset; you can consider citing and comparing with them. [2,3,4]
5. Empirical Focus: While the study is solid, it is highly empirical. A more theoretical analysis could help explain the findings and connect them to broader research questions. Insights into the theoretical mechanisms behind dataset curation and the resulting performance gains would strengthen the paper's contributions.

[1] Unpacking DPO and PPO: Disentangling Best Practices for Learning from Preference Feedback
[2] AIR: A Systematic Analysis of Annotations, Instructions, and Response Pairs in Preference Dataset
[3] R.I.P.: Better Models by Survival of the Fittest Prompts
[4] Direct preference optimization with an offset

**Questions:**

See above.

---

> ### Author Response · Authors · 2025-11-19
>
> We appreciate your recognition of our thorough study and address your points below.
>
> ## 1. Different Dataset Sizes
> - Our goal is to provide a **side-by-side performance comparison** of DPO datasets to characterize their strengths and weaknesses across benchmarks, not to normalize for size.
> - Most fall within the typical range of 10k–60k, with TuluDPO as the only larger outlier.
> - Such comparisons are valuable to **understand trade-offs between quantity and quality**.
> - Our UltraMix dataset shows that **quality dominates quantity** and that principled curation matters more than raw scale.
> - However, we agree that noting sizes in the main text will improve clarity!
>
> ## 2. Ultrafeedback expected to have higher accuracy
> - As noted in App. B, UltraFeedback’s GPT-4–based scalar scores do not guarantee reliable preference signals.
> - The original UltraFeedback paper explains why:
> 1. **Ambiguous Scores**: GPT-4 frequently assigns identical scores to chosen/rejected responses as completions are near-ties (sourced from 17 LLMs).
> 2. **Limited Human Agreement**: UltraFeedback reports only 59–68% human agreement, which is close to the ~75% consistency we observe.
> 3. **Reward Model Accuracy**: their own reward model achieves only 66–71% accuracy, confirming nontrivial noise in its preference signals.
>
> Other works such as AIR [2] independently confirm these findings, citing inherent ambiguity and limitations of scalar scoring.
>
> We also provide **empirical evidence for this claim**: Tables below show for Llama/Qwen that filtering out misaligned preference pairs ([Pref]) consistently improves performance.
>
> ### a) Llama
> |Model|LB1|LB2|Avg|
> |--|--|--|--|
> |TuluDPO|66.30|40.66|53.96|
> |TuluDPO [Pref]|67.34|41.46|**54.94**|
> |UltraFeedback|64.33|38.22|51.39|
> |UltraFeedback [Pref]|65.68|38.86|**52.85**|
> |ORPO|64.63|39.20|52.09|
> |ORPO [Pref]|65.73|39.71|**52.96**|
> |HelpSteer|63.04|37.64|50.44|
> |HelpSteer [Pref]|63.60|38.46|**51.36**|
>
> ### b) Qwen
> |Model|LB1|LB2|Avg|
> |--|--|--|--|
> |TuluDPO|66.73|48.53|59.56|
> |TuluDPO [Pref]|67.24|50.06|**60.48**|
> |UltraFeedback|66.19|47.70|58.39|
> |UltraFeedback [Pref]|66.47|48.29|**59.04**|
> |ORPO|66.20|47.38|58.52|
> |ORPO [Pref]|66.58|48.00|**59.09**|
> |HelpSteer|64.90|46.10|57.02|
> |HelpSteer [Pref]|65.66|47.51|**58.56**|
>
> ## 3. Model Completion Correlation
> - The referenced work [1] indeed shows that **prompt distribution** can influence preference training.
> - Our contribution is complementary: we provide **dataset-level evidence that prompt quality** (ambiguity, underspecification, poor writing) impacts preference alignment and correlates with lower rewards, not just distributional properties.
> - We will clarify this distinction in the updated version!
>
> ## 4. Comparison with Related Work
>
> ### a) [2] AIR Paper
> - AIR **creates a new preference dataset from scratch** by extending the UltraFeedback pipeline with judge annotations when sourcing from LLMs.
> - We focus on **post-hoc curation** of existing DPO datasets using reward and quality signals to identify misaligned pairs and improve dataset composition.
>
> ### b) [3] RIP Paper
> - RIP conducts micro-level reward model filtering based on response length and reward gaps.
> - We propose a curation recipe that combines multiple filters for preference coherence, input quality, difficulty, and task-awareness.
> - Ablations in App. F.4 show that **combining these signals yields better performance** than reward-only filtering, addressing the limitations in RIP.
>
> ### c) [4] ODPO Paper
> - ODPO is an **algorithmic contribution**, modifying the DPO loss to incorporate graded reward differences.
> - Our work is **data-centric**, focusing on curating and filtering DPO datasets.
>
> We will add these comparisons in the updated version!
>
> ## 5. Theoretical Explanation
> - We agree that a theoretical perspective can help clarify why high-quality datasets matter for DPO.
> - In our updated version, we will include a short section summarizing key theoretical mechanisms based on [5,6]:
>
> 1. DPO updates are driven by the log-likelihood margin between chosen and rejected pairs:
> $$m_\theta(x) = \log\pi_\theta(y_c\mid x)-\log\pi_\theta(y_r\mid x)$$
> 2. Prior work shows that DPO implicitly learns a reward model and that the signal-to-noise ratio of the margin governs optimization stability.
> 3. Thus, well-specified prompts increase expected margins and decrease variance, leading to improved performance.
>
> - However, **empirical studies are critical** as real-world performance depends on large-scale data behavior.
> - Many influential works [7,8] are **primarily empirical** and we contribute by providing a **systematic comparison of DPO datasets and a principled curation recipe** to improve dataset quality for alignment in practice.
>
> [5] Direct Preference Optimization: Your Language Model Is Secretly a Reward Model.
> [6] A General Theoretical Paradigm to Understand Learning from Human Preferences.
> [7] Tulu 3: Pushing Frontiers in Open Language Model Post-Training
> [8] SmolLM2: When Smol Goes Big

---

> > ### Comment · Reviewer_nUbS · 2025-11-26
> >
> > Thanks for your response.
> >
> > 1. Different Dataset Sizes
> >
> > It's reasonable that section 3 is not the main focus.
> >
> > 2. Ultrafeedback expected to have higher accuracy
> >
> > Authors' responses have addressed this concern.
> >
> > 3. Model Completion Correlation
> >
> > I am confused about the complementary dataset-level evidence. In sec 4.2, favoring hard prompts and high input quality prompts are established consensus, although compared between entire datasets. Similar results have already been discussed in previous work. The similarities and differences should be discussed more.
> >
> > 4. Comparison with Related Work
> >
> > Creating a new preference dataset from scratch and post-hoc curation both examine the quality of preference datasets, which may not be seen as a difference. The findings of your paper are similar: selecting a high input quality prompt, maximizing the difference between the chosen reward and rejected reward, etc. This is my major concern.
> >
> > I maintain my score.

---

> ### Author Response · Authors · 2025-11-27
>
> Thank you for opening up the discussion! We are glad that our response addressed two of your concerns. We respond to the remaining two.
>
> ## 1. How we are Different than Unpacking DPO/PPO [1], AIR [2], and RIP [3]
> - Fundamentally, how to **quantify data quality** is non-trivial and how prior works define "what contributes to the quality of a preference dataset" differs from our quantification
> - We highlight these distinctions and provide **new experimental results** corroborating that UltraMix's **non-trivial combination of signals** (not just "high quality") is required for best performance
>
> ### a) Distinction from Unpacking DPO/PPO [1]
> - [1] compares datasets **only monolithically** and makes two conclusions: a) synthetic datasets outperform human-annotated ones, b) datasets created using multi-aspect annotations outperform those with only aggregated ones
> - However, [1] **does NOT** look into per-sample quality **NOR** does it consider post-hoc curation
> - Our work goes beyond monolithic comparisons and performs a **per-sample diagnosis**, i.e., we look inside datasets and identify **which concrete samples are subpar**
> - We leverage sample-level annotations to propose a **principled curation recipe**
> - We do the **first large-scale quantification** of prompt quality and reward coherence for popular datasets
> - We show that even "high-quality" datasets (TuluDPO) contain **20–30% of samples with contradicting preference order**.
>
> ### b) Distinction from AIR [2]
> You mention that creating data from scratch (AIR) vs. post-hoc curation (Ours) might not be a significant difference. We argue this distinction is **crucial**:
> - AIR extends UltraFeedback's pipeline (*which we know from our initial response is flawed*) to create new dataset based on moderate reward margins and high absolute scores
> -  In contrast, we designed our workflow (annotations+recipe) **specifically for post-hoc optimization**
> - Thus, AIR **cannot be applied retroactively** to existing datasets without reannotating everything, while UltraMix can
>
> Also, AIR measures prompt quality as follows:
> 1. Generate multiple responses per prompt using diverse LLMs
> 2. Give each one a numerical score (LLM-based)
> 3. Determine quality as variance of scores
>
> This approach **is only possible when creating datasets from scratch** and **compute cost is huge** as it involves many LLMs!
>
> Our **post-hoc curation setup** quantifies prompt quality via systematic annotations for *clarity/reward coherence/etc.* without multi-LLM generation and **is thus scalable**!
>
> ### c) Distinction from RIP [3]
> You noted we maximize reward difference and thus are similar to RIP:
> - However, our recipe **does not maximize reward difference**, instead it checks reward coherence (*is the chosen answer really better?*)
> - Using **reward coherence** as a quality metric is a **novel aspect of our recipe**
> - Further, while RIP focuses on reward gaps/response length, we show that **relying on reward signals alone is insufficient**
> - Our recipe proposes to balance prompt, reward, and task filtering which is a **novel aproach**
>
> ## 2. New Experiments
>
> To strengthen our contributions, we provide 2 additional results:
>
> ### a) Table 1: Comparison to RIP-Style filtering
> - We present RIP-only (filtered on reward difference 75th %ile), RIP+UltraMix-Quality-Filter and UltraMix
> - **RIP-only underperforms** even if applying UltraMix's quality filter on top
> - This implies that many pairs have similar rewards and discarding them based on reward gap decreases performance
> - This shows that **preference order is the better signal**
> - Our combination of quality+reward coherence is a **distinct and superior contribution**
>
> ### b) Tables 2/3 (Llama/Qwen): Recipe Ablations
> - we assess the individual contribution of filtering steps: **[P] preference-reward filtering only**, **[Q] input-quality filtering only**, **[Q+D] input-quality+difficulty filtering**, compared to the **full UltraMix recipe**
> - Results show that no single step outperforms the TuluDPO baseline
> - [Q] and [Q+D] outperform [P], showing preference filtering is important but not the main driver
> - Best performance is only achieved when every step is combined
>
> The ablations show that UltraMix's effectiveness stems from the **combination of signals**, not only quality or reward filtering. **This is a new insight!**
>
> ## 3. Why our Contributions are Novel
>
> 1. We provide **the first large-scale DPO comparison** (5 datasets×14 benchmarks×8 models) which has not been done before
> 2. We design a **systematic, scalable, and generalizable post-hoc curation recipe** that is based on a **non-trivial combination of signals** (not just high quality) and outperforms baselines/related methods (RIP/AIR-style)
> 3. Reward coherence+task awareness **are novel signals**
> 4. We will **open-source annotated datasets** (UltraMix+5 source datasets) and our **annotation pipeline**
>
> **We urge you to reconsider your score and let us know if you still have concerns!**

---

> ### Author Response · Authors · 2025-11-27
>
> ### Table 1: Comparison to RIP-Style filtering
> | Model                                   | MMLU  | TRUTHFULQA | ARC-C | GSM8K | HELLASWAG | WINOGRANDE | MMLU_PRO | BBH   | GPQA  | MUSR  | IFEVAL | MATH  | HUMAN_EVAL | HUMAN_EVAL+ | LB1   | LB2   | Avg   |
> |-----------------------------------------|-------|------------|-------|-------|-----------|------------|----------|-------|-------|-------|--------|--------|-------------|--------------|-------|-------|-------|
> |Llama Model|
> | **Llama (RIP-only)**                    | 62.93 | 61.69 | 57.83 | 77.27 | 63.35 | 75.17 | 28.99 | 43.93 | 30.12 | 42.42 | 79.31 | 21.53 | 65.71 | 45.77 | 66.37 | 41.05 | 54.00 |
> | **Llama (RIP+UltraMix-Quality-Filter)** | 63.27 | 61.45 | 57.63 | 79.98 | 63.43 | 74.98 | 28.50 | 44.68 | 29.94 | 42.43 | 79.38 | 21.22 | 65.61 | 45.76 | 66.79 | 41.02 | 54.16 |
> | **Llama (UltraMix)**                    | **64.61** | **63.32** | **58.70** | **82.48** | **64.82** | **77.06** | **30.96** | **44.96** | **31.87** | **44.02** | **81.13** | **23.56** | **69.05** | **48.08** | **68.50** | **42.75** | **56.04** |
> |Qwen Model|
> | **Qwen (RIP-only)**                     | 73.10 | 54.47 | 60.33 | 76.85 | 62.67 | 72.98 | 43.48 | 48.08 | 31.28 | 47.19 | 78.03 | 43.13 | 80.49 | 61.83 | 66.73 | 48.53 | 59.56 |
> | **Qwen (RIP+UltraMix-Quality-Filter)**  | 73.53 | 57.60 | 62.12 | 79.19 | 63.59 | 73.69 | 43.16 | 50.96 | 31.63 | 47.87 | 77.28 | 47.55 | 78.05 | 60.49 | 68.29 | 49.74 | 60.48 |
> | **Qwen (UltraMix)**                     | **74.01** | **58.00** | **62.43** | **82.70** | **63.89** | **74.64** | **44.65** | **51.76** | **33.03** | **48.82** | **79.88** | **49.55** | **82.27** | **63.05** | **69.28** | **51.28** | **62.05** |
>
> ### Tables 2/3 (Llama/Qwen): Recipe Ablations
>
> ### a) Llama
> |Model|LB1|LB2|Avg|
> |--|--|--|--|
> |Llama-SFT  (Starting Model)|62.85|37.15|50.09|
> |Llama-DPO (TuluDPO, Baseline)|66.30|40.66|53.96|
> |Llama-DPO (UltraMix) [P]|64.96|38.75|51.74|
> |Llama-DPO (UltraMix) [Q[|66.16|40.24|53.29|
> |Llama-DPO (UltraMix) [Q+D]|65.93|39.99|52.86|
> |**Llama-DPO (UltraMix)** |**68.50**|**42.75**|**56.04**|
>
> ### b) Qwen
> |Model|LB1|LB2|Avg|
> |--|--|--|--|
> |Qwen-SFT (Starting Model)|64.76|45.77|56.55|
> |Qwen-DPO (TuluDPO, Baseline)|66.73|48.53|59.56|
> |Qwen-DPO (UltraMix) [P]|65.56|47.32|57.77|
> |Qwen-DPO (UltraMix) [Q]|65.68|48.09|58.18|
> |Qwen-DPO (UltraMix) [Q+D]|66.87|48.08|58.79|
> |**Qwen-DPO (UltraMix)**|**69.28**|**51.28**|**62.05**|
>
> **We will add all tables in the updated version of the paper!**

---

### Official Review · Reviewer_JCcE · 2025-10-19

**Soundness:** 2
**Presentation:** 3
**Contribution:** 2
**Rating:** 4
**Confidence:** 4

**Summary:**

This paper presents an empirical study of existing open-source DPO datasets. The authors analyze benchmarks resulting from fine-tuning models across various public datasets, and conduct sample-level analysis using Magpie. Overall, the study offers some useful insights for developers applying DPO to train models, though some experiments may not be convincing enough. Additionally, the methodologies employed largely rely on existing pipelines or models, resulting in limited technical contribution.

**Strengths:**

- This paper presents a thorough and extensive analysis of five open-source DPO datasets, examining aspects such as task type and prompt quality. The study offers valuable insights—for instance, highlighting that poorly written instructions can lead to subpar preference alignment.

- The experimental evaluation is comprehensive, employing different benchmarks to assess model performance. Furthermore, the authors validate their findings using different models, enhancing the reliability of the results.

**Weaknesses:**

- Unclear Motivation. The claim that "quality annotations are mostly missing" is not entirely accurate, as widely-used datasets like UltraFeedback and HelpSteer do provide fine-grained, human-annotated preference scores. Improving these details will help readers understand the contribution of this paper better.

- The current taxonomy requires refinement. Categorizing UltraFeedback and HelpSteer (lines 100-101) as "instruction-following" datasets is imprecise, as they are primarily preference datasets designed to enhance model helpfulness, not strictly instruction adherence.

- The benchmarking results are not fully convincing. They state that they "fix the training hyperparameters for each model," but optimal hyperparameters would vary across datasets. There are also factors (e.g., dataset size) that could significantly impact model performance.

- The analysis employing the MAGPIE pipeline, while somewhat useful for understanding type distribution, prompt quality, and difficulty across datasets, lacks methodological novelty, as it relies on existing tools and reward models.

- The heuristic selection of thresholds (e.g., 25th and 80th percentiles) appears arbitrary. Since this is an empirical study, a more principled or experimentally justified approach is necessary.

- Figure 3 is not convincing enough. Reward models are known to have biases and may not align perfectly with human judgment, while datasets like Helpsteer are annotated by human and should be more reliable.

**Questions:**

See weaknessness.

---

> ### Author Response · Authors · 2025-11-19
>
> Thank you for your feedback! Our responses follow below.
>
> ## 1. Clarification on Quality Annotations
> - Our statement refers to prompt quality (clarity, specificity, ambiguity) and related metadata (difficulty, category, etc.), which are essential for systematic curation but largely absent in DPO datasets.
> - As noted in App. B, UltraFeedback and HelpSteer do provide annotations, but they do not capture prompt quality or guarantee coherent preference pairs.
> - We fill this gap by introducing a curation recipe that adds these missing signals and validates preference pair consistency.
>
> ## 2. Dataset Taxonomy
> - While UltraFeedback and HelpSteer are not instruction-tuning datasets in the SFT sense, their prompts are **instruction-like**, hence our original grouping.
> - We agree that this taxonomy may be imprecise and will instead use **helpfulness-oriented** in the revised version.
>
> ## 3. Hyperparameters are Dataset-Dependent
> - **Fixing hyperparameters per model across all datasets** follows standard practice in dataset-quality studies [1–3], where controlled comparisons are required to isolate the effect of the data.
> - The prior work highlights that varying hyperparameters introduces **methodological heterogeneity**, making it difficult to attribute performance differences to the dataset itself.
> - By using the hyperparameters recommended in the original model papers (App. F.1), we ensure that performance changes reflect dataset composition and curation, not optimization settings.
>
> We will clarify this rationale in the updated version!
>
> ## 4. Using Magpie lacks methodological novelty
> - Our goal is a **replicable, data-centric study**, not introducing a new annotation framework.
> - While we use Magpie’s LLM-as-a-Judge pipeline, we do not claim novelty here.
> - Any similar pipeline could replace it **without affecting our methodology**.
> - Instead, our key novelty is the **systematic analysis and detailed curation recipe** that we build on top of annotations, leading to our **UltraMix dataset**.
>
> ## 5. Heuristic Thresholds
> - As noted in **App. G**, we only explore a limited set of quantiles due to compute constraints.
> - Nonetheless, **ablations in Sec. 5.1–5.2 show clear trends**: lowering thresholds and adding more high-reward samples consistently improves performance.
> - The progression UltraMix-170k → 187k → 190k yields **monotonic gains**, i.e. thresholds meaningfully shape dataset quality rather than acting arbitrarily.
> - However, we acknowledge that a grid search may improve benefits and leave it as future work.
>
> ## 6. Figure 3 + Reward models may be biased
> - Figure 3 only visualizes where original preferences agree with the reward model.
> - We explain why FsfairX is appropriate and why human-derived scores may still be ambiguous:
>
> ### a) Justifying FsfairX
> - Good reward models are necessary to detect misaligned preferences.
> - We use FsfairX only for preference rewards due to its strong human alignment.
> - To avoid reliance on one model, we compare FsfairX with Skywork-Reward in App. C.2 and find similar reward distributions, showing our **conclusions do not depend on one specific model**.
> - Further, the consistent performance gains show that **UltraMix’s improvements come from our curation strategy**, not from the choice of reward model.
>
> ### b) Why HelpSteer and UltraFeedback Annotations Are Not Always Reliable
> While both datasets include annotations, they do not guarantee reliable preference directions:
> 1. **UltraFeedback**: many chosen/rejected responses receive identical GPT-4 ratings due to near-tie completions, resulting in AI–human agreement of only 59–68% [4].
> 2. **HelpSteer**: provides only few scalar labels for helpfulness/correctness/coherence/complexity/verbosity, requiring preference to be inferred from aggregated scores, which can introduce uncertainty. This means that the preference order is not directly labeled by annotators.
>
> We provide empirical evidence for our claim: Tables below (Llama/Qwen on HelpSteer/UltraFeedback) show that filtering out misaligned preference pairs via our reward model ([Pref]) consistently improves performance.
>
> ## a) Llama
> | Model|LB1|LB2|Avg|
> |--|--|--|--|
> |UltraFeedback|64.33|38.22|51.39|
> |UltraFeedback[Pref]|65.68|38.86|52.85|
> |HelpSteer |63.04|37.64|50.44|
> |HelpSteer [Pref]|63.60|38.46|51.36|
>
> ## b) Qwen
> |Model|LB1|LB2|Avg|
> |--|--|--|--|
> |UltraFeedback|66.19|47.70|58.39|
> |UltraFeedback [Pref]|66.47|48.29|59.04|
> |HelpSteer| 64.90|46.10|57.02|
> |HelpSteer [Pref]|65.66|47.51|58.56|
>
> These results show that despite human annotations in Helpsteer, optimal preference may be assessed differently such that merely aggregating scores as done in most works is suboptimal.
>
> [1] The FineWeb Datasets: Decanting the Web for the Finest Text Data at Scale
> [2] Tulu 3: Pushing Frontiers in Open Language Model Post-Training
> [3] Unpacking DPO and PPO: Disentangling Best Practices for Learning from Preference Feedback
> [4] UltraFeedback: Boosting Language Models with Scaled AI Feedback

---

> > ### Comment · Reviewer_JCcE · 2025-11-28
> >
> > Sorry for my late reply. I have carefully read your rebuttal (including the global response), and I am overall satisfied. I will consider raising my score accordingly. However, I still have two things to further discuss with authors:
> > - (Major) Could you further explain why the same hyperparameters were used across previous works and yours, as I do not know how previous works study the effects of different datasets? Are there any assumption such as similar dataset sizes? Datasets like HelpSteer are relatively small, I remain somewhat puzzled by the use of identical hyperparameters when training on datasets of significantly different scales. I would appreciate further discussion on this point.
> > - (Optional) Would it be possible to leverage your findings to perform data filtering and conduct a fair comparison on the latest dataset (e.g., Helpsteer3) with baseline methods like RIP (as mentioned in your global response)?  Given my late reply, I understand that you may not have enough time to incorporate such additional experiments. However, it would be highly valuable if these results could be included.

---

> > > ### Author Response · Authors · 2025-12-01
> > >
> > > **Thank you for raising your score and further engaging in the discussion! We are happy to clarify further!**
> > >
> > > ## *Rationale for Fixed Hyperparameters in Post-Training*
> > > We followed the standard methodology established in state-of-the-art post-training literature (e.g., Tulu 3 [1] and Unpacking DPO/PPO [2]) to ensure a controlled *ceteris paribus* comparison.
> > >
> > > **We would like to address 3 points:**
> > >
> > > ### *1) Isolating Data Quality as the Variable*
> > > - In post-training, when the goal is to **evaluate the intrinsic quality of different dataset mixtures**, it is standard practice to *fix the optimizer settings (learning rate, beta values) and the number of epochs*, which we do in our work (see App. F.1).
> > > - **If we optimized hyperparameters individually for each dataset instead**, it would be *impossible to disentangle whether performance gains stem from better data or better hyperparameter tuning*.
> > > - Ref. [2] further explicitly notes that **fixing these parameters is necessary** to *systematically investigate the impact of data composition*, which we explicitly set as a goal of our work.
> > >
> > > Many works compare post-training datasets this way:
> > > 1. **Tulu 3 [1]** compares their own SFT/DPO datasets with various other mixes while *keeping the training recipe (LR, betas) constant* to determine which mix is best.
> > > 2. **Unpacking DPO/PPO [2]** compare 14 datasets, including UltraFeedback, SHP, and HH-RLHF, which have vastly different sizes (60k to 160k) but *keep the learning rate and betas fixed* to isolate the data component of their pipeline.
> > > 3. Similarly, **Magpie [3]** and **Zephyr [4]** follow this protocol when benchmarking their synthetic data against *differently-sized public datasets* like ShareGPT or UltraFeedback.
> > >
> > > Thus, we **remain consistent with prior work** and reproduce the *standard methodology* to compare post-training datasets.
> > >
> > > ### *2) Handling Varying Dataset Sizes via Epoch-based Training*
> > > - You are correct that **data sizes vary throughout our analysis**.
> > > - However, most datasets in our study (HelpSteer, ORPO, UltraFeedback, CodePrefs) fall within a **comparable range (10k–60k samples)** with only TuluDPO being a larger outlier.
> > > - **This is not uncommon**, and as mentioned above, *the same was done in Unpacking DPO/PPO [2]* for data sizes between 60k-160k.
> > >
> > > To ensure fairness, we follow the common approach in [1, 2] and **fix the number of epochs for each model**, rather than the number of update steps:
> > > - This ensures that the *model sees every sample the same number of times*, regardless of the number of samples the dataset has.
> > > - This *differs from pre-training* where token counts are fixed (e.g. FineWeb [6], DCLM [7]). Instead, in *post-training alignment*, the standard is to *allow the model to fully converge on the preference distribution of the dataset*, which requires epoch-based constraints.
> > >
> > > ### *3) Theoretical Justification*
> > > In general, DPO optimizes the *expected loss over the data distribution* (see App. A.3 and [5]), i.e. $\mathbb{E}_{(x,\, y\_w,\, y\_l)\sim \mathcal{D}} [\mathcal{L}\_{\mathrm{DPO}}]$
> > > - **Why LR is constant**: The optimal LR is determined by the Lipschitz constant of the gradients (the "sharpness" of the loss landscape), which is a *property of the model architecture and the task (preference alignment)*, not the dataset size $N$, explaining why this can be done for smaller datasets.
> > > - **Why Epochs are fixed**: Since batch size is fixed, the variance of the gradient estimator is constant across datasets. By fixing epochs (traversing $\mathcal{D}$ exactly once or $k$ times), we ensure the optimizer takes a number of steps *proportional to the information content, avoiding overfitting on small datasets* (which would happen if we fixed steps and looped over the small data repeatedly).
> > >
> > > Therefore our approach is also *theoretically sound*.
> > >
> > > ### **Our empirical results further back this approach**
> > > - Despite our curated UltraMix being 30% smaller, it *outperforms the larger TuluDPO (which has more gradient updates per epoch) on all benchmarks*, showing that the gains are not simply a matter of scale.
> > > - In our paper and the additional results from this rebuttal, we specifically show that the reason herefore is the application of our *principled data curation recipe*, yielding a better mixture.
> > >
> > > This confirms that our **fixed hyperparameter settings did not unfairly penalize the smaller datasets** and instead highlights that **data quality dominates data scale in DPO training**.
> > >
> > > [1] Tulu 3: Pushing Frontiers in Open Language Model Post-Training
> > > [2] Unpacking DPO and PPO: Disentangling Best Practices for Learning from Preference Feedback
> > > [3] Magpie: Alignment Data Synthesis from Scratch
> > > [4] Zephyr: Direct Distillation of LM Alignment
> > > [5] Direct Preference Optimization: Your Language Model is Secretly a Reward Model
> > > [6] The FineWeb Datasets: Decanting the Web for the Finest Text Data at Scale
> > > [7] DataComp-LM: In search of the next generation of training sets for language models

---

> > > > ### Author Response · Authors · 2025-12-01
> > > >
> > > > ## *(Optional) New Experiments on HelpSteer3 & Comparison with RIP*
> > > >
> > > > **Although suggested as optional**, we agree that evaluating our findings on the very latest datasets like HelpSteer3 strengthens our contributions even further.
> > > >
> > > > **Despite the short timeframe**, we have successfully annotated the single-turn HelpSteer3 dataset using our DPO-adapted Magpie pipeline and conducted the suggested comparisons with RIP-style filtering for both Llama and Qwen.
> > > >
> > > > *We compare 3 configurations*:
> > > > 1. **HelpSteer3 (Baseline)**.
> > > > 2. **HelpSteer3 [RIP]**: Applying *RIP-style filtering* (keeping only samples with high reward gaps, $>$60th percentile).
> > > > 3. **HelpSteer3 [Pref]** (Ours): Applying our *preference coherence filtering* (keeping only samples where chosen reward > rejected reward).
> > > >
> > > > We provide the averaged results across all 14 leaderboard benchmarks (LB1 and LB2) in below Tables:
> > > >
> > > > ### a) Llama
> > > > | Model                       | LB1   | LB2   | Avg   | Δ      |
> > > > |-----------------------------|-------|-------|-------|--------|
> > > > | Llama-HelpSteer3 (Base)     | 63.46 | 39.21 | 51.53 | –      |
> > > > | Llama-HelpSteer3 [RIP]      | 63.22 | 37.70 | 50.45 | -1.08  |
> > > > | **Llama-HelpSteer3 [Pref]** | **63.98** | **39.61** | **52.25** | **+0.72** |
> > > >
> > > > ### b) Qwen
> > > > | Model                       | LB1   | LB2   | Avg   | Δ      |
> > > > |-----------------------------|-------|-------|-------|--------|
> > > > | Qwen-HelpSteer3 (Base)      | 65.74 | 47.44 | 58.46 | –      |
> > > > | Qwen-HelpSteer3 [RIP]       | 65.42 | 46.59 | 57.79 | -0.67  |
> > > > | **Qwen-HelpSteer3 [Pref]**  | **66.02** | **47.73** | **58.83** | **+0.37** |
> > > >
> > > > *The results show*:
> > > > - **Preference Coherence is Superior**: Consistent with our global response and App. B.2, *filtering for preference coherence ([Pref]) yields superior results*.
> > > > - **RIP-style Filtering is Insufficient**: Merely selecting for large reward gaps (RIP) actually hurts performance (-1.08% on Llama) on this dataset. This suggests that *high reward gaps do not guarantee high-quality training signals if the underlying preference direction is noisy or if the data distribution is skewed*.
> > > > - **Generalizability**: These findings confirm that *our data curation insights hold true* even on the newest datasets like HelpSteer3.
> > > >
> > > > We will **open-source our annotated HelpSteer3 dataset** along with our other contributions to enable the community to build upon these results!

---

### Official Review · Reviewer_yvk1 · 2025-10-31

**Soundness:** 4
**Presentation:** 3
**Contribution:** 3
**Rating:** 6
**Confidence:** 4

**Summary:**

This paper conducts a comparative study on common DPO preference datasets (Ultrafeedback, Helpsteer, TuluDPO…) using the Magpie framework. Using their findings, they construct a new mixture UltraMix, which draws high quality data from all 5 datasets with coherent preferences. Training on UltraMix consistently improves performance across diverse downstream tasks.

**Strengths:**

1. Data Engineering for post-training is often overlooked but important. The paper presents a nice study and comparison of how to curate data for DPO and demonstrates that their recipe indeed improves performance.

2. The paper is well-written, and clear. Comprehensive analysis and experiments. Solid results.

3. It would be a good contribution if the authors can release their UltraMix dataset for the community to use and study.

**Weaknesses:**

1. My main concern is about the novelty of this paper. The main takeaway of this paper is data filtering is important, we need to select high quality and difficult data. Such findings are not new to the field and I don't know how much people would learn from reading this paper. The resulting dataset (UltraMix) itself might be more useful.

2. The quality filtering is conducted by an "independent reward model", in this case, FsfairX. I don't know how much of the gains comes from using this particular reward model v.s. using the same reward model to filter examples from all datasets. In other words, the gains from UltraMix might simply originate from FsfairX is a really really good reward model, which undermines the strength of this paper.

3. The authors' recipe has many "ingredients" in it: data quality filtering, difficulty filtering, consistency filtering ... It is hard to tell which part of the recipe makes it good. It could be because of some of them, it also could be because each one individually is not enough but together they are good. There are some missing ablations here.

**Questions:**

N/A

---

> ### Author Response · Authors · 2025-11-18
>
> Thank you for your feedback! We are pleased that you recognize our comprehensive analysis and data curation recipe as solid contributions. We hope to address your suggestions below!
>
> ## 1. Releasing UltraMix to the Public
> - We will **open-source our annotated UltraMix dataset and all intermediate variants** (UltraMix-170k, UltraMix-187k, UltraMix-190k) on HuggingFace to enable full reproducibility.
> - We will also release our **annotated versions of TuluDPO, Ultrafeedback, ORPO, Helpsteer, and Codepreferences**.
> - Further, we will release our **DPO-adapted Magpie annotation pipeline**.
>
> We believe this will contribute to the open-source community and help practitioners curate optimal DPO mixtures in the future!
>
> ## 2. Novelty could be better positioned
> - We empirically verify that “high quality data implies good models” but do not claim it as a new observation.
> - Instead, there are **two main contributions** that are of interest to practitioners:
> 1. Our **detailed curation recipe** outlines a systematic and generalizable approach to curate DPO datasets.
> 2. We study **what high quality means** for DPO datasets specifically.
>
> To this end, we dissect five open-source DPO datasets and find:
> 1. *Misaligned preference orders are common (20–30% of samples)* and we show that this directly degrades performance.
> 2. *Prompt quality and preference rewards are strongly correlated*, which we validate empirically rather than assuming it.
> 3. *Filtering based on a single signal (prompt quality or preference reward) is insufficient* and effective curation requires combining multiple signals.
>
> - Unlike prior work on synthetic data generation [1], we focus on **post-hoc curation of existing DPO datasets** without complex adjustments using our **simple but principled curation recipe**.
> - Finally, we provide **the first large-scale comparison** (5 datasets × 14 benchmarks × 8 models), which, to the best of our knowledge, has not been done before.
>
> We agree that our positioning could be better framed and will clarify these points in the revised version!
>
> ## 3. Additional Ablation Results
> - Thank you for highlighting the importance of understanding the contribution of the individual filtering steps of our recipe.
> - In App. F.4, we already evaluated UltraMix without the preference-reward filtering step and found that reward coherence is essential, improving overall performance by ~4%.
> - We agree that it is important to further clarify the individual contributions of our curation recipe and provide granular ablations for: **[P] preference-reward filtering only**, **[Q] input-quality filtering only**, **[Q+D] input-quality + difficulty filtering**, compared to the **full UltraMix recipe** that combines all steps.
>
> ### a) Llama
> |Model|LB1|LB2|Avg|
> |--|--|--|--|
> |Llama-SFT  (Starting Model)|62.85|37.15|50.09|
> |Llama-DPO (TuluDPO, Baseline)|66.30|40.66|53.96|
> |Llama-DPO (UltraMix) [P]|64.96|38.75|51.74|
> |Llama-DPO (UltraMix) [Q[|66.16|40.24|53.29|
> |Llama-DPO (UltraMix) [Q+D]|65.93|39.99|52.86|
> |**Llama-DPO (UltraMix)** |**68.50**|**42.75**|**56.04**|
>
> ### b) Qwen
> |Model|LB1|LB2|Avg|
> |--|--|--|--|
> |Qwen-SFT (Starting Model)|64.76|45.77|56.55|
> |Qwen-DPO (TuluDPO, Baseline)|66.73|48.53|59.56|
> |Qwen-DPO (UltraMix) [P]|65.56|47.32|57.77|
> |Qwen-DPO (UltraMix) [Q]|65.68|48.09|58.18|
> |Qwen-DPO (UltraMix) [Q+D]|66.87|48.08|58.79|
> |**Qwen-DPO (UltraMix)**|**69.28**|**51.28**|**62.05**|
>
> - The results show that no single step outperforms the TuluDPO baseline on its own.
> - [Q] and [Q+D] generally outperform [P], showing preference filtering is important but not the main driver.
> - Best performance is only achieved when every step is combined.
>
> These ablations clarify the role of each step in our curation recipe, demonstrating that all **stages are complementary** and the effectiveness of UltraMix stems from the **combination of signals**. We will give the full results on all benchmarks in the updated version!
>
> ## 4. Reward Model might be too good
> - The ablations above clarify that the effectiveness of our recipe stems from the **combination of signals** rather than relying solely on preference filtering with FsfairX.
> - We use FsfairX only to assess preference rewards due to its strong human alignment [2].
> - To avoid dependence on one reward model, we compared FsfairX with Skywork-Reward in **App. C.2**, showing that both models produce similar reward distributions for 2 datasets, thus **confirming our conclusions do not hinge on FsfairX**.
>
> While a full multi reward model study is beyond the scope, the consistent performance gains and new ablations show that UltraMix’s improvements come from **combining all components of our curation strategy**, not from the choice of reward model.
>
> [1] AIR: A Systematic Analysis of Annotations, Instructions, and Response Pairs in Preference Dataset
> [2] RLHF Workflow: From Reward Modeling to Online RLHF

---

### Author Response · Authors · 2025-11-27
**Global Response**

**We sincerely thank the reviewers for their feedback! To facilitate the AC’s decision, we provide a summary of our core contributions, key findings, and new results added during the rebuttal.**

## *Core Contributions*
We present a *rigorous, data-centric study and systematic curation recipe* that results in a new state-of-the-art DPO mixture:
1. **Scale and Rigor**: We provide the first large-scale DPO comparison across *5 popular datasets, 14 benchmarks, and 8 models* (1B-8B parameter range).
2. **Novel Curation Recipe**: We design a *systematic, scalable post-hoc curation recipe* that goes beyond simple quality filtering by combining it with Task Awareness and Reward Coherence, two novel signals critical for effective alignment.
3. **Efficiency & Performance**: Our curated UltraMix dataset consistently outperforms established baselines *while being 30% smaller*.
4. **Open Source Commitment**: We are releasing fully-annotated versions of all *5 source datasets, our 3 UltraMix variants, and our DPO-adapted annotation pipeline* to enable full reproducibility.

## *Key Findings*
Our systematic annotations helped us dissect DPO datasets and establish 3 critical findings:
1. **Misaligned preference orders are common**:  Even in high-quality datasets, 20–30% of samples have preference orders that contradict reward model signals (*chosen response is not always better than rejected one*), and we show in experiments that this directly degrades performance.
2. **Quality predicts reward**: We empirically validated that prompt quality and preference rewards are strongly correlated.
3. **Filtering based on a single signal (prompt quality or preference reward) is insufficient** and effective curation requires combining multiple signals.

Based on those, our **novel recipe** curates UltraMix (30% smaller than the best individual dataset + higher performance).

## *Distinctiveness from Prior Work*
We differ in three main aspects:
1. **Post-Hoc Curation vs. Generation**: Unlike *AIR* (or other data generators), our method focuses on post-hoc curation of existing datasets, and we show it is *scalable, easy to reuse and generalizable across datasets*, which makes it valuable for practitioners.
2. **Reward Coherence vs. Maximization**: Unlike RIP (which maximizes reward gaps and response lengths), we filter for samples where *a) chosen reward > rejected reward* and we *b) apply additional task and query filters*. We show in experiments (App. B.2) that reward coherence outperforms RIP and combining with additional filters gives even better results.
3. **Methodological Fairness**: We fix training hyperparameters across models, which is common in post-training to ensure performance differences are attributable strictly to data quality, providing a fair and rigorous comparison.

## *Rebuttal Additions & New Results*
Based on reviewer feedback, we have significantly strengthened the paper (updates highlighted in blue in the revision):
- **Importance of Preference Coherence (App. F.5)**: Tables 20 & 21 provide new results for Llama/Qwen on all 5 source datasets showing that simply removing misaligned pairs (preference coherence) improves performance across all benchmarks.
- **Granular Ablations (App. F.4)**: We added detailed ablations (Tables 18 & 19) proving that best performance is only achieved when *every step of our recipe is combined*. While quality and reward coherence filtering do improve performance, but alone are insufficient to beat the baseline, showing that our recipe is principled and grounded in our ablations.
- **Comparison to RIP (App. B.2)**: We added a direct comparison (Table 5) showing UltraMix outperforms RIP-style filtering by a large margin, proving that *reward coherence is a superior signal* to simple reward maximization. We also discuss other related works.
- **Theoretical Grounding (App. A.3)**: We added a section linking our findings to the theoretical mechanisms of DPO margins and noise reduction.
- **Reward Model Robustness (App. C.2)**: We verified that our findings hold across different reward models (FsfairX vs. Skywork), mitigating concerns about model dependence.

## *Reviewer Recognition*
We appreciate the reviewers' recognition of our work’s thoroughness and impact:
- *"The paper presents a nice study [...] and demonstrates that their recipe indeed improves performance."*
- *"Comprehensive analysis and experiments. Solid results."*
- *"Experimental evaluation is comprehensive, employing different benchmarks to assess model performance."*
- *"The paper’s detailed dataset analysis and creation of UltraMix reflect a high level of thoroughness."*
- *"Data Engineering for post-training is often overlooked but important."*

## *Final Comment*

We believe this work provides a *much-needed, rigorous foundation* for data-centric preference optimization.

We hope the AC agrees that the combination of our large-scale analysis, novel curation recipe, and open-source contributions *merits acceptance*.

---

> ### Author Response · Authors · 2025-12-01
> **Summary of Discussions and New Results**
>
> **In addition to our Global Response above**, we provide a summary of the **key discussions and the new experiments** added to the paper during the rebuttal period.
>
> ## *1. Resolution of Reviewer JCcE's Comments*
>
> The reviewer expressed overall satisfaction with our rebuttal (*"I am overall satisfied"*) pending two final clarifications, which we have **fully addressed in our latest responses**:
>
> 1. **Hyperparameter Justification**: The reviewer asked for clarification on our fixed hyperparameter settings as *they were unfamiliar with the common experimental setup* in post-training data-centric research. We provided a detailed response *citing four state-of-the-art post-training works* (Tulu 3, Unpacking DPO, Magpie, Zephyr) and offering *additional theoretical justification* to validate our experimental setup.
> 2. **New HelpSteer3 Experiment**: The reviewer requested an *optional comparison with RIP on the latest HelpSteer3 dataset*. Despite the short timeframe, we successfully annotated this dataset and ran the comparison. We showed that *our method outperforms RIP-style filtering* on this new dataset as well, demonstrating *high generalizability and consistency of our new preference coherence filtering method*.
>
> **We believe these additional results and clarifications directly resolve the reviewer's remaining inquiries.  We will open-source our newly annotated HelpSteer3 version.**
>
> ## *2. Resolution of Reviewer nUbS's Comments*
>
> The reviewer requested additional comparison to prior work (specifically RIP), which we **fully addressed in the revision** by adding:
>
> 1. **Direct Comparisons (UltraMix vs. RIP)**: New results in App. B.2 show that *preference coherence (our new method) significantly and consistently outperforms RIP-style baselines across Llama and Qwen models*.
> 2. **New Granular Ablations**: We also provided granular ablations for our UltraMix recipe (see updated App. F.4), proving that *our unique combination of signals (Preference Coherence + Task Awareness) is essential and non-trivial* compared to prior work.
>
> **These quantitative results provide the direct empirical evidence requested to differentiate our work from prior baselines.**
>
> ## *3. Conclusion*
>
> The rebuttal process has materially strengthened the paper with *new experiments (granular ablations, HelpSteer3), direct baseline comparisons (RIP), and theoretical grounding*. We respectfully ask the Area Chair to consider these **completed improvements** and the **explicitly positive trajectory of the reviewer discussions** when making their recommendation.

---

### Meta-Review · Area_Chair_m96X · 2026-01-09

**Summary:**

Summary of all reviewers' concerns:

1. The novelty of this paper is limited. The main takeaway of this paper is that data filtering is important, and we need to select high-quality and difficult data.

2. The quality filtering is conducted by an independent reward model, FsfairX. But it is unclear how much of the improvement actually comes from using this particular reward model vs. using the same reward model to filter examples from all datasets.

3. The method mixes many ingredients: data quality filtering, difficulty filtering, consistency filtering, etc., so it is difficult to tell which one actually helps. Some key ablations are missing.

4. Unclear Motivation. The claim that quality annotations are mostly missing is not accurate, as widely-used datasets like UltraFeedback and HelpSteer do provide fine-grained, human-annotated preference scores.

5. The current taxonomy requires refinement. Categorizing UltraFeedback and HelpSteer as instruction-following datasets is imprecise.

6. The benchmarking results are not fully convincing. The authors state that they fix the training hyperparameters for each model, but optimal hyperparameters would vary across datasets. There are also factors (e.g., dataset size) that could significantly impact model performance.

7. The analysis employing the MAGPIE pipeline lacks methodological novelty, as it relies on existing tools and reward models.

8. The heuristic selection of thresholds appears arbitrary. A more principled or experimentally justified approach is necessary.

9. Figure 3 is not convincing enough. Reward models are known to have biases and may not align perfectly with human judgment, while datasets like Helpsteer are annotated by humans and should be more reliable.

10. While the comparison of different datasets in Table 1 is useful, it does not account for size differences between datasets.

11. Since UltraFeedback is annotated by GPT-4o, it would be expected to have a higher level of accuracy. The paper should clarify why the preference signal in this dataset is weaker than anticipated.

12. The paper claims that model completion is correlated with prompt quality, but similar results have already been discussed in previous work, such as in [1]. Some related works have already discussed what contributes to the quality of a preference dataset. The authors can consider comparing with additional references such as  [2,3,4].

13. The study is highly empirical. A more theoretical analysis could help explain the findings and connect them to broader research questions.

**Reviewer Concerns:**

The rebuttal is thorough and largely addresses reviewers' concerns. In the rebuttal, the authors also added extensive new empirical results and provided detailed clarification of their major contributions. The authors have also incorporated their revisions into the updated manuscript. The authors present a data-centric study and systematic curation recipe, making a meaningful contribution on the empirical side.

**Reviewer Scores:**

Reviewer yvk1's initial rating is 6, which is already positive. Reviewer JCcE's original score was 4, but the reviewer is overall satisfied with the rebuttal and would consider raising the score accordingly. So I expect the score will at least increase from 4 to 6. Reviewer nUbS's original score was 4. Although new concerns were raised during the discussion, the authors addressed these concerns in great detail. Overall, I recommend accepting this paper.

---

### Decision · Program_Chairs · 2026-01-26

Accept (Poster)